# Long-term observations of cloud condensation nuclei over the Amazon rain forest – Part 2: Variability and characteristics of biomass burning, long-range transport, and pristine rain forest aerosols

Mira L. Pöhlker[1], Florian Ditas[1], Jorge Saturno[1,a], Thomas Klimach[1], Isabella Hrabě de Angelis[1], Alessandro C. Araùjo[2], Joel Brito[3,b], Samara Carbone[3,c], Yafang Cheng[1], Xuguang Chi[1,d], Reiner Ditz[1], Sachin S. Gunthe[4], Bruna A. Holanda[1], Konrad Kandler[5], Jürgen Kesselmeier[1], Tobias Könemann[1], Ovid O. Krüger[1], Jošt V. Lavrič[6], Scot T. Martin[7,8], Eugene Mikhailov[9], Daniel Moran-Zuloaga[1], Luciana V. Rizzo[10], Diana Rose[11,e], Hang Su[1], Ryan Thalman[12,f], David Walter[1], Jian Wang[12], Stefan Wolff[1], Henrique M. J. Barbosa[3], Paulo Artaxo[2], Meinrat O. Andreae[1,13], Ulrich Pöschl[1] and Christopher Pöhlker[1]

[1]*Multiphase Chemistry and Biogeochemistry Departments, Max Planck Institute for Chemistry, 55020 Mainz, Germany.*
[2] *Empresa Brasileira de Pesquisa Agropecuária (EMBRAPA), Trav. Dr. Enéas Pinheiro, Belém, PA, 66095-100, Brazil.*
[3] *Institute of Physics, University of São Paulo, São Paulo 05508-900, Brazil.*
[4] *EWRE Division, Department of Civil Engineering, Indian Institute of Technology Madras, Chennai 600036, India.*
[5] *Institut für Angewandte Geowissenschaften, Technische Universität Darmstadt, Germany.*
[6] *Department of Biogeochemical Systems, Max Planck Institute for Biogeochemistry, 07701 Jena, Germany.*
[7] *John A. Paulson School of Engineering and Applied Sciences, Harvard University, Cambridge, MA, USA.*
[8] *Department of Earth and Planetary Sciences, Harvard University, Cambridge, MA, USA.*
[9] *St. Petersburg State University, 7/9 Universitetskaya nab, St. Petersburg, 199034, Russia.*
[10] *Instituto de Ciências Ambientais, Químicas e Farmacêuticas, Universidade Federal de São Paulo (UNIFESP), Diadema, SP, Brazil.*
[11] *Institut für Atmosphäre und Umwelt, Goethe Universität, 60438 Frankfurt, Germany.*
[12] *Biological, Environmental & Climate Sciences Department, Brookhaven National Laboratory, Upton, NY 11973-5000, USA.*
[13] *Scripps Institution of Oceanography, University of California San Diego, La Jolla, CA 92037, USA.*
[a] *now at: Physikalisch-Technische Bundesanstalt, Bundesallee 100, D-38116 Braunschweig, Germany.*
[b] *now at: Laboratoire de Météorologie Physique, Université Clermont Auvergne, Aubière, France.*
[c] *now at: Institute of Agrarian Sciences, Federal University of Uberlândia, Uberlândia, Minas Gerais, Brazil.*
[d] *now at: Institute for Climate and Global Change Research & School of Atmospheric Sciences, Nanjing University, Nanjing, 210093, China.*
[e] *now at: Hessian Agency for Nature Conservation, Environment and Geology, Rheingaustr. 186, 65203 Wiesbaden, Germany.*
[f] *now at: Department of Chemistry, Snow College, Richfield, UT, USA.*

*Correspondence to:* M. Pöhlker (m.pohlker@mpic.de) and C. Pöhlker (c.pohlker@mpic.de)

**Abstract**

Size-resolved measurements of atmospheric aerosol and cloud condensation nuclei (CCN) concentrations and hygroscopicity were conducted over a full seasonal cycle at the remote Amazon Tall Tower Observatory (ATTO, March 2014 - February 2015). In a preceding companion paper, we presented annually and seasonally averaged data and parametrizations (Part 1, Pöhlker et. al. 2016a). In the present study (Part 2), we analyze key features and implications of aerosol and CCN properties for the following characteristic atmospheric conditions:

*(1)* Empirically pristine rain forest (*PR*) conditions, where no influence of pollution was detectable, as observed during parts of the wet season from March to May. The *PR* episodes are characterized by a bimodal aerosol size distribution (strong Aitken mode with $D_{Ait} \approx 70$ nm and $N_{Ait} \approx 160$ cm$^{-3}$, weak accumulation mode with $D_{acc} \approx 160$ nm and $N_{acc} \approx 90$ cm$^{-3}$), a chemical composition dominated by organic compounds, and relatively low particle hygroscopicity ($\kappa_{Ait} \approx 0.12$, $\kappa_{acc} \approx 0.18$).

*(2)* Long-range transport (*LRT*) events, which frequently bring Saharan dust, African biomass smoke, and sea spray aerosols into the Amazon Basin, mostly during February to April. The *LRT* episodes are characterized by a dominant accumulation mode ($D_{Ait} \approx 80$ nm, $N_{Ait} \approx 120$ cm$^{-3}$ vs. $D_{acc} \approx 180$ nm, $N_{acc} \approx 310$ cm$^{-3}$), an increased abundance of dust and salt, and relatively high hygroscopicity ($\kappa_{Ait} \approx 0.18$, $\kappa_{acc} \approx 0.35$). The coarse mode is also significantly enhanced during these events.

*(3)* Biomass burning (*BB*) conditions characteristic for the Amazonian dry season from August to November. The *BB* episodes show a very strong accumulation mode ($D_{Ait} \approx 70$ nm, $N_{Ait} \approx 140$ cm$^{-3}$ vs. $D_{acc} \approx 170$ nm, $N_{acc} \approx 3400$ cm$^{-3}$), very high organic mass fractions (~ 90 %), and correspondingly low hygroscopicity ($\kappa_{Ait} \approx 0.14$, $\kappa_{acc} \approx 0.17$).

*(4)* Mixed pollution (*MPOL*) conditions with a superposition of African and Amazonian aerosol emissions during the dry season. During the *MPOL* episode presented here as a case study, we observed African aerosols with a broad monomodal distribution ($D \approx 130$ nm, $N_{CN,10} \approx 1300$ cm$^{-3}$), with high sulfate mass fractions from volcanic sources (~ 20 %), and correspondingly high hygroscopicity ($\kappa_{<100nm} \approx 0.14$, $\kappa_{>100nm} \approx 0.22$) that were periodically mixed with fresh smoke from nearby fires ($D \approx 110$ nm, $N_{CN,10} \approx 2800$ cm$^{-3}$) with an organic-dominated composition and sharply decreased hygroscopicity ($\kappa_{<150nm} \approx 0.10$, $\kappa_{>150nm} \approx 0.20$).

Insights into the aerosol mixing state are provided by particle hygroscopicity ($\kappa$) distribution plots, which indicate largely internal mixing for the *PR* aerosols (narrow $\kappa$ distribution) and more external mixing for the *BB*, *LRT*, and *MPOL* aerosols (broad $\kappa$ distributions).

The CCN spectra (CCN concentration plotted against water vapor supersaturation) obtained for the different case studies indicate distinctly different regimes of cloud formation and microphysics depending on aerosol properties and meteorological conditions. The measurement results suggest that CCN activation and droplet formation in convective clouds are mostly aerosol-limited under *PR* and *LRT* conditions and updraft limited under *BB* and *MPOL* conditions. Normalized CCN efficiency spectra (CCN divided by aerosol number concentration plotted against water vapor supersaturation) and corresponding parameterizations (error function fits) provide a basis for further analysis and model studies of aerosol-cloud interactions in the Amazon.

## 1 Introduction

Clouds are a key factor in the Earth's atmosphere and climate system (Bony et al., 2015). Thus, sound scientific knowledge on the life cycle and highly dynamic properties of clouds is of significant importance for our understanding of atmospheric cycling and climate change (Seinfeld et al., 2016). A number of recent overview studies summarize the various facets of aerosol-cloud-precipitation-climate interactions in a detailed and comprehensive way (e.g., Andreae and Rosenfeld, 2008; Tao et al., 2012; Rosenfeld et al., 2014).

The Amazon Basin and its unique rain forest ecosystem are fundamentally shaped by the intense and large-scale (re)circulation of water between biosphere and atmosphere. Accordingly, the life cycle of shallow and deep convective clouds in the Amazon has been subject of numerous previous studies (e.g., Andreae et al., 2004; Freud et al., 2008; Rosenfeld et al., 2016b; Wendisch et al., 2016; Braga et al., 2017). In particular, the extent of anthropogenic influence on the cloud lifecycle through continuous land use change and combustion-related aerosol emissions has been actively debated (e.g., Roberts et al., 2003; Davidson et al., 2012; Goncalves et al., 2015). It is well established that the properties and dynamics of clouds can be fundamentally altered by changing cloud condensation nuclei (CCN) regimes, which are a fraction of the total (tropospheric) aerosol population (e.g., Rosenfeld et al., 2008; Reutter et al., 2009). To explore essential biogeochemical processes, such as aerosol-cloud interactions, in the Amazon rain forest, the Amazon Tall Tower Observatory (ATTO) has been established in 2010/11 (for details see Andreae et al., 2015). The central Amazon Basin is characterized by a pronounced seasonality in atmospheric composition in response to the north-south oscillation of the intertropical convergence zone. During the wet season, the ATTO site receives comparatively clear air masses of marine origin from the northeast that travel over mostly untouched rain forest areas, whereas during the dry season strongly polluted air masses are advected from the southeast, originating from numerous fires in the Amazon's arc of deforestation (for details see C. Pöhlker et al., 2018). Detailed information on characteristic differences in atmospheric state and processes for the contrasting wet vs. dry season conditions in the ATTO region can be found in a number of recent studies (e.g., Nölscher et al., 2016; M. Pöhlker et al., 2016; Moran-Zuloaga et al., 2017; Saturno et al., 2017a, Yañez-Serrano et al., 2018).

In terms of microphysical processes in cloud formation and development, the number concentration of CCN, $N_{CCN}(S)$, and the peak water vapor supersaturation, $S$, at the cloud base play a key role. Here, $S$ is predominantly determined by the updraft velocity, $w_b$, of the adiabatically rising air parcel at the cloud base. The relevant peak $S$ levels in the Amazon are assumed to range from ~0.1 % to ~1.0 % (e.g., Andreae, 2009; Reutter et al., 2009; Pöschl et al., 2010; C. Pöhlker et al., 2012; Farmer et al., 2015; M. Pöhlker et al., 2016). Moreover, a recent study suggests that also substantially higher $S$ ($\gg 1$ %) can be reached in deep convective clouds under certain conditions (Fan et al., 2018). However, a systematic and quantitative assessment of relevant peak $S$ distributions in the Amazonian atmosphere is still lacking (see discussion in Sect. 3.9). Depending on $N_{CCN}(S)$ and $S$, a certain number of cloud droplets at the cloud base, $N_{db}$, is formed (Rosenfeld et al., 2016b). In the Amazon Basin, $N_{db}$ ranges from few hundred droplets per cubic centimeter for clean to 1000 and 2000 cm$^{-3}$ for polluted conditions (Pöschl et al., 2010; Rosenfeld et al., 2016a; Braga et al., 2017). Upon cloud development and rising air masses, the initial droplets grow by condensation of water vapor, which can be observed as changes in the cloud drop size distribution (DSD). Thus, the DSD is a function of thermodynamic parameters (i.e., the updraft velocity $w_b$), aerosol conditions (i.e., $N_{CCN}(S)$), and the cloud evolution (i.e., the cloud depth $H$). Important bulk DSD properties are, in particular, the droplet number concentration, $N_d$, and the effective droplet radius $r_e$. For $r_e$

> 11 µm, the probability of droplet collision and coalescence processes increases to significant levels and warm rain formation is initiated (Cecchini et al., 2017b).

A detailed analysis of the properties and variability of the Amazonian CCN population is a prerequisite for the understanding of cloud cycling in this region. However, the CCN data from the basin is still sparse. Therefore, we conducted a systematic characterization of the trends and properties of the central Amazonian CCN population at the ATTO site. The first half of this study has been published recently as part 1 (M. Pöhlker et al., 2016). The present manuscript represents part 2 and focuses on the variability and properties of periods and conditions that are characteristic for the Amazonian atmosphere.

## 1.1 Brief summary of companion part 1 paper

The part 1 paper (M. Pöhlker et al., 2016) focuses on the multi-month variability in the Amazonian CCN population by presenting data from a full seasonal cycle. In particular, it presents annually averages of the key CCN parameters, a detailed analysis of the specific seasonal as well as diurnal cycles and a systematic analysis of different CCN parametrization schemes to represent the Amazonian CCN cycling in modeling studies.

The major findings of part 1 can be summarized as follows: (i) The CCN population in the central Amazon is predominantly defined by the overall aerosol concentration as well as the shape of the characteristic bimodal aerosol size distribution. Accordingly, a key property that has to be taken into account for the interpretation of the CCN results is the relative proportion of the Aitken and accumulation modes (mode maxima at $D_{Ait} \approx 70$ nm and $D_{Acc} \approx 150$ nm). (ii) The hygroscopicity parameters, $\kappa(S,D_a)$ with $D_a$ as midpoint activation diameter, of the two modes were found to be remarkably stable for most of the observation period ($\kappa_{Ait} = 0.14 \pm 0.03$ vs. $\kappa_{Acc} = 0.22 \pm 0.05$) with only weak seasonal and no diel variability. Accordingly, we concluded in part 1 that the shape of the aerosol size distribution is the predominant factor, whereas $\kappa(S,D_a)$ is only of secondary importance for the variability in the Amazonian CCN population, in agreement with previous studies (see references in M. Pöhlker et al., 2016). (iii) Furthermore, part 1 summarizes the CCN key parameters that allow an efficient modeling of the Amazonian CCN population. The prediction of CCN concentrations is particularly reliable when time series of total aerosol concentration and/or the aerosol size distribution are available.

We emphasized CCN efficiency spectra, which can be regarded as CCN signatures for a particular aerosol population, by describing their behavior for the atmospherically relevant $S$ range. Here, a rather simple analytical expression (i.e., single- or double-error function fits) suffice to represent the essence of the CCN-relevant properties of an aerosol population, which includes the characteristic shape of the aerosol size distribution and the $\kappa(S,D_a)$ size dependence. Furthermore, the CCN efficiency spectra are independent of the total aerosol number concentration (in contrast to CCN spectra) and, thus, can be flexibly scaled to the concentration range of interest to obtain CCN concentrations at a certain $S$ level. Finally, and beyond their potential use in models as CCN parametrization, the shape of the CCN efficiency spectra is very instructive to visualize the specific behavior of contrasting aerosol population in cloud formation. This aspect will be one focal point of the present study.

## 1.2 Aims and scope of this study

To complete the analysis started in part 1, this manuscript analyzes the CCN variability in original time resolution (~4.5 h), which is sufficient to resolve its short-term variability in relation to air mass changes as well as aerosol emission and transformation processes. In the present work, we will zoom into specific periods of the one-year CCN data set in two steps: First, we discuss the aerosol and CCN variability for two contrasting 2-months

periods that characterize the pollution minimum and maximum in relation to complementary trace gas and aerosol parameters. Second, we will analyze the following four case studies that represent characteristic events and conditions in the central Amazon region:

- During certain wet season episodes, when no tracers of pollution aerosols are detectable any more, the aerosol population can be regarded as empirically not distinguishable from pristine, i.e., completely unpolluted rain forest conditions. This empirically pristine state of the rain forest (*PR*) aerosol prevails during 10 to 40 days per year (depending on *PR* definition, see Sect. 2.7).

- Long-range transport (*LRT*) aerosol advection during the wet season brings Saharan dust, African biomass burning smoke, as well as marine aerosol particles from the transatlantic passage. The *LRT* case study represents conditions that prevail between 50 and 60 days per year (see Moran-Zuloaga et al., 2017).

- Biomass burning (*BB*) smoke from man-made forest fires in the various deforestation hotspots in the basin influences the atmospheric state at ATTO almost permanently during the dry season and for extended episodes during the transition periods (>100 days per year) (Saturno et al., 2017a). The *BB* case study in this work analyzes large deforestation fires in the southeastern basin, whose smoke reached ATTO after few days of atmospheric processing. Accordingly, the *BB* case study characterizes the typical conditions of aged smoke influencing the atmospheric state at ATTO.

- Mixed pollution (*MPOL*) from African *LRT* and local/regional fires represents a frequent aerosol scenario at ATTO (Saturno et al., 2017a). The advected African aerosols mainly comprise biomass and fossil fuel combustions emissions, although the exact composition of these dry season *LRT* plumes is still poorly analyzed. The *MPOL* case study focusses on a period when African volcanogenic aerosols were advected to ATTO – an event that has been well documented in Saturno et al., 2017b. We selected this episode since the microphysical properties of the volcanogenic aerosol are characteristic enough to discriminate them from the local/regional smoke emissions. Accordingly, the alternating pattern of *LRT* vs. local/regional pollution can be clearly resolved for the *MPOL* period. However, note that volcanogenic plumes are comparatively rare events, whereas African combustion emissions, which are much harder to discriminate from the local/regional emissions, are a more common scenario. Accordingly, the *MPOL* case study is an example for a complex aerosol mixture due to alternating African vs. local/region influences during the dry season.

In summary, the overall purpose of this study is to link the measured CCN abundance and properties with the characteristic emissions and transformation processes that govern the Amazonian aerosol population. With the CCN parametrization strategies, developed in part 1, we provide a basis for effective CCN prediction under characteristic aerosol and CCN conditions in the Amazon Basin.

## 2 Experimental section

### 2.1 Aerosol and trace gas measurements at the ATTO site

The present study is mostly based on *in situ* measurements at the remote ATTO site, which has been described in detail by Andreae et al. (2015). Further relevant information regarding the site, the measurement period, and the aerosol and trace gas instrumentation can be found in the part 1 paper by M. Pöhlker et al. (2016). The time frame of the present analysis, including the four in-depth case studies, overlapped with the two intensive observation periods (IOPs) of the international Green Ocean Amazon 2014/5 (GoAmazon2014/5) campaign (Martin et al., 2016a; 2016b). Specific details on the measurements of equivalent black carbon ($BC_e$) mass concentrations, $M_{BCe}$, with the multi-angle absorption photometer (MAAP) can be found in Saturno et al. (2017a; 2017c). Specific details on the measurements of the mole fraction of carbon monoxide (CO), $c_{CO}$, with a G1302 analyzer (Picarro Inc. Santa Clara, CA, USA) can be found in Winderlich et al. (2010).

Details on the aerosol chemical speciation monitor (ACSM, Aerodyne Research Inc., Billerica, MA, USA) measurements, which provide online information on the mass concentrations, $M_{species}$, of organics (OA), sulfate ($SO_4^{2-}$), nitrate ($NO_3^-$), ammonium ($NH_4^+$), and chloride ($Cl^-$), can be found in Ng et al. (2011). A detailed description of the long-term operation of the ACSM at the ATTO site can be found in Carbone et al. (2017). For the selected case study and seasonal time frames, we calculated the mean values of $M_{species}$ as well as corresponding mass fractions, $f_{species}$, according to

$$f_{species} = \frac{M_{species}}{M_{OA} + M_{SO4} + M_{NH4} + M_{NO3} + M_{Cl} + M_{BCe}} \qquad (1)$$

Furthermore, predicted hygroscopicity parameters, $\kappa_p$, were calculated based on the ACSM and MAAP results according to the following procedure adapted from Gunthe et al. (2009), Rose et al. (2011), and M. Pöhlker et al. (2016):

$$\kappa_p = f_{OA} \cdot 0.1 + f_{inorg} \cdot 0.71 + f_{BCe} \cdot 0 \qquad (2)$$

with $f_{inorg}$ including $SO_4^{2-}$, $NO_3^-$, $NH_4^+$, and $Cl^-$. Note that $M_{NH4}$ ranged below its detection limit (i.e., 0.28 µg m$^{-3}$, for 30 min averaging time) during the clean Amazon wet season months, making the obtained results unreliable (the questionable periods are marked in Table 3, which is discussed later in this study). For these periods with questionable results, $M_{NH4}$ was omitted in the calculation of the mass fractions, which has to be kept in mind in the interpretation of the results. Also $M_{Cl}$ and $M_{BCe}$ were below detection limits for certain conditions. Accordingly, $\kappa_p$ was calculated without $M_{NH4}$ and/or $M_{Cl-}$ and/or $M_{BCe}$. The nominal size range of the ACSM spans from 75 to 650 nm and the measurements are conducted size-integrated. Accordingly, the ACSM results tend to be dominated by larger particles with relatively high masses, which makes the $M_{species}$ results mostly representative for the accumulation mode composition. Accordingly, the calculated $\kappa_p$ was compared to the hygroscopicity parameter for the lowest measured $S$ level, $\kappa(0.11\,\%)$, corresponding to the largest measured critical diameter ($D_a \approx$ 170 nm).

### 2.2 CCN measurements and data analysis

A detailed description of the operation of the CCN counter (CCNC) and the subsequent data analysis can be found in the companion paper (M. Pöhlker et al., 2016), which is the basis for the present study. Briefly, size-resolved CCN measurements were conducted using a continuous-flow streamwise thermal gradient CCN counter (model CCN-100, DMT, Longmont, CO, USA) in combination with a differential mobility analyzer (model M, Grimm Aerosol Technik, Ainring, Germany) and a condensation particle counter (Grimm Aerosol Technik). The

DMA-selected size range spans from 20 to 245 nm. The analyzed supersaturation range spans from 0.11 to 1.10 %. A complete measurement cycle with scanning of all particle diameters and supersaturations took ~ 4.5 h. For further CCN-relevant information, we refer the reader to Rose et al. (2008) and Krüger et al. (2014).

The CCN efficiency spectra parameterization as introduced in part 1 plays a key role in the present manuscript. Note that we slightly revised and improved the fitting procedure from the companion part 1 paper. The main change implies that the fits are now forced through zero, which is physically more plausible and makes the single-error function (erf) fit parametrization more applicable for modelling studies. The erf fit (mode = 1) is represented by the following function

$$\frac{N_{CCN}(S)}{N_{CN,10}} = \frac{a_1}{2} + \frac{a_1}{2} \cdot erf\left(\frac{ln\frac{S}{S_1}}{w_1}\right) \qquad (3)$$

with $a_1$ as prefactor, $S_1$ as the supersaturation, at which half of the maximum activated fraction (*MAF*) of the aerosol particles acts as CCN (e.g., 50 % for $a_1 = 1$), and $w_1$ as the width of the erf fit. To simplify the fitting procedure, $a_1 = 1$ was assumed. For $a_1 = 1$ the erf converges against unity, corresponding to an activation of all particles as CCN at high $S$, which is adequate in most cases. Analogously, the double-erf fit (mode = 2) is represented by the function

$$\frac{N_{CCN}(S)}{N_{CN,10}} = \frac{a_1}{2} + \frac{a_2}{2} \cdot erf\left(\frac{ln\frac{S}{S_1}}{w_1}\right) + \left(\frac{a_1-a_2}{2}\right) \cdot erf\left(\frac{ln\frac{S}{S_2}}{w_2}\right) \qquad (4)$$

with the index 1 and 2 specifying the variables for both modes. To simplify the fitting procedure, $a_1 = 1$ was assumed.

Note further that in part 1 we tested different reference aerosol number concentrations, $N_{CN,Dcut}$ (e.g., $N_{CN,10}$ and $N_{CN,50}$), for the CCN efficiency spectra parametrization. In this study, we use only one reference concentration for clarity – namely $N_{CN,10}$. The choice of $N_{CN,10}$ can be explained by the fact that it is experimentally rather easily accessible (e.g., via condensation particle counter, CPC, measurements), whereas reference concentrations such as $N_{CN,50}$ require more elaborated experimental setups (e.g., scanning mobility particle sizer, SMPS, data).

The $\kappa$ distributions were calculated according to the procedure reported in Su et al. (2010) for every individual CCN measurement cycle and subsequently averaged for time periods of interest. The corresponding $N_{CN}$ $\kappa$ distributions were obtained by multiplication of the average $\kappa$ distributions with the average $N_{CN}$ size distributions within the same time frame. The entire CCN analysis has been conducted in IGOR Pro.

### 2.3 Backward trajectories

The backward trajectory (BT) analysis and classification in this study has been adopted from C. Pöhlker et al. (2018), where an in-depth description of the procedure can be found. Briefly: The BT analysis is based on the hybrid single particle Lagrangian integrated trajectory model (HYSPLIT, NOAA-ARL) with meteorological input data from the global data assimilation system (GDAS1) (Draxler and Hess, 1998). Three-day BTs have been calculated every 1 h with a starting height of 1000 m above ground level (AGL) at the ATTO site for the time period of 1 January 2008 until 30 June 2016. A sensitivity test confirmed that starting heights of the BTs at 200 and 1000 m AGL gave similar results. Accordingly, the chosen start height at 1000 m appears to be a good representation of the origin of the boundary layer air masses at ATTO. Subsequently, the resulting ensemble of 74 496 individual BTs has been classified into 15 clusters with *k*-means cluster analysis (CA). The supplement Fig. S1 shows a map of the northeastern Amazon Basin with the ATTO site and the mean track of the 15 BT

clusters. It illustrates that the air masses arrive almost exclusively in a rather narrow easterly wind sector (between 45° and 120°). Within this sector, four main directions of air mass advection can be identified: (i) a north-easterly (NE) track including the clusters NE1, NE2, and NE3; (ii) an east-northeasterly (ENE) track including the clusters ENE1, ENE2, ENE3, and ENE4; (iii) an easterly (E) track including the clusters E1, E2, E3, and E4; and (iv) a group of 'inland' trajectories in east-southeasterly (ESE) directions including clusters ESE1, ESE2, and ESE3 as well as one cluster towards the southwest (i.e., SW1). For a detailed characterization of the land cover, including potential trace gas, aerosol, and CCN sources, within the BT-derived footprint region of the ATTO site, we refer the reader to C. Pöhlker et al. (2018).

## 2.4 Satellite data and analysis

The satellite data products used in this study were obtained from the NASA Giovanni web interface (http://giovanni.gsfc.nasa.gov/, last access 26 May 2017), developed and maintained by the NASA Goddard Earth Sciences Data and Information Services Center (GES DISC) (Acker and Leptoukh, 2007). The following satellite products have been used:

- Cloud top temperature data (AIRX3STD_v006 product) from the atmospheric infrared sounder (AIRS) instruments on board of the satellites Terra and Aqua (data included from 04 Jul 2002 to 30 Jun 2016). For the corresponding time series in this study, the Aqua and Terra data were averaged per day for a representative region (i.e., $ROI_{ATTO}$, see below).

- Cloud cover data is obtained by the moderate resolution imaging spectroradiometers (MODIS) on board of the Terra and Aqua satellites (included data from 04 Jul 2002 to 30 Jun 2016). The obtained Aqua and Terra time series were averaged for the $ROI_{ATTO}$. Note that cloud cover strongly depends on the spatial resolution of the instrument as outlined in King et al. (2013).

- Cloud droplet effective radius, $r_e$, is calculated from MODIS products (included data from 04 Jul 2002 to 30 Jun 2016) for the $ROI_{ATTO}$. Since $r_e$ varies with vertical cloud development and total CCN abundance, we filtered the $r_e$ data by cloud top temperature (King et al., 2013).

- Precipitation rate data, $P_{TRMM}$, is obtained from the tropical rainfall measuring mission (TRMM) within the $ROI_{ATTO}$. The TRMM_3B42_Daily_v7 product has been used for the time period 01 Jan 1998 until 30 Jun 2016.

The satellite data were used as time series of area averaged data products within a region of interest around the ATTO site ($ROI_{ATTO}$: 59.5° W to 54° W and 3.5° S to 2.4° N) as specified in Fig. S1.

## 2.5 Seasonal cycles of remote sensing and in situ data

To provide an overall picture of the seasonal cycle of selected aerosol, meteorological, and cloud microphysical parameters representative for the ATTO region, various multi-year data sets were analyzed and compared. Remote sensing data products were used in the time frames outlined in section 2.4. For $M_{BCe}$, $c_{CO}$, as well as the particle concentrations in the accumulation mode range, $N_{acc}$, the Aitken mode range, $N_{Ait}$, and in the total particle population, $N_{CN,10}$, 4-6 years of ATTO site measurements were available. Additionally, $M_{BCe}$ data measured at the ZF2 site, located 40 km NW of Manaus, were used to reflect the conditions back to 2008 (e.g., Martin et al., 2010a). In terms of sources and conditions, the ATTO and ZF2 sites are comparable (Saturno et al., 2017a; C. Pöhlker et al., 2018). Accordingly, combined $M_{BCe}$ time series from ATTO and ZF2 spanning from Jan 2008

to May 2017 were included here. The $c_{CO}$ data include ATTO measurements from Mar 2012 to Apr 2017. The $N_{acc}$, $N_{Ait}$, and $N_{CN,10}$ data are based on SMPS measurements at ATTO from Feb 2014 to Jan 2017.

### 2.6 Aerosol sampling and scanning electron microscopy with x-ray spectroscopy

Aerosol samples for electron micro-spectroscopy were collected by impaction. A home-made single stage impactor (flow rate = 1-1.5 lpm; and nominal cut-off $D_{cut} \approx 500$ nm) was used for collection. The collection efficiency below $D_{cut}$ decreases steeply, however, a certain fraction of particles in this size range is still collected. Moreover, a fraction of very small particles is additionally collected via diffusive deposition and therefore available for the microscopic analysis. Aerosol particles were deposited onto silicon nitride substrates ($Si_3N_4$, membrane width 500 µm, membrane thicknesses 100 or 150 nm, Silson Ltd., Northhampton, UK). Afterwards samples were stored in airtight containers at -20 °C immediately after sampling.

Without further treatment like sputter-coating, particles were analyzed in a high-resolution scanning electron microscope (SEM; FEI Quanta 200F, FEI, Eindhoven, The Netherlands). An acceleration voltage of 12.5 kV with a spot size of approximately 3 nm was used. X-ray emission was analyzed with an energy-dispersive X-ray analysis (EDX; Edax Genesis, Edax Inc.). The system is able to record characteristic X-ray emissions for all elements with $Z > 5$. Obviously, in the present work Si could not be quantified due to the silicon substrate (Kandler et al., 2011).

### 2.7 Definition of empirically pristine rain forest aerosol conditions at ATTO

The term pristine is bound to a pre-human reference state with prevailing natural atmospheric conditions, in the absence of any anthropogenic influences. Andreae (2007) pointed out that in the present-day atmosphere "there are no places where we can expect to find truly pristine conditions". This is particularly true with respect to long-lived trace gases, such as $CO_2$ and $CH_4$, which have accumulated in the atmosphere due to man-made activities. However, also the aerosol abundance and composition is substantially perturbed by anthropogenic emissions. This also includes aerosols at remote locations, which are altered to varying degrees by a globally pervasive background pollution. It has been controversially debated if and to what extent certain marine and remote continental locations still approximate pristine atmospheric conditions (e.g., Andreae, 2009; Martin et al., 2010b; Pöschl et al., 2010; Chi et al., 2013; Hamilton et al., 2014). Frequently, conditions with low anthropogenic influences are characterized as 'clean' or 'near-pristine', although these terms are rather differently and often vaguely defined. The discussion is inherently difficult since truly pristine conditions are not available any more to quantify how close the present-day atmosphere still gets to its original pre-human state. Accordingly, truly pristine conditions remain hypothetical and any degree of a near-pristine state has to be defined indirectly with respect to the absence of man-made emissions.

In the Amazon Basin, aerosol conditions during most of the wet season are comparatively clean with low – though detectable – concentrations of background pollution. During the multi-months wet season period that can overall be characterized as near-pristine, certain episodes can be identified where pollution concentrations drop even further and below analytical detection limits. Under these conditions, the aerosol composition can be regarded as empirically undistinguishable from a pristine state. The present study uses these episodes to operationally define a pristine state of the aerosol composition and properties in a tropical rain forest environment, which we call empirically pristine rain forest (*PR*) aerosol.

The selection of a robust marker is crucial for the assessment of the *PR* state. In previous studies, the concentrations of BC, CO, and CN as well as the air mass origin by means of BTs have been used as corresponding markers at other locations, however all of them are characterized by certain limitations (Hamilton et al., 2014 and reference therein). For the present study, we explored the suitability of BC, CO, CN, and BTs as well as

combinations of them as *PR* markers. Their strengths and limitations can be summarized as follows:

- BT-based *PR* filter: BTs help to identify periods with comparatively clean air mass advection due to fact that those BTs, which reach furthest to the northeast across the South American continent, overpass the largest fraction of uninhibited and, thus, untouched forest regions (C. Pöhlker et al., 2018). Accordingly, a BT filter allows to exclude local and regional sources of anthropogenic pollution. However, BTs do not allow to relia-

10 bly filter out long-range transport aerosols, particularly from African sources. Accordingly, BTs are not suitable as a stand-alone *PR* filter, however, they are useful in combination with other *PR* markers as outlined below.

- CN-based *PR* filter: The total particle concentration *per se* is not a good filter for clean or (near-)pristine states since natural aerosol conditions range from very low (e.g., $N_{CN}$ in Antarctica reaching down to

15 10 cm$^{-3}$; Fiebig et al., 2014) to very high concentrations (e.g., $N_{CN}$ in the upper troposphere reaching above 10 000 cm$^{-3}$; Andreae et al., 2018). Accordingly, $N_{CN}$ is also not a good stand-alone *PR* filter. However, with respect to the 'typical' Amazonian aerosol concentrations in the boundary layer (i.e., few hundreds up to few thousands of particle per cm$^3$), the total particle concentrations provides at least a helpful reference level for the extent of pollution.

- BC-based definition of *PR*, called *PR*$_{BC}$: BC represents a unique indicator for combustion aerosol particles, which in the Amazon Basin almost entirely originate from anthropogenic sources (i.e., fossil fuel and biomass burning in South American and Africa). Accordingly, we defined *PR*$_{BC}$ episodes by means of the absence of detectable BC$_e$. Petzold and Schönlinner (2004) determined the detection limit of the MAAP "as the minimum resolvable absorbance" by considering "the variability of blank filter optical properties". The

detection limit corresponds to the resulting mean absorbance of the blank filter + 3 times the standard deviation (std) resulting in an absorption coefficient of 0.132 Mm$^{-1}$ for 30-min averages. The $M_{BCe}$ was calculated by using mass absorption cross sections (MAC) retrieved by fitting MAAP absorption coefficients at 637 nm and single particle soot photometer (SP2) rBC mass measurements during the different seasons as explained in Saturno et al. (2017a). Using the MAC values measured at ATTO (MAC for the wet season:

11.4 ± 1.2 m$^2$ g$^{-1}$; MAC for the dry season: 12.3 ± 1.3 m$^2$ g$^{-1}$), the conversion to $M_{BCe}$ corresponds to 0.011 – 0.012 µg m$^{-3}$. Note that this threshold would be higher if a "traditional" MAC value of 6.6 m$^2$ g$^{-1}$ were used to calculate $M_{BCe}$ (~0.019 µg m$^{-3}$). Here, we selected a threshold concentration of $M_{BCe}^{*}$ = 0.01 µg m$^{-3}$, when the ATTO-specific MAC values are taken into account. Accordingly, *PR*$_{BC}$ periods fulfill the criterion $M_{BCe} < M_{BCe}^{*}$ for ≥6 h (after applying a 5-point running average to the 1 h $M_{BCe}$ data).

- CO-based definition of *PR*, called *PR*$_{CO}$: The BC approach has the potential drawback that it may overemphasize periods with strong precipitation, since heavy rain removes the BC/aerosol content in actually polluted air masses irrespective of the gas phase composition (see Hamilton et al., 2014). Accordingly, the *PR*$_{CO}$ filter is based on the gas phase combustion marker, CO. In our definition, *PR*$_{CO}$ conditions prevailed during periods, in which the ATTO $c_{CO}$ concentrations dropped below the monthly background CO levels at

the following reference stations (https://www.esrl.noaa.gov/gmd/dv/site/CPT.html, last access 07 Apr 2018): Whenever the ATTO $c_{CO}$ data was associated with northern hemispheric BTs (i.e., NE or ENE BT clusters,

Fig. S1) and dropped below the average of the monthly CO levels obtained at the three background stations at Ragged Point in Barbados (RPB), Assekrem in Algeria (ASK), and Izaña in Cape Verde (IZO), the episode was flagged as $PR_{CO}$. Analogously, whenever the ATTO $c_{CO}$ data was associated with southern hemispheric BTs (i.e., E and ESE BT clusters) and dropped below the monthly CO levels obtained at the background station on Ascension Island (ACS), the episode was also flagged as $PR_{CO}$. Note that the time series of monthly $c_{CO}$ levels from the reference stations was linearly interpolated to hourly values for the comparison with ATTO $c_{CO}$ data. The BTs from the northern hemisphere (i.e., NE and ENE BT clusters) account for ~70 % of all BTs during the wet season. Among these ~70 %, ~30 % can be attributed to the NE BT clusters. The locations of the CO background stations, the wet season ATTO BT ensemble, as well as the hemispheric CO distribution is shown in Fig. S2.

The seasonality in the frequency of occurrence, $f$, of the $PR_{BC}$ and $PR_{CO}$ filters is generally very similar, as shown in Fig. 1: The first $PR_{BC}$ and $PR_{CO}$ episodes occur in February. Their highest $f$ is reached around the second half of April and the first half of May. Afterwards, $f$ decreases steeply and $PR$ conditions occur only occasionally in June and July. Overall, $PR_{BC}$ and $PR_{CO}$ episodes overlap for ~25 % of the total time that is flagged by at least one of the two filters. Since both approaches are associated with certain limitations and uncertainties, we further analyzed two combinations of $PR_{BC}$ and $PR_{CO}$: the union of both sets ($PR_{BC \cup CO}$) and the intersection of both sets ($PR_{BC \cap CO}$) as shown in Fig. 1. For the in-depth analysis of aerosol and CCN properties in this work, we chose the $PR_{BC \cap CO}$ filter, which we consider as most strict and, thus, robust since the criteria of both filters have to be fulfilled.

With respect to the $PR$ filters, the following aspects are worth mentioning: $PR$ episodes at ATTO exclusively occur during the wet season due to a combination of two effects: First, the ATTO site is very remote and the characteristic wet season air mass advection occurs mostly over uninhabited areas (see C. Pöhlker et al., 2018). Second, the high precipitation rates entail strong scavenging and, thus, relatively short aerosol particle lifetimes, which reduces the long-range transport of the background pollution aerosol load. In contrast, dry season $PR$ episodes have almost never been observed at ATTO due to the extensive biomass burning emissions in South America and Africa in combination with low scavenging rates and, thus, long atmospheric aerosol life times. This is relevant since wet vs. dry season conditions have likely been associated with different atmospheric states, for example with respect to VOC concentrations and aerosol populations as well as photochemical conditions. While wet season $PR$ states are still experimentally accessible as outlined in this work, the dry season $PR$ state appears to be entirely swamped by pollution.

The primary filters are based on the combustion markers BC and CO, which do not allow a discrimination between wild vs. man-made fires. Accordingly, the contribution of wild fire emissions, which were part of an originally pristine atmosphere in the Amazon, are erroneously filtered out. Generally, wild fires in the tropical Amazonian forests are rare events due to the fact that the dense and moist canopies – if unperturbed – effectively maintain a fire-immune state (e.g., Cochrane, 2003; Nepstad et al., 2008). Nevertheless, wild fires play a certain, although minor, role for the ATTO observations, since the wet season BTs cover the Guianan savanna ecoregions that account for ~8 % of the ATTO footprint region (Olson et al., 2001; C. Pöhlker et al., 2018). Within these regions, the infrequent occurrence of wild fires is part of the savanna-specific fire regime (de Carvalho and Mustin, 2017). Figure S3 illustrates the relevance of wild fires for ATTO under wet season conditions.

Moreover, Saharan dust as well as marine aerosols from the Atlantic Ocean are advected towards the ATTO region via wet season LRT plumes, which are most frequent in February and March (Moran-Zuloaga et al.,

2017). Saharan dust and marine aerosols can also be considered as part of an original pristine atmospheric state in the Amazon. However, all LRT plumes arriving at ATTO are filtered out by the $PR_{BC}$ and $PR_{CO}$ approaches due to the fact virtually all LRT plumes contain a substantial fraction of smoke from mostly man-made fires in West Africa (Moran-Zuloaga et al., 2017). Accordingly, the role of the present-day LRT plumes arriving at

ATTO has to be differentiated carefully as they represent a (partly internal) mixture of natural and anthropogenic aerosols.

## 3   Results and discussion

### 3.1   Aerosol and cloud microphysical seasonality in the Amazon

Prior to the in-depth analysis of the aerosol and CCN cycling for characteristic conditions, this section provides an overview of the aerosol and cloud microphysical seasonality in the ATTO region. The pollution markers $c_{CO}$ and $M_{BCe}$ in Fig. 2a show a pronounced seasonal cycle with a prevalence of clean conditions in the wet season vs. the biomass burning-related pollution maximum in the dry season (Andreae et al., 2015). The annual minimum in $M_{BCe}$ levels occur at the end of April (with weekly $M_{BCe}$ means of ~0.03 µg m$^{-3}$), whereas its highest

concentrations were observed in August and September (with weekly $M_{BCe}$ means of ~0.40 µg m$^{-3}$). The seasonal cycle of $c_{CO}$ shows a temporal shift of about 1 month with its minimum in the beginning of June (with weekly $c_{CO}$ means ~100 ppb) and largest values from October to December (with weekly $c_{CO}$ means ~200 ppb). The phase shift between the $c_{CO}$ and $M_{BCe}$ seasonality can be explained by the spatiotemporal interplay of combustion-related BC and CO sources, aerosol wet scavenging, as well as alternating advection of northern vs. south-

ern hemispheric (NH vs. SH) air masses (Martin et al., 2010b; Andreae et al., 2012; 2015). Similar to the $M_{BCe}$ trends, the total aerosol particle concentration, $N_{CN,10}$, tracks the seasonality in biomass burning activities (in South America and Africa) with lowest concentrations in March and April ($N_{CN,10}$ weekly means from 200 to 300 cm$^{-3}$) and highest values between August and November ($N_{CN,10}$ weekly means from 1000 to 2000 cm$^{-3}$) as shown in Fig 2b. The CCN concentrations at a supersaturation of 0.5 %, $N_{CCN}(0.5\,\%)$, which were calculated

based on long-term SMPS data and the $\kappa$-Köhler parametrization as outlined in our part 1 paper, mostly tracked the seasonal trends in $N_{CN,10}$. Its minimum around March and April showed weekly mean $N_{CCN}(0.5\,\%)$ values around 200 cm$^{-3}$, whereas the maximum showed values between 1000 and 2000 cm$^{-3}$.

Figure 2c shows the seasonal cycles of two precipitation data products: First, $P_{TRMM}$ data represents the area-averaged precipitation rate in the ROI$_{ATTO}$ (see Fig. S1). The $P_{TRMM}$ data reveal a rather broad wet season precip-

itation maximum from March to May. The smallest precipitation rates are observed from September to November. Second, the $P_{BT}$ data represents the cumulative precipitation along the BTs arriving at ATTO (for details see C. Pöhlker et al., 2018). Thus, $P_{BT}$ represents a measure for the extent of rain-related aerosol scavenging – particularly of long-range transport aerosols – during air mass transport towards ATTO. A pronounced maximum in $P_{BT}$ and the related scavenging is observed for the months April and May, which coincides with the minimum in

aerosol parameters (i.e., $M_{BCe}$ and $N_{CN,10}$) (see also Moran-Zuloaga et al., 2017).

Figure 2d shows the seasonal cycles in cloud fraction and cloud top temperature within the ROI$_{ATTO}$. Both are predominantly influenced by the position of the intertropical convergence zone (ITCZ) with its belt of extended deep convective cloud systems and strong precipitation (e.g., Moran-Zuloaga et al., 2017). According to Fig. 2d, the densest cloud cover and deepest convection (represented by lowest cloud top temperature) occurs

upon northwards passage of the ITCZ in the middle of the wet season (i.e., March and April) as well as upon

southwards passage of the ITCZ at the end of the dry season (around November). The maximum in deep convection in March and April – expectedly – corresponds to the peak in $P_{TRMM}$. A global picture of the spatiotemporal trends in cloud microphysical properties can be found in King et al. (2013).

Figure 2e presents the satellite-derived effective radius, $r_e$, of liquid cloud droplets that links the seasonality in aerosol and cloud properties. The $r_e$ data have been filtered by cloud top temperature to discriminate the different $r_e$ growth states at different heights of the convective clouds. It is well established that increased CCN loads entail an influence on cloud properties, which typically results in a corresponding decrease in droplet diameter (e.g., Freud et al., 2008; Stevens and Feingold, 2009). Figure 2e underlines this behavior by means of a clear seasonality in $r_e$ with its maximum during the clean wet season (i.e., April and May) and its minimum during the polluted dry season (i.e., August to November). A detailed understanding of the impact of the CCN loading on the cloud microphysical properties, however, is the subject of ongoing studies at ATTO and further locations worldwide. In essence, Fig. 2 provides a coherent picture of the aerosol, cloud, and precipitation seasonality as well as their corresponding linkages. The following sections will zoom into this overall picture by presenting detailed aerosol and CCN data from characteristic wet and dry season conditions of the year 2014.

### 3.2 Aerosol and CCN time series for representative wet season conditions

During the Amazonian wet season (February to May), the influence of local and regional anthropogenic pollution (i.e., from biomass and fossil fuel burning) decreases to a minimum and simultaneous strong precipitation causes efficient aerosol scavenging (see Fig. 2). The combination of both effects results in a comparatively clean state of the Amazonian atmosphere (Martin et al., 2010b; Andreae et al., 2015). During this time of the year, biogenic aerosols from the surrounding rain forest ecosystem, such as secondary organic aerosol (SOA) from the oxidation of biogenic volatile organic compounds (BVOC) as well as primary biological aerosol particles (PBAP), prevail (Pöschl et al., 2010; Huffman et al., 2012; Yañez-Serrano et al., 2015). However, the regionally and biogenically dominated background state of the atmosphere is frequently perturbed by the episodic advection of long-range transport (LRT) aerosols from Africa in air masses that bypass the major rain fields and, therefore, 'survive' the intense scavenging (Moran-Zuloaga et al., 2017). The frequent intrusion of LRT aerosols is a characteristic feature during the Amazonian wet season and represents a strong and important influence on the rain forest ecosystem (e.g., Chen et al., 2009; Bristow et al., 2010; Baars et al., 2011; Abouchami et al., 2013; Yu et al., 2015; Rizzolo et al., 2016). These LRT plumes mostly comprise a complex mixture of Saharan dust, African biomass burning smoke, and marine aerosols from the transatlantic air passage (e.g., Talbot et al., 1990; Swap et al., 1992; Gläser et al., 2015).

The 2-months period in Fig. 3 can be regarded as representative for typical wet season conditions in the ATTO region as it includes both scenarios: periods with the prevalence of the local (biogenic) background aerosol and episodes under the influence of LRT plumes, as well as intermediate states. In general, the wet season 2014 showed average hydrological conditions without significant precipitation anomalies within the $ROI_{ATTO}$, which is in stark contrast to 2015/16 with its pronounced El Niño influence and an associated negative precipitation anomaly (see C. Pöhlker et al., 2018; Saturno et al., 2017a). Furthermore, the pollution tracers, $N_{CN,10}$, $M_{BCe}$, and $c_{CO}$, showed comparatively low values with concentrations around $N_{CN,10} = 330 \pm 130$ cm$^{-3}$, $M_{BCe} = 0.03 \pm 0.05$ µg m$^{-3}$, and $c_{CO} = 110 \pm 10$ ppb (given as mean $\pm$ 1 std for entire time period in Fig. 3) in agreement with previous studies (e.g., Andreae et al., 2012 and 2015; Artaxo et al., 2013). In terms of atmospheric circulation, the first half of the 2-month time frame was dominated by backward trajectory arrivals from the northeast (blue

and green BT clusters, see Fig. 3a and Fig. S1), whereas during the second half the dominant wind direction shifted towards easterly directions (orange and red BT clusters, see Fig. 3a and Fig. S1) (compare Andreae et al., 2015; Moran-Zuloaga et al., 2017). This gradual swing of the dominant wind direction from NH to SH relates to the northwards movement of the ITCZ.

In Fig. 3, the orange background shading emphasizes several episodes with detectable LRT influence, according to Moran-Zuloaga et al. (2017). A comparatively strong LRT plume at ATTO – labeled as *LRT* case study in Fig. 3 – is the subject of a detailed CCN analysis in section 3.5. The interested reader will find further in-depth analysis of this particularly LRT event in Moran-Zuloaga et al., 2017 (where it is discussed as event "2014_7"). Andreae et al. (2012) argued that the "atmospheric state and processes in the Amazon Basin cannot be understood without the consideration of pollutant inputs by long-range transport". This is evidently true for the major LRT plumes in Fig. 3 with $M_{BCe}$ reaching up to 0.3 μg m$^{-3}$ as well as associated increases in $N_{acc}$. However, also a closer analysis of the extended and relatively clean period from 14 April until 31 May 2014 reveals that during ~85 % of the time detectable amounts of background pollution were present (i.e., $M_{BCe}$ exceeding $M_{BCe}$*, see Sect. 2.7). Although (highly) diluted, the advected aerosols can impact the CCN population, as discussed in Sect. 3.4. Only when pollution levels indeed dropped below detection limits and the conditions satisfy our rather strict $PR_{BC \cap CO}$ filter, empirically pristine aerosol conditions prevail, which are emphasized by a blue shading in Fig. 3. A statistical overview of the relative fraction of *PR* episodes for the years 2012 to 2016 is shown in Fig. 1. It shows that *PR* conditions typically occur from March to May with their highest abundance around late April and early May when weekly frequencies of occurrence reach up to ~20 %, according to the strict $PR_{BC \cap CO}$ filter, or even higher. Note that the *PR* episodes in 2014 mostly occurred in April and May (compare Fig. 3 and Fig. S4), which is in good agreement with the multi-year observations.

The following picture emerges for the CCN parameters: The time series of the $\kappa(S,D_a)$ size distributions in Fig. 3d clearly illustrates the different $\kappa(S,D_a)$ of Aitken and accumulation modes as discussed in our part 1 study (M. Pöhlker et al., 2016). Overall, $\kappa(S,D_a)$ shows a clear variability for low $S$, representing the accumulation mode, and rather stable values for higher $S$, representing the Aitken mode (see Fig. 3h). The occurrence of the LRT plumes stands out clearly by causing a significant enhancement of $\kappa(S,D_a)$ with $\kappa_{Acc}$ reaching 0.4 and $\kappa_{Ait}$ reaching 0.25 (see Fig. 3d and h). Moreover, the LRT events are also associated with increased $N_{CCN}(S)$ and $N_{CCN}(S)/N_{CN}$ values (Fig. 3g and i). Note that the occurrence of the LRT events precisely coincides with the minima in $P_{BT}$, and, thus, 'windows' in aerosol scavenging (see Moran-Zuloaga et al., 2017).

For the extended and comparatively clean period from 14 April until end of May, the SMPS contour plot (Fig. 3c) reveals equally strong Aiken and accumulation modes as well as a 'patchy' appearance, due to frequent rainfall causing (local) aerosol scavenging. As outlined in part 1, the aerosol abundance in the particle size range >40 nm predominantly defined the measured CCN population (M. Pöhlker et al., 2016). Accordingly, the $N_{CCN}(S)$ and $N_{CCN}(S)/N_{CN}$ time series directly track the SMPS-derived patchy pattern. The low $S$ levels – i.e., $N_{CCN}(0.11\ \%)$ – which do not activate Aitken mode particles, closely followed the accumulation mode concentration, $N_{acc}$, time series, whereas the higher $S$ levels (i.e., $N_{CCN}(1.10\ \%)$), which also activated particles in the Aitken mode, closely tracked $N_{CN,10}$ (= $N_{Ait} + N_{acc}$). Two of the subsequent case studies will focus in more detail on the specific CCN properties of the *PR* (Sect. 3.4) and *LRT* (Sect. 3.5) conditions (see also Table 1).

### 3.3 Aerosol and CCN time series for representative dry season conditions

During the dry season (August to November), the central Amazon is under continuous influence of pronounced anthropogenic pollution. The predominant type is biomass burning smoke from deforestation fires, which coined the term "biomass burning season" (Freud et al., 2008). In addition, various urban and industrial emission sources in eastern and southern Brazil as well as southern Africa may also contribute by hard-to-define quantities (e.g., Andreae et al., 1994; Saturno et al., 2017a). The minimum in precipitation rates and, thus, in aerosol scavenging fosters the distribution of pollution aerosols over large areas by extending their atmospheric lifetime (see Fig. 2).

Figure 4 represents the dry season counterpart of Fig. 3 and shows the corresponding time series for a characteristic dry season period from 01 August until 30 September 2014. The BT clusters in Fig. 4a show that easterly and southeasterly BTs prevailed, since the ITCZ was located north of the ATTO site. The BT clusters, which are most characteristic for the dry season, approached the Amazon River delta from the southern Atlantic and then follow the river in western direction towards ATTO (red and orange clusters, see Fig. S1). These 'dry season river BTs' are the subject of a more detailed discussion in the case study in Sect. 3.7. For certain episodes, the wind swings further to southeasterly directions and brings air masses from inland directions (black, brown, and grey BT clusters, see Fig. S1). These 'dry season inland BTs' are further discussed in the case study in Sect. 3.6.

In contrast to the wet season, the accumulation mode clearly dominates the size distribution, which explains the increased CCN efficiencies, particularly at low $S$ (Fig. 4c and i). The frequent 'pulses' in the accumulation mode concentration can be attributed to (aged) biomass burning plumes, which impact the ATTO site episodically, typically for few days (see Freitas et al., 2005). The $M_{BCe}$, $N_{CN,10}$, and $c_{CO}$ levels are typical for dry season conditions in the northeast Amazon Basin with $M_{BCe} = 0.55 \pm 0.35$ µg m$^{-3}$, $N_{CN,10} = 1520 \pm 780$ cm$^{-3}$ and $c_{CO} = 140 \pm 50$ ppb (given as mean $\pm$ 1 std for the time period in Fig. 4) (e.g., Rissler et al., 2004; Andreae et al., 2012, 2015; Artaxo et al., 2013; Saturno et al., 2017a).

The most obvious event in Fig. 4 is the advection of a strong biomass burning (BB) plume from 17 to 23 August 2014, the highest pollution levels that were observed during the entire CCN measurement period. This event can be recognized by means of strong pulses in $N_{Acc}$, $M_{BCe}$, and $c_{CO}$ as well as a dip in $\kappa(S,D_a)$. The ACSM-derived organic-to-sulfate ratio confirms that the biomass burning pulse comprised a predominantly organic aerosol. In general, the measured $\kappa(S,D_a)$ levels respond inversely to the organic-to-sulfate ratio, confirming previous studies, which stated that organic matter and sulfate constituents are mostly defining the aerosol's hygroscopicity in the Amazon (Roberts et al., 2002; Gunthe et al., 2009). The major biomass burning plume in Aug 2014 is the subject of the detailed case study *BB* in Sect. 3.6.

Beside this major biomass burning plume, several shorter pulses, which were supposedly also caused by upwind fires, were observed throughout the dry season period and their frequency of occurrence increased towards the end (i.e., after 12 September). Phenomenologically, the major and minor biomass burning plumes were similar as they exhibit peaks in $N_{CN}$, $M_{BCe}$, and $c_{CO}$ and simultaneous dips in $\kappa(S,D_a)$ and the organic-to-sulfate ratio. The second half of September comprised conditions with comparatively high sulfate concentrations and a sequence of short biomass burning plumes. This period is influenced by a mixture of different (pollution) aerosol populations from African long-range transport and regional South American sources. A detailed description of the case study *MPOL* can be found in Sect. 3.7.

Similar to the wet season, different $\kappa(S,D_a)$ levels for the Aitken and accumulation modes as well as comparably low $\kappa(S,D_a)$ variability over time ($\kappa_{Ait} = 0.14 \pm 0.03$ vs. $\kappa_{Acc} = 0.23 \pm 0.04$, covering the entire time period

in Fig. 4d and h) were observed. The *MAF(S)* values tend to reach unity, except for *MAF*(0.11 %) under the influence of biomass burning smoke (i.e., for the smaller and major smoke plumes).

**3.4   Case study *PR* on empirically pristine rain forest aerosols**

Aerosol-cloud interactions remain to be the largest uncertainty in global climate models and a better understanding of a preindustrial atmospheric state is essential to reduce this uncertainty (Carslaw et al., 2013; Seinfeld et al., 2016). As outlined in Sect. 2.7, the Amazonian wet season provides the rare chance to analyze episodes of very clean continental conditions, which are our best approximation of a pristine rain forest atmosphere. This case study extracts the characteristic aerosol and CCN properties during the identified *PR* periods. Figure 5 zooms into three selected episodes – called *PR1*, *PR2*, and *PR3* as highlighted in Fig. 3 – and combines meteorological, aerosol, and CCN time series for a detailed analysis.

The meteorological parameters in Fig. 5a, b, and c illustrate typical wet season conditions: (i) a rather high degree of cloudiness, which can be seen by means of the strong cloud-related dimming of the incoming shortwave radiation, $SW_{in}$, (ii) frequent local ($P_{ATTO}$) and regional ($P_{TRMM}$) rain events, (iii) a comparatively stable northeasterly wind direction, which is consistent with the BT analysis in Fig. 3, (iv) a rather constant wind speed, $U$, for most of the time, which was getting more vigorous during rain events, (v) high relative humidity (RH) conditions, being inversely related to $SW_{in}$ and reaching saturation during the nights, and (vi) a characteristic time series of the equivalent potential temperature, $\theta_e$, which tracked the daily onset of solar heating (see simultaneous increase of $SW_{in}$ and $\theta_e$) and further provides valuable information on vertical atmospheric mixing, particularly in connection with rain events. Specifically, sudden drops in $\theta_e$ indicate a downward transport of air masses from upper tropospheric layers, which occurred frequently with the onset of strong rain (for more details see Wang et al., 2016).

The corresponding aerosol variability is shown as an aerosol number size distribution (d$N$/dlog$D$) contour plot in Fig. 5d. Under *PR* conditions we found a dominant Aitken mode and a comparatively weak accumulation mode, as reported previously (e.g., Andreae et al., 2015; M. Pöhlker et al., 2016). Moreover, the Aitken and accumulation modes reveal a patchy and discontinuous abundance with rather sudden concentration increases and drops. These observations can be explained by a combination of different effects and processes: First, the strong and persistent (mostly combustion-related) sources of accumulation mode aerosol particles, which are responsible for the continuous and dominant accumulation mode in the dry season, were absent.

Second, the frequent rain events acted as an efficient aerosol removal mechanism via aerosol rain-out (i.e., particle activation into cloud/rain droplets) and wash-out (i.e., particle collection by falling droplets below clouds). The wash-out efficiency is slightly higher (~ factor 1.5) for Aitken than accumulation mode particles (Wang et al., 2010; Zikova and Zdimal, 2016). In contrast, the rain-out efficiency, corresponding to CCN activation, is typically much higher for accumulation than Aitken mode particles (M. Pöhlker et al., 2016). Accordingly, the rain pulses strongly modulated the aerosol's abundance via sudden and efficient deposition, which is visible in Fig. 5d as characteristic 'notches' in the aerosol contour plot. The notches represent the (removed) part of the aerosol size fraction that was activated as CCN into cloud droplets. Illustrative examples can be found during days with strong rain showers, such as 27 April, 05 May, and 06 May. Note that the 'depth of the notches', corresponding to the smallest activated particles, could in principle be used to estimate the *S* levels during the corresponding events (Krüger et al., 2014). Further note in this context that besides depletion of the accumulation relative to the Aitken mode, aerosol activation and cloud processing are known to also foster the opposite effect:

The growth of Aitken mode particles into the accumulation mode via formation of aqueous phase reaction products (i.e., sulfate and aqueous phase SOA) in the cloud droplets, followed by droplet re-evaporation and deposition of the newly formed compounds onto the particles (e.g., Ervens, 2015; Farmer et al., 2015). Possibly, during 07 May, cloud processing was responsible for the formation of a rather strong accumulation mode from an existing Aitken mode population.

As a third process, the Aitken mode population in the rain forest boundary layer (BL) was frequently replenished by pulse-like appearance of particles with diameters <50 nm. These events are supposed to be convection-related downward transport of air masses from an upper troposphere (UT) particle pool and the subsequent growth of the injected fine particles (Wang et al., 2016; Andreae et al., 2018). Remarkably, the combination of all these effects results in a comparatively constant total particle abundance, $N_{CN}$, across the rain showers, due to compensating effects of accumulation mode particle losses and simultaneous increases in Aitken mode abundance (see details in Wang et al., 2016). The convection-related downward transport of fine UT particles and their subsequent growth – presumably by the condensation of low-volatility vapors – is a characteristic feature of the Amazonian wet season. At least three pronounced examples for this process are included in the time frame of Fig. 5 (i.e., 27/28 April, 04/05 May, and 16/17 May). After their injection into the forests BL, the fine particles (initial diameters between 20 and 50 nm for the events in Fig. 5) reveal a 'banana-like' growth into the Aitken mode size range (~70 nm) in the course of about 12-24 h. For the events in Fig. 5, we calculated an initial, and thus maximum, growth rate of 0.6 to 6 nm h$^{-1}$, which agrees well with the 1 to 6 nm h$^{-1}$ reported by Kulmala et al. (2004) for tropical regions as well as the reported 5 nm h$^{-1}$ in Zhou et al. (2002) for an Amazonian site. Note that these 'Amazonian bananas' differ from the classical new particle formation (NPF) events that have been reported for various continental sites (i.e., in Northern temperate regions) (Kulmala et al., 2004), since the number concentrations are lower by orders of magnitude and their nucleation and initial growth does not occur in the BL, but in the UT (Ekman et al., 2008; Engstrom et al., 2008; Pöschl et al., 2010; Andreae et al., 2018). Accordingly, the UT particle population that is frequently injected into the BL is already aged to a certain extent and, thus, presumably reflects chemical processes different from the atmospheric chemistry in the BL. The physicochemical details of the UT nucleation and growth are still largely unknown and subject of ongoing research (e.g., Andreae et al., 2018).

The CCN properties during *PR* conditions are represented by time series of $\kappa(S,D_a)$ size distributions (Fig. 5d) and $N_{CCN}(S)$ for two selected $S$ (Fig. 5e). The temporal pattern of the $\kappa(S,D_a)$ size distributions, which provides indications of the aerosol particles' chemical composition, reflects the pattern of the underlying d$N$/dlog$D$ contour plot. Consistent with our observations in part 1 (M. Pöhlker et al., 2016), the accumulation mode reveals higher $\kappa(S,D_a)$ levels than the Aitken mode, likely due to chemical aging through cloud processing and a related increase in hygroscopicity (Farmer et al., 2015). The lowest $\kappa(S,D_a)$ levels were observed for the 'Amazonian bananas' (see Fig. 5d). Both, the accumulation and Aitken mode $\kappa(S,D_a)$ levels show a variability that tracks the Aitken and accumulation mode abundance. Note that $N_{CCN}(0.5\,\%)$ and particularly $N_{CCN}(0.2\,\%)$ show pronounced increases during periods with increased $M_{BCe}$ levels (e.g., 25 April, 05 May, and 17 May). This emphasizes the remarkable impact of diluted pollution on the CCN population in an aerosol-limited regime according to Reutter et al. (2009).

Figure 6a and b summarize the average aerosol and CCN key properties under *PR* conditions.[1] Figure 6a shows the characteristic average $N_{CN}(D)$ size distribution with a pronounced bimodal appearance, comprising a dominant Aitken mode ($D_{Ait} \approx 70$ nm, $N_{Ait} \approx 160$ cm$^{-3}$) and a comparatively weak accumulation mode ($D_{acc} \approx 160$ nm, $N_{acc} \approx 90$ cm$^{-3}$) (see Table 2). This bimodal shape is typical for clean Amazonian conditions as reported previously (e.g., Gunthe et al., 2009) and further resembles aerosol size distributions under marine background conditions (e.g., Atwood et al., 2017). The corresponding $N_{CCN}(S,D)$ size distributions for all $S$ levels show that for S < 0.3 % mostly accumulation mode particles were activated, whereas for S > 0.3 % the Aitken mode particles also acted as CCN. Furthermore, Fig. 6a shows the average $\kappa(S,D_a)$ size dependence with a characteristic step-wise increase of $\kappa(S,D_a)$ towards larger $D$. The Aitken mode $\kappa(S,D_a)$ levels are rather low and sharply defined (mean ± std: 0.12 ± 0.01), whereas the accumulation mode $\kappa(S,D_a)$ levels are slightly higher (0.18 ± 0.02). The results suggest that the Aitken mode particles, which are frequently injected into the BL via downward transport from the UT, are mostly comprised organic matter. This observation agrees well with recent results showing that "the UT particles consist predominantly of organic material, with minor amounts of nitrate and very small fractions of sulfate" (Andreae et al., 2018). The hygroscopicity of organic material, $\kappa_{org}$, is typically assumed as ~0.10, however, $\kappa_{org}$ can vary substantially (close to 0 up to 0.3) as a function of the organic material and its oxygen-to-carbon (O:C) ratio (Jimenez et al., 2009; Thalman et al., 2017). The $\kappa(S,D_a)$ levels of the accumulation mode similarly indicate the presence of predominantly organic particles, however with somewhat more inorganic constituents than in the Aitken mode. This is consistent with the corresponding ACSM results in Table 3, underlining that organic matter accounts for most of the mass (90 %), whereas nitrate (4 %) and sulfate (6 %) add only small contributions. Note that $M_{NH4}$, $M_{Cl}$, and $M_{BCe}$ were below detection limit for *PR* conditions and were omitted in the calculation of the mass fractions accordingly (see Sect. 2.1).[2] A predicted average hygroscopicity parameter, $\kappa_p$, of 0.16 ± 0.01 was calculated based on the ACSM results – excluding $M_{NH4}$, $M_{Cl}$, and $M_{BCe}$ – and agrees with the measured value of $\kappa(0.11$ %$) = 0.18 \pm 0.05$ (Table 3).

Figure 6b displays the CCN efficiency spectrum for *PR* conditions, which can be regarded as CCN signature of the corresponding aerosol population (for details refer to the companion part 1 paper). The pronounced bimodal particle size distribution with its characteristic Hoppel minimum and the step-wise increase of $\kappa(S,D_a)$ (see Fig. 6a) result in a weak plateau at about $S = 0.4$ %, which required to apply a double-erf fit. For comparison, we also applied a single-erf fit. Expectedly, the double-erf fit is the better representation of the experimental data, although the single-erf fit also covers the data reasonably well, since the plateau is not particularly pronounced. However, a closer look reveals differences between the single- vs. double-erf fits for small and large $S$. For very low S (<0.1 %), the double-erf fit indicates that the *PR* aerosol particles start acting as CCN only above about $S = 0.06$ %, which can probably be explained by the absence of suitable CCN in the size range of several hundred nanometers, which are indeed comparatively sparse under *PR* conditions according to Fig. 5. However, this size range was not covered directly by our CCN measurements, making it hard to draw conclusions. Since the

---

[1] The aerosol and CCN key properties are similar for all *PR* filters (i.e., $PR_{BC}$, $PR_{CO}$, $PR_{BC \cup CO}$, $PR_{BC \cap CO}$, see Sect. 2.7). For completeness, the $N_{CN}(D)$ and $\kappa(S,D_a)$ size distributions for all four *PR* filters are summarized in Fig. S5. For comparison, Fig. S6 shows $N_{CN}(D)$ size distributions after a classification by means of a statistical analysis of the total aerosol concentrations, which resemble the *PR* filter-based results. Furthermore, Fig. S7 show the corresponding $N_{CN}(D)$ and $\kappa(S,D_a)$ size distributions filtered for the comparatively clean BT clusters NE1, NE2, and NE3.

[2] $M_{BCe}$ was per definition below detection limit for *PR* conditions.

double-erf fit describes the data more accurately than the single-erf fit, its extrapolation for S > 1.1 % is likely more accurate and suggests that 'full' activation (~90 %) is reached at $S \approx 1.5$ %.

The CCN efficiency spectra represent a tool to visualize characteristic differences in the behavior of certain (contrasting) aerosol populations in cloud formation. Of particular relevance is the slope of the CCN efficiency spectra, $\mathrm{d}(N_{CCN}(S)/N_{CN,10})/\mathrm{d}S$, as the sensitivity of the activated CCN fraction of a given aerosol population within a given $S$ range to changes in supersaturation, $\Delta S$. Accordingly, high $\mathrm{d}(N_{CCN}(S)/N_{CN,10})/\mathrm{d}S$ slopes indicate a regime, in which already subtle a $\Delta S$ has relatively strong effects on the $N_{CCN}(S)$ and, thus, $N_d$ concentrations (assuming constant $N_{CN,10}$), whereas low $\mathrm{d}(N_{CCN}(S)/N_{CN,10})/\mathrm{d}S$ values indicate a regime that is characterized by more stable $N_{CCN}(S)$ and $N_d$ concentrations, even upon large $\Delta S$.

The CCN efficiency spectra can be linked to a concept introduced by Reutter et al. (2009), which classifies the sensitivity of $N_d$ concentrations in response to changes in $N_{CN}$ and $w_b$. This results in three distinct regimes: (i) an *aerosol-limited* regime, which is characterized by high $S_{max}$ (>0.5 %), a high $w_b/N_{CN}$ ratio, high activated fractions $N_d/N_{CN}$, and a linear relationship between $N_{CN}$ and $N_d$; (ii) an *updraft-limited* regime, which is characterized by rather low $S_{max}$ (<0.2 %), a low $w_b/N_{CN}$ ratio, low activated fractions $N_d/N_{CN}$, and a linear relationship between $w_b$ and $N_d$; and (iii) a *transitional* regime with intermediate states and non-linear dependencies of $N_d$ on $N_{CN}$ and $w_b$. Note that the concept of Reutter et al. has been developed for the conditions of pyro-convective clouds and, thus, represents an extreme case, which has to be kept in mind in the subsequent comparison with the CCN efficiency spectra.[3] Pöschl et al. (2010) applied the concept of Reutter et al. (2009) to wet season rain forest conditions. Based on Reutter et al. and Pöschl et al., *PR* conditions with the associated low aerosol concentrations ($N_{CN,10} \approx 260$ cm$^{-3}$, Table 4) can be characterized as aerosol-limited, due to the overall low abundance of potential CCN. However, the CCN efficiency spectrum in Fig. 6b suggests that the *PR* aerosol population also has a certain sensitivity to $w_b$, since changes in updraft velocity $\Delta w_b$ – and the associated $\Delta S$ (even for S > 1 %) – strongly modulate the activated fraction $N_{CCN}(S)/N_{CN,10}$. In this sense, the *PR* CCN efficiency spectrum has a very characteristic shape and differs clearly from the other conditions, as it keeps increasing over a wide $S$ range with a somewhat larger slope for S < 0.3 % and a slightly decreasing slope for S > 0.3 %.[4] Accordingly, the *PR* case appears to be aerosol- *and* updraft-sensitive, which makes the corresponding aerosol-cloud interaction highly dynamic. For more quantitative insights, modelling runs according to Reutter et al. based on the characteristic *PR* bimodal $N_{CN}(D)$ size distribution in Fig. 6a as well as adjusted $N_{CN}$ and $w_b$ ranges are required.

### 3.5 Case study *LRT* on Saharan dust, African smoke and Atlantic marine aerosols

The African continent is of significant importance for the Amazonian atmospheric composition as it represents a major source of desert dust and pollution aerosols (e.g., Swap et al., 1992; Andreae et al., 1994; Yu et al., 2015; Rizzolo et al., 2016). A systematic overview of the properties and relevance of *LRT* plume arrivals in the ATTO region during the Amazonian wet season can be found in Moran-Zuloaga et al. (2017). Furthermore, a general

---

[3] The cloud parcel modelling by Reutter et al. (2009) was conducted for the specific case of pyro-convective conditions, developing over strong fires. Accordingly, a typical (monomodal) biomass burning aerosol size distribution with an accumulation mode maximum at 120 nm was used. In this sense, the study is bound to rather specific and extreme conditions. However, Reutter et al. state that the "key features of the three regimes of CCN activation (…) are not specific for young biomass burning aerosols and pyro-convective conditions but likely to apply also for other types of aerosols and meteorological conditions".

[4] For completeness, the CCN efficiency spectra for all four *PR* filters according Sect. 2.7 (i.e., $PR_{BC}$, $PR_{CO}$, $PR_{BC \cup CO}$, $PR_{BC \cap CO}$) are shown in the comparison Figure S8.

characterization of the CCN population's response to *LRT* conditions during the wet season can be found in our part 1 paper (M. Pöhlker et al., 2016). Based on these previous studies, the present paper analyzes the characteristic impact of *LRT* plumes on the wet season CCN population in detail.

The characteristic impact of LRT plumes on the aerosol and CCN data can already be seen in Fig. 3. For a detailed analysis, the strongest *LRT* episode in Fig. 3 has been chosen and is represented in Fig. 7 by selected meteorological, aerosol, and CCN time series. Note that the advected dust plumes typically cause clear increases in both, the coarse mode, which plays a secondary role here, and the accumulation mode, which explains their relevance for the CCN variability (Moran-Zuloaga et al., 2017). The *LRT* influence is visible in Fig. 7 by means of increases in $M_{BCe}$ (due to the *LRT* plume's smoke component), the accumulation mode abundance in the d$N$/dlog$D$ contour plot, the concentrations $N_{CN}$ and $N_{CCN}(S)$, as well as in $\kappa(S,D_a)$.

The plume's influence on the ATTO site lasted for about four days with its onset on 09 April ~12:00 and its end on 13 April ~12:00, which is in good agreement with the corresponding remote sensing data in Moran-Zuloaga et al. (2017). During these four days, meteorology determined the variability of the *LRT* aerosol in the rain forest BL via air mass advection (i.e., wind speed and direction), rain-related scavenging (i.e., $P_{ATTO}$ and $P_{TRMM}$), and convective mixing (represented by SW$_{in}$ as a proxy). Throughout the *LRT* period, the dominant wind direction was mostly NE to E, which is characteristic for wet season conditions (compare Fig. 3). Note that the BTs arriving from the NE are particularly prone to bring dust-laden air into the ATTO region (Moran-Zuloaga et al., 2017). Furthermore, only few (major) rain events and related aerosol scavenging occurred during this time, which is a further prerequisite for efficient *LRT* transport. The arrival of the *LRT* plume during afternoon hours of 09 April occurred via convective downward mixing of the aerosol into the forest BL (see simultaneous increases in SW$_{in}$, $\theta_e$, and the accumulation mode in d$N$/dlog$D$) and its influence lasted until 13 April, when the concentration time series decreased gradually. Note that the advected *LRT* plumes are transported into the ATTO region as compact and stratified aerosol layers up to altitudes of 2-3 km (Moran-Zuloaga et al., 2017). Aerosol concentrations were highest during the afternoon hours on 09, 10 and 11 April, when convection (see SW$_{in}$) is most efficient in mixing the aerosols downward into the BL. Furthermore, a sudden rain-related air mass change, lasting for about 12 h (see wind direction and $\theta_e$ changes) in the end of 09 April/beginning of 10 April, was associated with a simultaneous drop in *LRT* aerosol concentrations. The CCN population responded noticeably to the injection of the LRT event, which can be seen in $N_{CCN}(S)$, $N_{CCN}(S)/N_{CN,10}$, and $\kappa(S,D_a)$ in Fig. 3. Figure 7 displays the $N_{CCN}(0.2\ \%)$ and $N_{CCN}(0.5\ \%)$ time series, which show a two- to threefold increase upon *LR*T aerosol intrusion. Moreover, the $\kappa(S,D_a)$ data can be used as an indirect measure for the aerosol chemical composition, reflecting the average ratio of organic vs. inorganic constituents in the particles. In Fig. 7d, the occurrence of elevated $\kappa(S,D_a)$ levels coincides with the variability in d$N$/dlog$D$. For this particular event, $\kappa_{Acc}$ increased from ~0.3 to ~0.4, whereas $\kappa_{Ait}$ increased from ~0.15 to ~0.25.

Figure 6c summarizes the characteristic $N_{CN}(D)$ and $N_{CCN}(S,D)$ size distributions as well as the size dependence of $\kappa(S,D_a)$ during *LRT* influence. In general, the comparison of Fig. 6a and c contrasts the characteristic *LRT* vs. *PR* conditions and emphasizes that the intrusion of African *LRT* plumes into the wet season BL has significant influence on the aerosol and CCN populations. Specifically, the accumulation mode under *LRT* conditions ($D_{acc} \approx 180$ nm, $N_{acc} \approx 300$ cm$^{-3}$) is about 5 times enhanced in comparison to the *PR* episodes. The Aitken mode ($D_{Ait} \approx 80$ nm, $N_{Ait} \approx 120$ cm$^{-3}$) shows comparable absolute strength as under *PR* conditions (see Table 2), but appears in the *LRT* $N_{CN}(D)$ distribution only as a shoulder. The $N_{CCN}(S,D)$ size distributions show that supersaturations S ≤ 0.3 % activate particles mostly in the accumulation mode range, whereas S > 0.3 % starts activating

the Aitken mode population. The $\kappa(S,D_a)$ size distribution shows $\kappa_{Acc}$ reaching rather high levels up to 0.40 ($\kappa_{Acc} = 0.35 \pm 0.04$) and $\kappa_{Ait}$ with values up to 0.2 ($\kappa_{Ait} = 0.18 \pm 0.02$). In comparison to the *PR* conditions, $\kappa_{Acc}$ was clearly enhanced due to the presence of aged, internally mixed, and salt-rich particles (see discussion in subsequent section). In contrast, the $\kappa_{Ait}$ levels were only slightly higher than during the *PR* periods, probably due to a broader accumulation mode base, which reached into the Aitken mode range. Note that the relative increase in $\kappa(S,D_a)$ for the contrasting LRT and *PR* conditions is large enough to exert an influence on the CCN population, which is an exceptional case throughout the Amazonian seasons, since the $\kappa(S,D_a)$ levels for most of the year vary within a rather narrow range (0.1 to 0.2) (see part 1 paper).

The African *LRT* plumes that frequently impact the Amazon Basin during the wet season comprise a complex mixture of different aerosols components, including (i) a fraction of Saharan dust (mostly >1 μm), (ii) biomass burning aerosols from fires in West Africa (mostly in the accumulation mode), and (iii) marine aerosols from the plume's transatlantic passage (in coarse and accumulation modes) (Andreae et al., 1986; Talbot et al., 1990; Swap et al., 1992; Weinzierl et al., 2017). Note that the African fires in West Africa are to a large extent agriculture-related and, thus, man-made (e.g., Barbosa et al., 1999; Capes et al., 2008). Fresh soot and uncoated dust particles typically have low hygroscopicities, whereas sea salt particles (i.e., NaCl and sulfates) represent comparatively efficient CCN. The LRT plumes make a rather long atmospheric journey – about 10 days according to Gläser et al. (2015) – which is associated with strong atmospheric processing and typically results in complex internally mixed particles (see Andreae et al., 1986). The resulting mixtures of dust and salt as well as coated soot particles, as commonly found in the dust plumes, readily act as CCN at realistic $S$ levels (<1 %) (Andreae and Rosenfeld, 2008).

Table 3 summarizes the ACSM results on aerosol chemical composition during *LRT* influence.[5] Note that the ACSM only measures the non-refractory fraction of the particle mass and, thus, does not detect a large fractions of the dust and salt constituents. During *LRT* conditions and with respect to the non-refractory part, organic mass dominated the particle composition (67 %), with minor nitrate contributions (3 %) and larger fractions of ammonium (11 %) and sulfate (9 %). Furthermore, a slight sea salt-related increase in detected chloride mass (1 %) relative to the *PR* state was observed. Based on the ACSM results, a predicted $\kappa_p$, of $0.24 \pm 0.01$ was obtained, which agrees reasonably well with the measured hygroscopicity level $\kappa(0.11\ \%) = 0.35 \pm 0.04$. Accordingly, the observed elevated fractions of sulfate and ammonium were responsible for part of the increase in $\kappa(S,D_a)$. The remaining difference between $\kappa_p$ and $\kappa(0.11\ \%)$ can likely be explained by further refractory inorganics that were not covered by the ACSM. As an illustration of the complex mixing state of the African *LRT* aerosol population, Fig. S10 and S11 provide selected SEM-EDX data from a characteristic *LRT* aerosol sample. Note that these results merely serve as a qualitative example here, whereas a quantitative microspectroscopic analysis of *LRT* samples is subject of ongoing work. The observed particles' morphology underlines the influence of atmospheric processing and shows a high degree of internal mixing. Chemically, the EDX maps show dust particles, sulfate salts, as well as a rather large abundance of NaCl particles.

Figure 6d displays the characteristic CCN efficiency spectrum for the *LRT* conditions, which – expectedly – shows a rather steep increase at low $S$ due to two effects pointing in the same direction: an enhanced accumulation mode and the presence of rather 'good' CCN with comparatively high $\kappa(S,D_a)$ levels. This large initial slope

---

[5] ACSM data was available from Aug 2014 to Sep 2016. Accordingly, the ACSM time frame does not cover the highlighted *LRT* event, which is discussed in detail in this work. Instead, we analyzed the ACSM data for 12 comparable *LRT* episodes in 2015 and 2016 according to Moran-Zuloaga et al. (2017) (see also Table 1).

corresponds to a very small value for the characteristic variable $S_1 = 0.09$ %, which shows that at comparatively low $S$ (lowest $S_1$ among all analyzed conditions) already 50 % of the aerosol population were activated as CCN (Table 4). At $S$ levels around 0.3 % already ~80 % of the aerosols acted as CCN, 'full' activation (90%) is already reached for $S \approx 0.5$ %. At higher $S$ (i.e., > 0.9 %) the efficiency spectrum approaches unity. This is in stark contrast to the $PR$ CCN efficiency spectrum in Fig. 6b, where 50 % of the particles were activated at significantly higher $S$ around 0.5 %, whereas 'full' activation (90%) was hardly reached at high $S$ around 1.7 %. Thus, the $LRT$ episodes show a pronounced sensitivity to $\Delta S$ only in the low $S$ regime, in contrast to the near pristine case with a sensitivity to $\Delta S$ spanning across a wide $S$ range. With the characteristic shape of the CCN efficiency spectrum (i.e., rather strong accumulation mode) and the rather low aerosol concentrations ($N_{CN,10} \approx 440$ cm$^{-3}$, Table 4), the $LRT$ conditions are located at the border between aerosol-limited and transitional regimes (compare Reutter et al. 2009 and Pöschl et l. 2010).

### 3.6 Case study $BB$ on aged biomass burning smoke

During the dry season, the ATTO site is frequently influenced by smoke from biomass burning (BB) activities in different regions of the Amazon forest (Freitas et al., 2005; Andreae et al., 2012; 2015; C. Pöhlker et al., 2018). As a characteristic example, the case study $BB$ focuses on the strong biomass burning plume that was observed at the ATTO site in the middle of August 2014. During this event, pollution aerosol and trace gas concentrations reached their annual maxima with $N_{CN,10}$ peaking at 5000 cm$^{-3}$, $c_{CO}$ at 350 ppb, and $M_{BCe}$ reaching up to 2.5 μg m$^{-3}$, as shown in Fig. 4. Selected meteorological, aerosol, and CCN time series of the event are shown in Fig. 8. This case study provides the opportunity to analyze the CCN properties of aged smoke from a rather defined large-scale $BB$ plume in the Amazon region.

Figure 8 shows that the influence of the $BB$ smoke plume on the ATTO site lasted for about 6 days (18 to 22 August) with a gradual onset and decay. In terms of meteorology, this period was characterized by mostly cloud-free conditions (see SW$_{in}$) without precipitation, comparatively low RH levels, and some variability in wind direction. Note that during the presence of this major smoke plume, an atmospheric dimming in the ATTO region could be recognized (i.e., compare SW$_{in}$ maxima for the seven consecutive days). The signature of the smoke aerosol particles can very clearly be seen in the d$N$/dlog$D$ contour plot, $N_{CN}$, as well as $M_{BCe}$. In terms of particle size, the pronounced increase mostly occurred in the accumulation mode, which appears to be comparatively broad for this specific event. The CCN concentrations – e.g., $N_{CCN}(0.2$ %$)$ and $N_{CCN}(0.5$ %$)$ – track the relative increase in total aerosol abundance and show a 4 to 5-fold increase as well. At the same time, the presence of the pyrogenic aerosols correspond to a clear drop in aerosol hygroscopicity in both, the Aitken ($\Delta\kappa_{Ait} \approx$ -0.05) and accumulation modes ($\Delta\kappa_{Acc} \approx$ -0.1), relative to the conditions before and after the major $BB$ plume (see overlay of $\kappa(S,D_a)$ size distributions and the d$N$/dlog$D$ contour plot in Fig. 8d). This drop in $\kappa(S,D_a)$ is associated with a high organic-to-sulfate, OA/SO$_4^{2-}$, ratio (Fig. 8f), reflecting the dominant role of organic constituents in biomass burning particles as documented in previous studies (Fuzzi et al., 2007; Artaxo et al., 2013; Lathem et al., 2013).

Geographically, the location of the fires and, thus, the origin of the strong biomass burning plume could be found by means of a combination of BTs and satellite data products as shown in Fig. S12 (Draxler and Hess, 1998; Acker and Leptoukh, 2007; Justice et al., 2011). The strong pollution at the ATTO site resulted from the presence of intense fires in the southern Amazon and a temporary 'swing' of the BT track over the fire locations. During the dry season, the BTs mostly belong to the E and ESE clusters (see Fig. S1 and 4). For the period before and after the biomass burning plume, the BT track follows a 'coastal path' and enters the continent in the

region of the Amazon River delta (about 0° N 50° W). Subsequently, the air masses 'follow' the Amazon River in westerly directions to the ATTO site. During the peak period of the *BB* event the BTs deviate from the 'coastal path' and follow an 'inland path' across the southeast of Brazil (see Fig. S12a and Fig. 4). At about 7° S and 55° W, the BTs intersect a region with strong fire activities, which are clearly visible in satellite products such as NO₂ total column measurements. These fires are localized along the Cuiabá-Santarém highway (BR-163). This highway corridor is known as a region of intense logging and burning of primary forest and its conversion to cattle pasture (Nepstad et al., 2002; Fearnside, 2007; C. Pöhlker et al., 2018). The transport time of the smoke from the fires to the ATTO site is about 2-3 days (given by the BTs), which provides a reference for the residence time and aging of the aerosol particles in the atmosphere. The satellite image in Fig. S12b shows the smoke plume that originated in the BR-163 region and traveled northwestwards. It clearly impacts the area around Manaus and the ATTO site precisely during the 'event days' in August.

For a further characterization of the *BB* case study, we calculated a ratio of excess $N_{CN,10}$ to excess $c_{CO}$ ($\Delta N_{CN,10}/\Delta c_{CO}$) for the event of $17.9 \pm 0.7$ cm⁻³ ppb⁻¹ (see Fig. S13a), which agrees well with the typical range for a variety of vegetation fires ($30 \pm 15$ cm⁻³ ppb⁻¹), in contrast to much higher $\Delta N_{CN,10}/\Delta c_{CO}$ levels for urban ($100 – 300$ cm⁻³ ppb⁻¹) and power plant emissions (up to 900 cm⁻³ ppb⁻¹) (Janhäll et al., 2010; Kuhn et al., 2010; Andreae et al., 2012). Furthermore, the ratios of excess $N_{CCN}(S)$ to excess $c_{CO}$, $\Delta N_{CCN}(S)/\Delta c_{CO}$, for the individual $S$ levels range between $6.7 \pm 0.5$ cm⁻³ ppb⁻¹ for the lowest $S = 0.11$ % and values around $18.0 \pm 1.3$ cm⁻³ ppb⁻¹ for higher $S$ (see Table S4 and Fig. S13b). The $\Delta N_{CCN}(S)/\Delta c_{CO}$ ratios converge against $\Delta N_{CN}/\Delta c_{CO}$ already for comparably small $S$ (i.e., $> 0.24$ %), which can be explained by the fact that small $S$ already activate a substantial fraction of the pronounced (mostly pyrogenic) accumulation mode (see discussion below). Kuhn et al. (2010) reported $\Delta N_{CCN}(0.6\%)/\Delta c_{CO}$ ratios around 26 cm⁻³ ppb⁻¹ for biomass burning plumes, which is consistent with our observations (i.e., $\Delta N_{CCN}(0.6\%)/\Delta c_{CO} = 17.9 \pm 1.3$ cm⁻³ ppb⁻¹, see Table S4). The obtained $\Delta N_{CCN}(S)/\Delta c_{CO}$ ratios were utilized in a dedicated CCN parameterization scheme in our part 1 study (M. Pöhlker et al., 2016).

Figure 6e summarizes the corresponding $N_{CN}$ and $N_{CCN}(S)$ size distributions as well as the size dependence of $\kappa(S,D_a)$ for the *BB* case study. It shows a very strong accumulation mode ($D_{acc} \approx 170$ nm, $N_{acc} \approx 3400$ cm⁻³), 'swamping' the Aitken mode ($D_{Ait} \approx 70$ nm, $N_{Ait} \approx 140$ cm⁻³) almost completely, giving the entire distribution a monomodal appearance. For $S < 0.5$ % almost the entire aerosol population is activated as CCN. The averaged $\kappa(S,D_a)$ levels are rather low for both, the Aitken and accumulation modes ($\kappa_{Ait} = 0.14 \pm 0.01$ and $\kappa_{acc} = 0.17 \pm 0.02$), in which $\kappa_{acc}$ is among the lowest values found in the accumulation mode size range throughout the entire study. The low $\kappa(S,D_a)$ levels can be explained by the fact that pyrogenic aerosols predominantly contain organic constituents and rather low levels of inorganic species. The ACSM results during the *BB* period emphasize the predominant mass fraction of organics (87 %) as well as the minor contribution by nitrate (2 %), ammonium (3 %), and sulfate (4 %) (Table 3). The predicted $\kappa_p$ of $0.15 \pm 0.01$ agrees reasonably well with the measurement result, $\kappa(0.10 \%) = 0.18 \pm 0.01$.

Figure 6f displays the characteristic CCN efficiency spectrum for the *BB* period, which levels out clearly below unity. The dashed line shows the erf fit forced to go from zero to one (i.e., with the pre-defined variable $a_1 = 1$), which does not match the data points for $S > 0.5$ %. This fit has been included here, since the activated fraction must reach 100 % eventually. For comparison, the solid line is an erf fit of the data where the plateau is an open fit parameter. This erf fit matches the data points very well and the fit parameters as presented in Table 4 reveal the plateau at 93 %. Physically, this indicates the presence of an externally mixed aerosol population with 7 % of the particles being hydrophobic (e.g., a certain fraction of rather fresh soot) and not acting as CCN in the

measured $S$ range. The slope of the efficiency spectrum shows a steep increase for $S < 0.4$ %. According to these findings, the $\Delta S$ sensitive range is rather small with $S < 0.4$ %. According to Reutter et al. (2009), the $BB$ case study conditions with the large number of available CCN (e.g., $N_{CCN}(0.47\ \%) > 4000\ cm^{-3}$) fall into the updraft-limited regime.

It can be expected that the large aerosol load, its optical properties, and its ability to serve as CCN must already influence cloud properties and the stability of the thermodynamic profile of the atmosphere at both, local and regional scales (Cecchini et al., 2017a). As mentioned above, the incoming solar radiation was dimmed compared to the days before and after this $BB$ event. Furthermore, the cloud fraction decreased, which might result from stabilizing the atmosphere due to increased absorption of solar radiation in and above the boundary layer, the "semidirect effect" (e.g., Koren et al., 2004; Rosenfeld et al., 2008; Rosario et al., 2013). A detailed investigation of the direct radiative forcing and the modification of cloud properties by aerosol particles in the Amazon rain forest are clearly beyond the scope of this study. Nevertheless, the results shown here may serve as input for dedicated regional climate simulations.

### 3.7  Case study *MPOL* on the complex aerosol mixtures during the dry season

The previous $BB$ case study presented the aerosol and CCN properties of a strong and well-defined regional biomass burning plume. However, the dry season aerosol mixture at ATTO can be rather complex due to the influence of a variety of sources, such as biomass burning, fossil fuel combustion, and industrial emissions from local/regional sources as well as from African long-range transport, in addition to the natural aerosol background (i.e., PBAP, SOA, marine aerosols) (e.g., Freud et al., 2008; Artaxo et al., 2013; Saturno et al., 2017a; C. Pöhlker et al., 2018). Accordingly, the dry season aerosol mixture is highly variable and no single episode reflects the conditions comprehensively. Therefore, we defined a mixed pollution (*MPOL*) case study, which is evidently influenced by local/regional biomass burning in South America and the long-range transport of African aerosols. The *MPOL* case serves as an example for the highly variable aerosol conditions during the Amazonian dry season. The 7-day *MPOL* period in September 2014 shows an exceptionally low OA/SO$_4^{2-}$ mass ratio (Table 3). Saturno et al., 2017b recently showed that the high sulfate concentrations can be explained by the long-range transport of volcanogenic aerosols from Africa (i.e., from the Nyiragongo and Nyamuragira volcanoes in Eastern Congo) into the Amazon (compare also Fioletov et al., 2016). While the volcanogenic aerosol shows characteristic properties that allow to discriminate this *LRT* component from the local/regional *BB* aerosol, African *BB* aerosols can be transported similarly, however, an experimental discrimination of African vs. Amazonian *BB* aerosol is difficult (see Saturno et al., 2017a). Upon arrival of the African plume in the Amazonian atmosphere, the LRT aerosol has been mixed with local/regional fires emissions.

Meteorologically, the time frame of the *MPOL* period was mostly cloud-free (see daily profile in SW$_{in}$) and without precipitation (see $P_{TRMM}$ and $P_{ATTO}$), as well as characterized by rather stable wind conditions (Fig. 9). With respect to the aerosol and CCN parameters, the relevant time series, such as $N_{CN}$, $M_{BCe}$, $N_{CCN}(S)$, show a high variability. Specifically, the d$N$/dlog$D$ contour plot shows an alternating pattern: On one hand, and marked by a grey shading, rather extended periods with comparably low aerosol concentrations (around 1000 cm$^{-3}$) and high $\kappa(S,D_a)$ values (up to 0.35) can be observed. The $\kappa(S,D_a)$ levels reaching ~0.35 are among the highest in the entire dry season 2014 (compare Fig. 4). On the other hand, and marked by a light purple shading, these conditions are interrupted by several pulses with strongly enhanced aerosol concentrations (up to 5000 cm$^{-3}$) as well as substantially lower $\kappa(S,D_a)$ values (around 0.1). The changes in $\kappa(S,D_a)$ occur with simultaneous changes in the

aerosol chemical composition: Slight increases of the OA/SO$_4^{2-}$ ratio indicates that the pulses comprise organic-rich aerosol due to the influence of local/regional *BB* plumes, whereas the lower OA/SO$_4^{2-}$ aerosol is comparatively rich in sulfate due to the volcanogenic origin.

A combination of BTs and satellite data during the *MPOL* period shows that all BTs follow an easterly direction along the Amazon River (Fig. S6), which is the most frequent scenario during the dry season (see C. Pöhlker et al., 2018). The same study also shows that during this period, the shores of the Amazon River can be regarded as the core region of the ATTO site footprint, where BT densities are highest. Accordingly, all aerosol and trace gas sources in these areas are of primary importance for the ATTO region. For instance, this is true for a deforestation hotspot in the area around the cities Oriximina and Òbidos as well as a variety of anthropogenic sources (i.e., industry, power plants, cities, shipping) (compare Fig. S14 and C. Pöhlker et al., 2018). During the *MPOL* time frame, several (smaller) deforestation fires were observed within the fetch of the corresponding BTs (Fig. S14). These fires, which mostly burnt for less than a day according to satellite observation, are likely responsible for the low $\kappa(S,D_a)$ and high OA/SO$_4^{2-}$ pulses within the *MPOL* period. Thus, the pulses can be considered as the advection of nearby biomass burning plumes into the ATTO region. According to the BT data, the smoke experienced a transport and aging time of 12-18 h. Thus, the smoke plumes during *MPOL* are fresher than during the *BB* case study, which experienced atmospheric aging for 2-3 days. This difference in aging can be related to differences in the corresponding $\kappa(S,D_a)$ values with fresh smoke during *MPOL* showing a lower hygroscopicity ($\kappa_{<100\,nm,MPOL} \approx 0.10$) and the aged smoke showing higher values ($\kappa_{Ait,BB} \approx 0.14$, $\kappa_{acc,BB} \approx 0.17$). Note that biomass burning smoke cannot explain the sulfate-rich aerosol during *MPOL* since BB-related sulfate contents are typically below 6-7 % (e.g., compare *BB* case in Table 3 as well as Fuzzi et al., 2007; Gunthe et al., 2009; Saturno et al., 2017b).

Figure 6g and h summarize the $N_{CN}(D)$, $N_{CCN}(S,D)$, and $\kappa(S,D_a)$ size distributions during *MPOL*, separately for the biomass burning pulses (*MPOL-BB* with OA/SO$_4^{2-} \approx 4$) and LRT influence (*MPOL-LRT* with lower OA/SO$_4^{2-} \approx 3$). In both cases, the size distributions show a broad monomodal distribution, which did not allow a stable double log-normal mode fitting of Aitken and accumulation modes (see Table 4). The mode is centered at ~135 nm for *MPOL-LRT*, whereas the relatively fresh biomass burning smoke during *MPOL-BB* shows a modal diameter of ~113 nm, in agreement with ~100 nm for fresh biomass burning smoke reported by Andreae et al. (2004). The corresponding $N_{CCN}(S,D)$ size distributions show similar shapes, however, with substantial absolute differences in CCN concentrations. Furthermore, clear differences are also observed for the average $\kappa(S,D_a)$ size distributions: During *MPOL-LRT*, $\kappa(S,D_a)$ shows a pronounced size dependence. The average $\kappa$ for sizes <100 nm is rather size independent ($\kappa_{<100nm} = 0.14 \pm 0.01$). Instead, the average $\kappa_{>100nm}$ increases with $D$ up to 0.26 (on average: $\kappa_{>100nm} = 0.22 \pm 0.03$). According to the shape of the $\kappa(S,D_a)$ size distribution in Fig. 6g, it can be assumed that the aerosol hygroscopicity was even higher for $D$ larger than the covered size range here (i.e., >170 nm). During the dry season, the highest $\kappa(S,D_a)$ levels for accumulation mode particles were observed during the peak abundance of sulfate at the ATTO site. The hygroscopicity during the *MPOL-BB* pulses shows generally lower values and weaker size dependence for particles <150 nm. The average $\kappa_{<150nm} = 0.10 \pm 0.01$ was even smaller than during *BB* and *PR* conditions and represents one of the lowest values measured during the entire observation period. For larger particles (e.g., >150 nm), $\kappa(S,D_a)$ strongly increased towards values comparable to those during *MPOL-LRT*, showing that both conditions are superimposed (on average: $\kappa_{>150nm} = 0.20 \pm 0.04$).

Figure 6i shows the CCN efficiency spectra for the two *MPOL* states. The slope of the *MPOL-LRT* spectrum is significantly steeper than the slope of the *MPOL-BB* spectrum. The $S_1$ level of 50 % activation for the sulfate-rich aerosol population (0.16 %) is clearly lower than during the short biomass burning pulses (0.28 %). This rather large difference can be explained by the smaller modal diameter of the *MPOL-BB* relative to the *MPOL-LRT* case. Furthermore, $\kappa$ is significantly decreased for the biomass burning pulses, which are dominated by organic constituents. However, note that the absolute CCN concentration is – while being less hygroscopic – significantly higher for *MPOL-BB* periods.

In general, the *MPOL* case study emphasizes the following aspects: (i) The dry season aerosols that arrive at the ATTO site can be rather complex mixtures of superimposed emissions from different sources with contrasting chemical properties. (ii) The properties of the aerosol and CCN population can change rather suddenly and substantially (see $N_{CN}$ and $N_{CCN}(S)$ changes by factor of 3-4 within few hours). Similarly, the aerosol hygroscopicity can vary quite strongly from the lowest value observed in the entire study (~0.1) to values around 0.35 and higher, which belong to the largest values observed here during the dry season. These quickly and substantially changing aerosol regimes will presumably also impact the cloud conditions during dry season. (iii) Finally, the Amazonian aerosol and CCN population can be substantially perturbed by emissions from sources (e.g., volcanoes in the Eastern Congo) that are remarkably far away from the ATTO site (~10 000 km) (Saturno et al., 2017b). This emphasizes very clearly that intercontinental influences have to be considered carefully in the analysis of the Amazonian atmospheric composition.

## 3.8 Aerosol particle hygroscopicity distributions and aerosol mixing state

In this section, we investigate the aerosol particle mixing state for the different case study conditions with the help of the aerosol particle hygroscopicity distribution – or $\kappa$ distribution – concept introduced by Su et al. (2010). This approach visualizes the spread of $\kappa$ values among particles of a given size. Specifically, in an ideal internal particle mixture, all particles have the same chemical composition and therefore the same hygroscopicity, resulting in a narrow and defined $\kappa$ distribution. In an external mixture, the particles at a given size can have widely different chemical compositions and hygroscopicities, resulting in a broad $\kappa$ distribution.

Figure 10 summarizes two versions of $\kappa$ distributions for the contrasting case study conditions: (i) the 'classical' $\kappa$ distributions according to Su et al. (2010), which emphasize the spread of $\kappa$ levels for all particle diameters across the measured size range, and (ii) $\kappa$ distributions weighted with the corresponding average particle size distributions from Fig. 6, which provide a quantitative overview of particle abundance as a function of hygroscopicity and size ($N_{CN} \kappa$ distribution). The comparison of the $\kappa$ and $N_{CN} \kappa$ distributions for the contrasting case study conditions emphasizes similarities and differences between the corresponding aerosol populations, which allows drawing conclusions on the aerosol mixing state and microphysical properties.

The $\kappa$ distributions for most conditions reflect a bimodal character of the corresponding aerosol distributions with distinctly different properties in the Aitken and accumulation modes as outlined in the discussion of Fig. 6. Specifically, all distributions show an increasing spread of $\kappa$ levels towards larger particle diameters, which suggests a higher degree of external particle mixing and therefore a higher diversity of particle properties (i.e., hygroscopicity) in the accumulation than in the Aitken mode. As an example, the *BB* and *MPOL-BB* cases show a $\kappa$ spread in the accumulation mode range that reaches from values well below 0.1 up to levels of about 1. Remarkably, the *PR* $\kappa$ distribution differs from all other cases since it shows overall the smallest spread of $\kappa$ over the entire size range. This suggests the Aitken and accumulation mode particles under *PR* conditions are two distinct

and chemically rather homogeneous aerosol populations with a comparatively high degree of internal mixing. As an example, the *PR* Aitken mode particle population cover a defined $\kappa$ range between ~0.1 and ~0.15. In contrast, the *LRT*, *BB*, and *MPOL* aerosol populations appear to be more externally mixed.

In addition to the diversity of the hygroscopicity as visible in the $\kappa$ distributions, the $N_{CN}$ $\kappa$ distributions further emphasize the quantitative abundance of particles in the hygroscopicity-size space. Accordingly, the $N_{CN}$ $\kappa$ distributions can be regarded as signature-like representations of the aerosol microphysical state under certain conditions. Note that the $N_{CN}$ $\kappa$ distributions in Fig. 10 differ substantially from each other. The *PR* case shows a unique signature. The *LRT* and *MPOL-LRT* cases show certain similarities due to the fact that both are characterized by similar aerosol size distributions and comparatively high $\kappa$ levels. The comparison of the relatively fresh *MPOL-BB* smoke and the relatively aged *BB* smoke emphasizes the aging-related increases in particle size and hygroscopicity by means of a characteristic shift of the dominant mode in the $N_{CN}$ $\kappa$ distributions. In both cases the spread of $\kappa$ is rather large, indicating a comparatively strong external mixing in the smoke plumes. In essence, the $\kappa$ distributions and $N_{CN}$ $\kappa$ distributions represent valuable overview representations, which combine characteristic aerosol properties in terms of particle size, particle concentration, $\kappa$ diversity, and aerosol mixing state in a fingerprint-like manner. So far, only very few studies (e.g., Mahish et al., 2018) have used the $\kappa$ distribution or related concepts to characterize ambient aerosol properties. In the light of the results in Fig. 10, we suggest that this concept should be used more broadly as it provides valuable insights into the particle mixing state beyond the more established characterizations of aerosol and CCN properties.

### 3.9 Overview of CCN efficiency spectra and corresponding CCN spectra

CCN efficiency spectra serve as normalized CCN signatures. Their shape is influenced by (i) the shape of the $N_{CN}$ size distribution (i.e., relative strength of Aitken vs. accumulation modes), (ii) the aerosol composition through the $\kappa(S,D_a)$ values and its size dependence as well as (iii) the mixing state of the aerosol as represented in the $\kappa$ and $N_{CN}$ $\kappa$ distributions. CCN spectra provide quantitative information on the actual CCN concentrations as a function of $S$. From the CCN efficiency spectra, CCN spectra are readily obtained by multiplication with the corresponding average $N_{CN}$ concentrations (Table 4). This last section combines the CCN efficiency spectra (i.e., the erf fits) and CCN spectra of all analyzed case study conditions from this work with the seasonally averaged spectra from the companion part 1 study and literature data. Figure 11 shows an overview of all spectra. The direct comparison emphasizes characteristic similarities and differences as a basis for a concluding discussion on aerosol-cloud-interactions. Note in this context that Farmer et al. (2015) presented "cumulative numbers of CCN, normalized to CN", which is conceptually related to the CCN efficiency spectra reported here. Accordingly, a comparison of the spectra reported by Farmer et al. (2015) with Fig. 11 may help to put the characteristics of the Amazonian CCN population into a broader context.

The peak supersaturation and its variability is an essential parameter to understand aerosol-cloud interactions. However, only spares quantitative information on the actually relevant $S$ range is available. The spectra in Fig. 11 are plotted for a broad $S$ range from 0.001 to 20 %. According to Reutter et al. (2009), this range can be generally subdivided into very low ($S < 0.1$ %), low ($S < 0.2$ %), transitional ($0.2$ % $< S < 0.5$ %), and high ($S > 0.5$ %) supersaturation regimes. Furthermore, a recent study by Fan et al. (2018) suggests that under certain conditions even very high supersaturations (S $\gg$ 1 %) can be reached in deep convective clouds. One approach to actually quantify ATTO-relevant average peak $S$ at cloud base, $S_{cloud}(D_H,\kappa)$, uses the position of the Hoppel

minimum, $D_H$, as outlined in Hoppel et al. (1986) and Krüger et al. (2014). According to results in our companion part 1 paper and the present study (see Table 2), ATTO-relevant $S_{cloud}(D_H,\kappa)$ values range from ~0.15 % to ~0.34 %. This $S_{cloud}(D_H,\kappa)$ range is shown in as a '$S$ landmark' in Fig. 11 for orientation.

The array of condition-specific CCN efficiency spectra in Fig. 11a shows the diversity of spectral shapes. The *PR*-specific spectrum represents the lower limiting case with the 'slowest' increase in activated fraction upon increasing $S$ (e.g., 50 % activation reached at S ≈ 0.5 %), whereas the *BB* and *LRT* spectra represent the upper limiting cases, reaching high activated fractions at relatively low $S$ (50 % activation at $S ≈ 0.15$ % for *BB* case and at $S ≈ 0.1$ % for *LRT* case, respectively). The slopes of the spectra, $d(N_{CCN}(S)/N_{CN,10})/dS$, indicate how sensitively the activated fractions react towards changes in supersaturation, $\Delta S$. Overall, the highest sensitivities towards $\Delta S$ occur between ~0.1 % and ~1.0 %. However, the cases have to be investigated individually to obtain condition-specific 'high-sensitivity regimes'. For low $S$ ($< 0.1$ %) – corresponding to the activation of accumulation and coarse mode particles – the aerosol populations that act as CCN most efficiently are those with strong accumulation modes and high $\kappa$ levels, such as the sea salt-rich *LRT* and sulfate-rich *MPOL-LRT* populations. In contrast, the *PR* aerosol acts as CCN least efficiently as it is characterized by a small fraction of accumulation mode particles with rather low $\kappa$. In the transitional $S$ regime with $0.2$ % $< S < 0.5$ % – corresponding to the activation of particles between the Aitken and accumulation modes (i.e., the Hoppel minimum size range) – the individual spectra show their highest divergence and thus the largest differences in CCN efficiency. Note that this range is collocated with the experimentally deduced and ATTO-relevant $S_{cloud}(D_H,\kappa)$ levels. For high $S$ ($> 1$ %) – corresponding to the activation of Aitken mode and even smaller particles – full activation is reached and sensitivities towards $\Delta S$ decrease.

For comparison, the season-specific CCN efficiency spectra from our part 1 study are shown in Fig. 11c. Generally, the shapes of the condition-specific spectra and the season-specific spectra agree well (e.g., relationship between wet vs. dust seasons and *PR* vs. *LRT* conditions). However, the characteristic spectral features are more pronounced for the conditions than for the seasons, due to the fact that the case studies on certain conditions incorporate more defined aerosol plumes and/or populations. As an example, the LRT season spectrum includes several LRT events, however, also clean periods in between, whereas the *LRT* case study spectrum includes only the core period of one defined LRT episode (see Table 1).

Since the size and composition of aerosol particles change dynamically due to atmospheric aging and processing, also the CCN spectra change accordingly. Figure 11e emphasizes the dynamic character of the shape of the CCN efficiency spectra by means of different aging states of biomass burning plumes. Here we combined four biomass burning-related CCN efficiency spectra: (i) the *BB* case study spectrum from this work, which represents smoke after ~2-3 days of atmospheric aging, (ii) the *MPOL-BB* case study spectrum from this work, which represents smoke after ~1 day of aging, (iii) "cloud-processed smoke" from biomass burning regions in southeast Brazil after hours to days of atmospheric aging according to Andreae et al. (2004), and (iv) "fresh smoke in the mixed layer", also from southeastern Brazil, which was sampled in the fire plumes and thus was aged for minutes to hours only (also from Andreae et al., 2004). The spectral shapes in Fig. 11e clearly differ: The lowest activated fractions were observed for the fresh smoke (50 % activation at $S ≈ 1.0$ %; 90 % at $S ≈ 4.0$ %), followed by the "cloud-processed smoke" (50 % activation at $S ≈ 0.5$ %; 90 % at $S ≈ 2.0$ %), then the *MPOL-BB* spectrum (50 % activation at $S ≈ 0.3$ %; 90 % at $S ≈ 1.2$ %), and finally the *BB* spectrum (50 % activation at $S ≈ 0.2$ %; 90 %  at $S ≈ 0.4$ %) with the highest activated fractions as upper limiting case. Atmospheric aging tends to increase the particle size through coagulation and condensational growth as well as to enhance the

particle hygroscopicity through oxidation, aqueous phase chemistry, and reaction product deposition. Therefore, aging tends to increase the strength of the accumulation at the expense of the Aitken mode and increases $\kappa$. Accordingly, it evolves the smoke's CCN efficiency spectra from the "fresh smoke" conditions – as the initial state – towards the *BB* case study conditions and presumably even further. This underlines the well-known trend that atmospheric aging increases the suitability of a given particle population to act as CCN (Andreae and Rosenfeld, 2008). Important to note in the context of this study is the following: The CCN efficiency spectra represent signatures of a given aerosol population in a given state of atmospheric aging. Accordingly, the atmospheric aerosol aging and the dynamic evolution of the CCN efficiency spectra's shape has to be kept in mind upon discussion and utilization of the spectra in follow-up studies.

From all CCN efficiency spectra the corresponding CCN spectra were calculated as displayed in Fig. 11b, d, and f. Moreover, measured CCN spectra from previous mostly short-term campaigns in the Amazon Basin are also shown and agree well with our results (Roberts et al., 2001, 2003; Andreae et al., 2004; Gunthe et al., 2009). Beyond the literature data shown in Fig. 11, good agreement was further observed with Andreae (2009) and Martins et al. (2009). Overall, Fig. 11 may serve as a basis for dedicated cloud-microphysical studies on the characteristic differences between the aerosol/CCN populations and their impacts on cloud properties. Particularly, the results obtained here (i.e., $N_{CN}$ concentrations and CCN efficiency spectra) can be used to investigate the sensitivity of clouds under *PR* conditions, as an approximation for pristine atmospheric conditions in tropical continental regions. The large differences between *PR* background conditions and perturbed atmospheric states related to anthropogenic (aged) *BB* smoke in terms of total aerosol concentration as well as the shape of the CCN efficiency spectra are evident. Similarly, the differences between the *PR* and *LRT* case studies suggest that the frequent *LRT* events in the wet season are related to simultaneous changes in the cloud microphysical state, although the changes in total aerosol concentration are comparatively small.

## 4  Summary and conclusions

In a recent synthesis paper on aerosol-cloud interaction and its highly uncertain representation in global climate models, Seinfeld et al. (2016) proposed that long-term and focused observations in "geographic areas that are critical in climate response" are necessary to obtain a detailed process understanding. Further, studies in "those regions of the present-day atmosphere that approximate preindustrial conditions" will help to "replicate preindustrial aerosol-cloud relationships". In these regions – such as the Amazon Basin – clouds are particularly sensitive to small changes in CCN concentrations (Carslaw et al., 2013). Accordingly, observations are needed here to obtain crucial knowledge on the man-made perturbation of preindustrial aerosol-cloud interactions. This work aims to help tackling this major scientific challenge by presenting detailed long-term aerosol and CCN data for characteristic atmospheric states at the Amazon Tall Tower Observatory (ATTO) in the central Amazon Basin, which is a unique, climate-relevant location with atmospheric conditions oscillation between pristine and anthropogenically strongly perturbed states.

The basis for this work are size-resolved measurements of atmospheric aerosol and cloud condensation nuclei (CCN) concentrations and hygroscopicity at ATTO over a full seasonal cycle (Mar 2014 - Feb 2015). The results of these observations are presented in two papers: The recently published part 1 paper provides an in-depth analyses of the multi-month patterns in the Amazonian CCN population as well as seasonal averages of the key CCN parameters (M. Pöhlker et al., 2016). Further, part 1 compares and discusses different CCN parametrization schemes and their suitability to represent the Amazonian CCN cycling in modelling studies. The present

part 2 study completes this picture by analyzing the CCN variability at the original time resolution (4.5 h), which is sufficient to resolve its short-term variability in relation to air mass changes as well as aerosol emission and transformation processes.

A focal point of both studies is the concept of CCN efficiency spectra, which represent a tool to visualize the different behaviors of contrasting aerosol populations in cloud formation and, thus can be regarded as 'CCN signatures'. Analytically, the CCN efficiency spectra can be described precisely and in a physically correct way by single- or double-error function (erf) fits. In contrast to other common analytical functions, the erf approach describes the measurement results very accurately and allows to extrapolate CCN properties to experimentally hardly accessible supersaturations in the low and high $S$ regimes.

Here, we zoom into the long-term CCN data in two steps: First, we discuss the aerosol and CCN variability for two 2-months periods that represent contrasting regimes in the aerosol, cloud microphysical, and precipitation seasonality in the central Amazon. The selected periods provide insights into the characteristic atmospheric cycling during the clean wet season and the polluted dry season. Second, we focus on the following four selected case study periods, which represent particularly relevant atmospheric states:

Empirically pristine rain forest (*PR*) and, thus, approximately preindustrial conditions are one of the scientifically most relevant atmospheric states at ATTO. Here, we defined *PR* conditions by means of black carbon and CO concentrations as pollution tracers. At ATTO, the *PR* frequency of occurrence peaks between the second half of April and the first half of May. Under *PR* conditions, aerosol concentrations are very low (~290 cm$^{-3}$) and the aerosol population has a characteristic bimodal shape with a dominant Aitken and comparatively weak accumulation mode ($D_{Ait} \approx 70$ nm; $N_{Ait} \approx 160$ cm$^{-3}$ vs. $D_{acc} \approx 160$ nm; $N_{acc} \approx 90$ cm$^{-3}$). The aerosol particles are composed of mostly organic matter with minor amounts of inorganic constituents (OA: 0.64 µg m$^{-3}$, NO$_3^-$: 0.03 µg m$^{-3}$, SO$_4^{2-}$: 0.04 µg m$^{-3}$, Cl$^-$: < 0.01 µg m$^{-3}$, BC$_e$: < 0.01 µg m$^{-3}$). The observed low $\kappa(S,D_a)$ levels agree with the particles' composition and show a size dependence ($\kappa_{Ait} = 0.12 \pm 0.01$ vs. $\kappa_{acc} = 0.18 \pm 0.01$). The CCN efficiency spectrum shows a characteristic shape since it converges towards full activation rather slowly (50 % activation at ~0.5 %, 90 % activation at ~1.5 %). Thus, the CCN population is sensitive towards $\Delta S$ in both, the low and high $S$ regime. Accordingly, the CCN population can be regarded as both, aerosol sensitive due to the low total aerosol concentrations and updraft sensitive according to the CCN efficiency spectrum with its sensitivity to $\Delta S$ across a wide $S$ range.

During long-range transport (*LRT*) episodes within the wet season, major amounts of aged North African dust, West African biomass burning smoke, and Atlantic marine aerosols are advected on an event basis into the basin (mostly Feb - Apr). Total aerosol concentrations (~440 cm$^{-3}$) are slightly enhanced relative to the *PR* state and the aerosol population has a bimodal shape with a minor Aitken and a stronger accumulation mode ($D_{Ait} \approx 80$ nm, $N_{Ait} \approx 120$ cm$^{-3}$ vs. $D_{acc} \approx 180$ nm, $N_{acc} \approx 300$ cm$^{-3}$). Besides the non-refractory fraction of organics and inorganics (OA: 1.81 µg m$^{-3}$, NO$_3^-$: 0.08 µg m$^{-3}$, NH$_4^+$: 0.30 µg m$^{-3}$, SO$_4^{2-}$: 0.25 µg m$^{-3}$, Cl$^-$: 0.04 µg m$^{-3}$, BC$_e$: 0.21 µg m$^{-3}$), a larger refractory fraction of mineral dust and sea salts can be found in internally mixed particles. The observed $\kappa(S,D_a)$ levels are increased compared to the *PR* state in agreement with the chemical composition ($\kappa_{Ait} = 0.18 \pm 0.02$ vs. $\kappa_{acc} = 0.35 \pm 0.04$). The CCN efficiency spectrum shows a steep increase – and thus high sensitivity to $\Delta S$ – at low $S$ and quickly converges against full activation towards high $S$ (50 % activation at ~0.09 %, 90 % activation at ~0.53 %). Thus, the CCN regime under *LRT* influence is at the border of the aerosol limited and transitional regimes.

Biomass burning (*BB*) is the predominant anthropogenic influence during the Amazonian dry season, which alters the atmospheric composition substantially. During the *BB* event analyzed here, we found strongly enhanced aerosol concentrations (~3600 cm$^{-3}$) and a size distribution dominated by the accumulation mode ($D_{Ait} \approx 70$ nm, $N_{Ait} \approx 140$ cm$^{-3}$ vs. $D_{acc} \approx 170$ nm, $N_{acc} \approx 3400$ cm$^{-3}$). The aged smoke particles comprise mostly organic matter (OA: 21.1 µg m$^{-3}$, NO$_3^-$: 0.55 µg m$^{-3}$, NH$_4^+$: 0.58 µg m$^{-3}$, SO$_4^{2-}$: 0.82 µg m$^{-3}$, Cl$^-$: 0.03 µg m$^{-3}$, BC$_e$: 0.89 µg m$^{-3}$). The observed $\kappa(S,D_a)$ levels were comparatively low and weakly size dependent ($\kappa_{Ait} = 0.14 \pm 0.01$ vs. $\kappa_{acc} = 0.17 \pm 0.02$). The corresponding CCN efficiency spectrum shows a rather steep increase at low *S* and converges against a threshold (~93 %) below full activation at high *S* (50 % activation at $S \approx$ 0.15 %). With respect to the large particle concentrations and the large sensitivity of the CCN efficiency spectrum to $\Delta S$ in the low *S* regime, the *BB* case falls into the updraft limited regime.

During the dry season, the central Amazon often experiences mixed aerosol populations from regional, continental, and even trans-continental sources. Here, we analyzed one characteristic example of such mixed pollution (*MPOL*) scenarios, in which a mixture of African volcanogenic emissions and nearby Amazonian fires impacted the ATTO site: Under conditions with a predominant influence of the African volcanogenic emissions, we found a broad monomodal size distribution ($D_{mode} \approx 130$ nm, $N_{mode} \approx 1300$ cm$^{-3}$), strongly enhanced sulfate levels (OA: 5.50 µg m$^{-3}$, SO$_4^{2-}$: 1.75 µg m$^{-3}$), and correspondingly elevated hygroscopicities ($\kappa_{<100nm} = 0.14 \pm 0.01$ vs. $\kappa_{>100nm} = 0.22 \pm 0.03$). Under conditions with predominant influence by the nearby fires, we found high concentrations in a monomodal distribution ($D_{mode} \approx 110$ nm, $N_{mode} \approx 2800$ cm$^{-3}$), an enhancement of organic matter on top of the sulfate background (OA: 7.88 µg m$^{-3}$, SO$_4^{2-}$: 2.03 µg m$^{-3}$), and low hygroscopicities ($\kappa_{<150nm} = 0.10 \pm 0.01$ vs. $\kappa_{>150nm} = 0.20 \pm 0.04$). Accordingly, the interplay of the aged volcanogenic plume and the fresh smoke resulted in large variations of the total aerosol concentration, aerosol composition, and CCN properties. We suppose that the highly variable CCN population results in associated (microphysical) variations in cloud properties.

Hygroscopicity distributions were analyzed for all conditions, providing detailed and characteristic insights into the mixing state of the different types of aerosols. We found that the spread of $\kappa$ increases with size for all conditions. The $\kappa$ distributions suggest that the *PR* aerosol population is rather internally mixed, whereas the *BB*, *LRT*, and *MPOL* aerosols show more external mixing states. In essence, the $\kappa$ distributions and $N_{CN}$ $\kappa$ distributions represent valuable overview representations, combining characteristic aerosol properties in terms of particle size, particle concentration, $\kappa$ levels, and aerosol mixing state in a fingerprint-like manner. This representation helps to further understand aerosol-cloud interactions, such as the shapes of the CCN efficiency spectra: As general tendencies, more externally mixed aerosols, resulting in broader $\kappa$ distributions, also broaden the CCN efficiency spectra – analogous to broad $N_{CN}$ distributions. Internally mixed aerosols with more defined $\kappa$ distributions tend towards steeper segments in the CCN efficiency spectra. Furthermore, externally mixed aerosols with distinctly different $\kappa$ levels tend to introduce further steps/plateaus into the CCN efficiency spectra, in addition to plateaus caused by multimodal $N_{CN}$ distributions. However, the CCN efficiency spectra for the conditions reported here are primarily shaped by the particle size distributions and average $\kappa$ levels, whereas the diversity of $\kappa$ seems to play a minor role. To clarify exactly how the signatures and patterns in $\kappa$ and $N_{CN}$ $\kappa$ distributions are related to the signatures and shapes of the CCN efficiency spectra, dedicated future studies at contrasting locations and modelling support is required.

Finally, the CCN efficiency spectra and CCN spectra for all analyzed cases are discussed in an overview, which emphasizes the following observations: (i) The combination of CCN efficiency spectra for all analyzed

conditions show a large variability. (ii) Within this range, the estimated peak supersaturations at cloud base, $S_{\text{cloud}}(D_{\text{H}},\kappa)$, are collocated with the intermediate $S$ range, in which the CCN efficiency spectra shows their highest variability and, thus underlines the impact of changing aerosol populations on cloud properties. (iii) The sensitivity of the CCN populations to changes in $S$ is conditions-specific with highest susceptibilities towards $\Delta S$ between $S = 0.1$ % and $S = 1$ %. (iv) The combination of CCN efficiency spectra for different, particularly contrasting, aerosol populations provides a basis for follow-up studies on the aerosol-related differences in cloud properties in the Amazon region and beyond. (v) Finally, the atmospheric aging of aerosol particles in the atmosphere is reflected in a corresponding evolution of the shape of the CCN efficiency spectra and has to be kept in mind upon their utilization.

## 5 Data availability

The data of the key results presented here have been deposited in supplementary data files for use in follow-up studies. The data has been made available in NASA Ames format. Specifically, all *PR* filters are available as time series of corresponding flags. Moreover, all data from Fig. 6 (i.e., $N_{\text{CN}}(D)$, $N_{\text{CCN}}(S,D)$, and $\kappa(S,D_{\text{a}})$ size distributions as well as measured $N_{\text{CCN}}(S)/N_{\text{CN},10}$ and $N_{\text{CCN}}(S)$ values) are provided. All numbers are given as mean and one std. The time series of the CCN key parameters for the entire measurement period can be found in the supplementary material of the part1 paper (M. Pöhlker et al., 2016). For data requests beyond the available data, please refer to the corresponding authors.

**Acknowledgements**

This work has been supported by the Max Planck Society (MPG) and the Max Planck Graduate Center with the Johannes Gutenberg University Mainz (MPGC). For the operation of the ATTO site, we acknowledge the support by the German Federal Ministry of Education and Research (BMBF contract 01LB1001A) and the Brazilian Ministério da Ciência, Tecnologia e Inovação (MCTI/FINEP contract 01.11.01248.00) as well as the Amazon State University (UEA), FAPEAM, LBA/INPA and SDS/CEUC/RDS-Uatumã. This work has been financial support from the German Research Foundation (DFG grant KA 2280/2); the St. Petersburg state University, Russia (project 11.37.220.2016); as well as the EU FP7 project BACCHUS (project no. 603445); the Sao Paulo Research Foundation (FAPESP grants 13/05014-0 and 13/50510-5); and the Atmospheric System Research program, US Department of Energy. Further, the results in section 3 were supported within RSF grant 18-17-00076. This paper contains results of research conducted under the Technical/Scientific Cooperation Agreement between the National Institute for Amazonian Research, the State University of Amazonas, and the Max-Planck-Gesellschaft e.V.; the opinions expressed are the entire responsibility of the authors and not of the participating institutions. We highly acknowledge the support by the Instituto Nacional de Pesquisas da Amazônia (INPA). The Office of Biological and Environmental Research of the Office of Science is acknowledged for funding, specifically the Atmospheric Radiation Measurement (ARM) Climate Research Facility and the Atmospheric System Research (ASR) Program. We would like to especially thank all the people involved in the technical, logistical, and scientific support of the ATTO project, in particular Susan Trumbore, Carlos Alberto Quesada, Matthias Sörgel, Thomas Disper, Andrew Crozier, Bettina Weber, Nina Ruckteschler, Uwe Schulz, Steffen Schmidt, Antonio Ocimar Manzi, Alcides Camargo Ribeiro, Hermes Braga Xavier, Elton Mendes da Silva, Nagib Alberto de Castro Souza, Adir Vasconcelos Brandão, Amauri Rodriguês Perreira, Antonio Huxley Melo Nascimento, Thiago de Lima Xavier, Josué Ferreira de Souza, Roberta Pereira de Souza, Bruno Takeshi, and Wallace Rabelo

Costa. We acknowledge the use of FIRMS data and imagery from the Land, Atmosphere Near real-time Capability for EOS (LANCE) system operated by the NASA/GSFC/Earth Science Data and Information System (ESDIS) with funding provided by NASA/HQ. Satellite product analyses and visualizations used in this paper were produced with the Giovanni online data system, developed and maintained by the NASA GES DISC. We

5     also acknowledge the MODIS and OMI mission scientists and associated NASA personnel for the production of the data used in this research effort. Further, we thank the GoAmazon2014/5 team for the fruitful collaboration and discussions. We appreciate the help of Hauke Paulsen with the fire map analysis. We thank Peter Hoor for stimulating ideas on CO-based *PR* filtering. Moreover, we thank Qiaoqiao Wang, Oliver Lauer, Jing Ming, Maria Praß, Kathrin Reinmuth-Selzle, and Daniel Rosenfeld for support and stimulating discussions. Finally, we

10     appreciate the constructive comments by two anonymous referees, which were very fruitful to extend and improve important aspects of the paper.

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

**Table A1. List of frequently used symbols.**

| Symbol | Quantity and Unit |
|---|---|
| $a_{mode}$ (mode =1, 2) | prefactor of single- and double erf fits |
| $D$ | mobility equivalent particle diameter, nm |
| $D_a(S)$ | midpoint activation diameter determined from CCN activation curve, nm |
| $D_{Ait}$ | position of Aitken mode maximum, nm |
| $D_{Acc}$ | position of accumulation mode maximum, nm |
| $D_H$ | position of Hoppel minimum, nm |
| $D_{cut}$ | cut-off diameter of aerosol impactor, nm |
| $f$ | frequency of occurrence |
| $\kappa$ | hygroscopicity parameter |
| $\kappa(S,D_a)$ | hygroscopicity parameter determined from CCN activation curve |
| $\kappa_{Acc}$ | mean hygroscopicity parameter for accumulation mode particles |
| $\kappa_{Ait}$ | mean hygroscopicity parameter for Aitken mode particles |
| $\kappa_p$ | predicted hygroscopicity parameter based on ACSM results |
| $MAF(S)$ | maximum activated fraction determined by CCN activation curve |
| $M_{BCe}$ | mass concentration of $BC_e$, µg m$^{-3}$ |
| $M_{BCe}^{*}$ | threshold $M_{BCe}$ level for definition of $PR_{BC}$ periods, µg m$^{-3}$ |
| $M_{Org}$ | ACSM-derived organic mass concentration, µg m$^{-3}$ |
| $M_{NO3}$ | ACSM-derived nitrate mass concentration, µg m$^{-3}$ |
| $M_{NH4}$ | ACSM-derived ammonium mass concentration, µg m$^{-3}$ |
| $M_{SO4}$ | ACSM-derived sulfate mass concentration, µg m$^{-3}$ |
| $M_{Cl}$ | ACSM-derived chloride mass concentration, µg m$^{-3}$ |
| $N$ | number of data points |
| $N_{CCN}(S)$ | CCN number concentration at a certain $S$, cm$^{-3}$ |
| $N_{CCN}(S,D)$ | CCN number concentration at a certain $S$ and $D$, cm$^{-3}$ |
| $N_{CCN}(S) / N_{CN,10}$ | CCN efficiency |
| $N_{CN,Dcut}$ | CN number concentration ($>D_{cut}$), cm$^{-3}$ |
| $N_{CN,10}$ | CN number concentration ($>10$ nm), cm$^{-3}$ |
| $N_{CN,Acc}$ | accumulation mode particle number concentration, cm$^{-3}$ |
| $N_{CN,Ait}$ | Aitken mode particle number concentration, cm$^{-3}$ |
| $N_d$ | Cloud droplet number concentration, cm$^{-3}$ |
| $N_{db}$ | Cloud droplet number concentration at cloud base, cm$^{-3}$ |
| $P_{ATTO}$ | precipitation rate at ATTO, mm h$^{-1}$ |
| $P_{BT}$ | cumulative precipitation from HYSPLIT BTs, mm |
| $P_{TRMM}$ | precipitation from TRMM satellite mission, mm h$^{-1}$ |
| $r_e$ | cloud droplet effective radius, µm |
| $S$ | water vapor supersaturation, % |
| $S_{cloud}(D_H,\kappa)$ | average peak supersaturation in cloud, % |
| $S_{mode}$ (mode =1, 2) | midpoint activation supersaturation from CCN efficiency spectra, % |
| $SW_{in}$ | incoming short-wave radiation (285 to 2800 nm), W m$^{-2}$ |
| $\theta_e$ | equivalent potential temperature, K |
| $U$ | wind speed, m s$^{-1}$ |
| $w_b$ | updraft velocity at cloud base, m s$^{-1}$ |
| $w_{mode}$ (mode =1, 2) | width of CCN efficiency spectra |
| $\sigma$ | width of log-normal fit in Aitken and accumulation mode fitting |
| $\sigma_{ap}$ | aerosol absorption coefficient, Mm$^{-1}$ |
| $x_0$ | position of log-normal fit in Aitken and accumulation mode fitting, nm |

**Table A2. List of frequently used acronyms.**

| Acronym | Description |
| --- | --- |
| ACSM | aerosol chemical speciation monitor |
| ATTO | Amazon tall tower observatory |
| BB | biomass burning |
| BC | black carbon |
| $BC_e$ | equivalent black carbon |
| BL | boundary layer |
| BT | backward trajectory |
| BVOC | biogenic volatile organic compound |
| CCN | cloud condensation nuclei |
| CCNC | cloud condensation nuclei counter |
| CPC | condensation particle counter |
| DSD | droplet size ditribution |
| erf | Gaussian error function |
| FIRMS | fire information for resource management system |
| GDAS | global data assimilation system |
| GoAmazon2014/5 | green ocean Amazon 2014/5 |
| HYSPLIT | hybrid single particle Lagrangian integrated trajectory model |
| IOP | intensive observation period |
| lpm | liters per minute |
| LRT | long-range transport |
| ITCZ | intertropical convergence zone |
| MAAP | multi-angle absorption photometer |
| MAC | mass absorption coefficient |
| MODIS | moderate resolution imaging spectroradiometer |
| MPOL | mixed pollution |
| MPOL-BB | mixed pollution with dominance of regional BB |
| MPOL-LRT | mixed pollution with dominance of LRT |
| NOAA | national oceanic and atmospheric administration |
| NASA | national aeronautics and space administration |
| NPF | new particle formation |
| OA | organic aerosol |
| PBAP | primary biological aerosol particles |
| PR | empirically pristine rain forest aerosol conditions |
| $PR_{BC}$ | BC-based PR filter |
| $PR_{CO}$ | CO-based PR filter |
| $PR_{BC \cap CO}$ | PR filter as intersection of $PR_{BC}$ and $PR_{CO}$ sets |
| $PR_{BC \cup CO}$ | PR filter as union of $PR_{BC}$ and $PR_{CO}$ sets |
| rBC | refractory black carbon |
| RH | relative humidity |
| $ROI_{ATTO}$ | region of interest, covering ATTO region |
| SE | standard error |
| SMPS | scanning mobility particle counter |
| SOA | secondary organic aerosol |
| SP2 | single particle soot photometer |
| se | standard error |
| std | standard deviation |
| UT | upper troposphere |
| UTC | coordinated universal time |

**Table 1. Overview of conditions and corresponding time frames for detailed aerosol and CCN analysis. Note that time frames of different length were averaged for certain aerosol and CCN data products (i.e., size distributions, composition, CCN key parameters) depending on data availability.**

| Conditions | | Time frames [UTC] | Specific remarks |
|---|---|---|---|
| empirically pristine rain forest (*PR*) aerosol | *all available PR periods* | defined in Sect. 2.7; filter time series available as separate data file | - For $N_{CN}(D)$, $N_{CCN}(S,D)$, $\kappa(S,D_a)$, CCN efficiency spectra (Fig. 6a and b): all available $PR_{BC \cap CO}$ episodes within CCN measurement period (Mar 2014-Feb 2015) were averaged. |
| | *PR1* | 24 Apr 06:00 - 29 April 10:00 2014 | - Episodes *PR1*, *PR2*, and *PR3*, highlighted in Fig. 3 and shown in detail in Fig. 5. |
| | *PR2* | 04 May 23:00 - 08 May 10:00 2014 | - For ACSM results (Table 3): all $PR_{BC \cap CO}$ episodes within time frame 01 Aug 2014 to 30 Sep 2016 were averaged. |
| | *PR3* | 16 May 06:00 - 17 May 16:00 2014 | |
| long-range transport (*LRT*) aerosol (in wet season) | *LRT* | 09 Apr 12:00 - 13 Apr 12:00 2014 | - *LRT* time frame averaged for $N_{CN}(D)$, $N_{CCN}(S,D)$, $\kappa(S,D_a)$, CCN efficiency spectra (Fig. 6c and d). <br> - *LRT* episode highlighted in Fig. 3 and shown in detail in Fig. 7. <br> - For ACSM results (Table 3): all LRT episodes according to Moran-Zuloaga et al., 2017 within time frame 01 Aug 2014 to 30 Sep 2016 were averaged. |
| | *all available LRT episodes* | see Moran-Zuloaga et al. (2017) | |
| biomass burning (*BB*) aerosol | *BB* | 18 Aug 00:00 - 22 Aug 00:00 2014 | - *BB* time frame averaged for: $N_{CN}(D)$, $N_{CCN}(S,D)$, $\kappa(S,D_a)$, CCN efficiency spectrum (Fig. 6e and f) and ACSM results (Table 3). <br> - *BB* episode highlighted in Fig 4 and shown in detail in Fig. 8. |
| mixed pollution (*MPOL*) aerosol (in dry season) | *MPOL-BB* | Entire period: 22 Sep 00:40 - 01 Oct 03:30 2014 (for details refer to Fig. 9) | - Subcategories *MPOL-BB* and *MPOL-LRT* during *MPOL* episode averaged independently for: $N_{CN}(D)$, $N_{CCN}(S,D)$, $\kappa(S,D_a)$, CCN efficiency spectra in (Fig. 6g, h and i), and ACSM averages (Table 3). <br> - *MPOL* highlighted in Fig. 4 and shown in detail in Fig. 9. |
| | *MPOL-LRT* | | |

**Table 2. Properties (position, $x_0$, integral number concentration, $N_{CN}$, width, $\sigma$) of Aitken and accumulation modes from single or double log-normal fits of the total particle size distribution. Values are given as means of the case study periods, whereas corresponding seasonally averaged results can be found in the part 1 study. The errors represent the uncertainty of the fit parameters. No meaningful double log-normal fit was obtained for the monomodal *MPOL* case – thus, a single log-normal fit was conducted to describe the properties of the main peak. For the cases $PR_{BC \cap CO}$ and *LRT* with clearly resolved bimodal size distributions, values for the position of the Hoppel minimum, $D_H$, as intersection of fitted and normalized modes as well as estimated average peak supersaturation in cloud, $S_{cloud}(D_H, \kappa)$, according to Krüger et al. (2014) and part 1 study are listed. The error in $S_{cloud}(D_H, \kappa)$ is the experimentally derived error in $S$. Analogous information for the $PR_{BC}$, $PR_{CO}$, and $PR_{BC \cup CO}$ filters can be found in Table S1.**

| Conditions | Mode or size range | $N_{CN}$ [cm⁻³] | $\kappa$ | $x_0$ [nm] | $\sigma$ | $R^2$ | $D_H$ [nm] | $S_{cloud}(D_H,\kappa)$ [%] |
|---|---|---|---|---|---|---|---|---|
| empirically pristine rain forest (*PR*) aerosol ($PR_{BC \cap CO}$ filter) [a] | Aitken | $162 \pm 4$ | $0.12 \pm 0.01$ | $69 \pm 2$ | $0.46 \pm 0.01$ | 0.99 | $103 \pm 2$ | $0.29 \pm 0.01$ |
| | accumulation | $86 \pm 8$ | $0.18 \pm 0.01$ | $157 \pm 1$ | $0.44 \pm 0.01$ | | | |
| long-range transport (*LRT*) aerosol | Aitken | $125 \pm 12$ | $0.18 \pm 0.02$ | $79 \pm 3$ | $0.60 \pm 0.03$ | 0.99 | $118 \pm 2$ | $0.15 \pm 0.01$ |
| | accumulation | $313 \pm 11$ | $0.35 \pm 0.04$ | $179 \pm 2$ | $0.52 \pm 0.01$ | | | |
| biomass burning (*BB*) aerosol | Aitken | $140 \pm 29$ | $0.14 \pm 0.01$ | $70 \pm 1$ | $0.20$ [b] | 0.99 | -- | -- |
| | accumulation | $3439 \pm 39$ | $0.17 \pm 0.02$ | $167 \pm 1$ | $0.58 \pm 0.01$ | | | |
| mixed pollution (*MPOL*) aerosol — *MPOL-LRT* | < 100 nm | $1272 \pm 27$ | $0.14 \pm 0.01$ | $135 \pm 5$ | $0.85 \pm 0.01$ | 0.99 | -- | -- |
| | > 100 nm | | $0.22 \pm 0.03$ | | | | | |
| *MPOL-BB* | < 150 nm | $2764 \pm 84$ | $0.10 \pm 0.01$ | $113 \pm 1$ | $0.81 \pm 0.01$ | 0.99 | -- | -- |
| | > 150 nm | | $0.20 \pm 0.04$ | | | | | |

[a] Double log-normal fits for the *PR* cases were limited to size range 50 to 350 nm since presence of particles in nucleation mode size range (<50 nm) interferes with fit of Aitken mode.
[b] Width of Aitken mode in double log-normal fit of *BB* case was predefined to ensure meaningful convergence of fit.

**Table 3. Aerosol chemical composition from ACSM measurements at ATTO for characteristic conditions and seasons. ACSM data was available for time period from 01 Aug 2014 to 30 Sep 2016 and the averaged values are shown as mean ± se (se was rounded up if se < 0.01). The shown ACSM 3σ detection limits for 30 min averaging time were obtained from Ng et al. (2011). The MAAP-based $M_{BCe}$ detection limits are specified in Sect. 2.7. Note that $M_{NH4}$, $M_{Cl}$, and $M_{BCe}$ ranged below the instruments detection limit in certain cases, which makes the corresponding results unreliable. The predicted average aerosol hygroscopicity parameter, $κ_p$, was calculated according to the part 1 study (M. Pöhlker et al., 2016) and is shown as mean ± se. The $κ(0.10 \%)$ results are shown with the experimentally derived error. Analogous information for the $PR_{BC}$, $PR_{CO}$, and $PR_{BC∪CO}$ filters can be found in Table S2.**

| Conditions and seasons | | Mass concentrations $M_{species}$ [µg m⁻³]  &  (Mass fraction [%]) | | | | | | $OA/SO_4^{2-}$ | $κ_p$ | $κ(0.10 \%)$ |
| --- | --- | --- | --- | --- | --- | --- | --- | --- | --- | --- |
| | | OA | NO₃⁻ | NH₄⁺ | SO₄²⁻ | Cl | BCe | | | |
| empirically pristine rain forest (PR) aerosol ($PR_{BC∩CO}$ filter) | | 0.64 ± 0.02 *(90)** | 0.03 ± 0.01 *(4)** | 0.17 ± 0.01 [a] *(--)** | 0.04 ± 0.01 *(6)** | < (0.01 ± 0.01) [a] *(--)** | < (0.01 ± 0.01) [a] *(--)** | ~ 53 | 0.16 ± 0.01* | 0.18 ± 0.05 |
| Long-range transport (LRT) aerosol | | 1.81 ± 0.04 *(67)* | 0.08 ± 0.01 *(3)* | 0.30 ± 0.01 *(11)* | 0.25 ± 0.01 *(9)* | 0.04 ± 0.01 *(1)* | 0.21 ± 0.01 *(8)* | ~ 24 | 0.24 ± 0.01 | 0.35 ± 0.04 |
| Biomass burning (BB) aerosol | | 21.14 ± 0.50 *(88)* | 0.55 ± 0.02 *(2)* | 0.58 ± 0.02 *(2)* | 0.82 ± 0.02 *(3)* | 0.03 ± 0.01 *(0)* | 0.89 ± 0.03 *(4)* | ~ 26 | 0.15 ± 0.01 | 0.18 ± 0.01 |
| Mixed pollution (MPOL) aerosol | MPOL-LRT | 5.50 ± 0.06 *(65)* | 0.22 ± 0.01 *(3)* | 0.54 ± 0.01 *(6)* | 1.75 ± 0.03 *(21)* | 0.03 ± 0.01 *(0)* | 0.37 ± 0.01 *(4)* | ~ 3 | 0.28 ± 0.01 | 0.26 ± 0.05 |
| | MPOL-BB | 7.88 ± 0.18 *(68)* | 0.36 ± 0.03 *(3)* | 0.68 ± 0.03 *(6)* | 2.03 ± 0.08 *(18)* | 0.05 ± 0.01 *(0)* | 0.57 ± 0.02 *(5)* | ~ 4 | 0.26 ± 0.01 | 0.24 ± 0.02 |
| wet season (Feb-May) | | 1.02 ± 0.01 *(78)** | 0.05 ± 0.01 *(4)** | 0.20 ± 0.03 [a] *(--)** | 0.14 ± 0.01 *(11)** | 0.02 ± 0.01 *(1)** | 0.07 ± 0.01 *(5)** | ~ 23 | 0.19 ± 0.01* | 0.21 ± 0.05 |
| transition periods (Jun/Jul & Dec/Jan) | | 3.26 ± 0.04 *(77)* | 0.11 ± 0.01 *(3)* | 0.32 ± 0.01 *(7)* | 0.32 ± 0.01 *(8)* | 0.02 ± 0.01 *(1)* | 0.20 ± 0.01 *(5)* | ~ 21 | 0.21 ± 0.01 | 0.24 ± 0.05 |
| dry season (Aug-Nov) | | 5.86 ± 0.05 *(79)* | 0.19 ± 0.01 *(3)* | 0.33 ± 0.01 *(5)* | 0.64 ± 0.01 *(9)* | 0.02 ± 0.01 *(0)* | 0.34 ± 0.01 *(5)* | ~ 11 | 0.18 ± 0.01 | 0.21 ± 0.04 |
| Detection limits of ACSM and MAAP | | 0.15 | 0.01 | 0.28 | 0.02 | 0.01 | 0.01 | -- | -- | -- |

[a] Here, the measured $M_{species}$ ranged below the instrument detection limits and the shown values are questionable. In these cases (i.e., $PR_{BC∩CO}$ and wet season average), mass fractions and $κ_p$ were calculated by omitting the corresponding $M_{species}$ as outlined in Sect. 2.1. The corresponding mass fractions and $κ_p$ are marked by *.

**Table 4.** Error function (erf) fit parameters describing CCN efficiency spectra, $N_{CCN}(S)/N_{CN,10}$ vs. $S$, as model input data representing (i) $PR_{BC \cap CO}$, *LRT*, *BB*, and *MPOL* conditions according to present work, (ii) seasonal averages from the companion part 1 paper (M. Pöhlker et al., 2016), and (iii) CCN efficiency spectra adapted from Roberts et al. (2001) and Andreae et al. (2004). A reference concentration of $N_{CN,10}$ has been used in all cases. $N_{CN}$ is shown as mean ± se. The errors in $S_{mode}$ and $w_{mode}$ represent the uncertainty of the fit parameters (see Sect. 2.2). All CCN efficiency spectra are plotted in Fig. 11 for comparison. Analogous information for the $PR_{BC}$, $PR_{CO}$, and $PR_{BC \cup CO}$ filters can be found in Table S3.

| Conditions and seasons | | $N_{CN}$ [cm$^{-3}$] | erf fit | mode | $a_{mode}$ | $S_{mode}$ [%] | $w_{mode}$ | $R^2$ | comment |
|---|---|---|---|---|---|---|---|---|---|
| empirically pristine rain forest (*PR*) aerosol (*PR*$_{BC \cap CO}$ filter) | | 260 ± 3 | single | 1 | 1 | 0.43 ± 0.01 | 1.61 ± 0.07 | 0.99 | fit parameters for CCN efficiency spectra representing conditions defined in present work |
| | | | double | 1 | 1 | 0.11 ± 0.01 | 0.46 ± 0.30 | 0.99 | |
| | | | | 2 | 0.21 ± 0.09 | 0.61 ± 0.08 | 1.13 ± 0.20 | | |
| long-range transport (*LRT*) aerosol | | 439 ± 3 | single | 1 | 1 | 0.09 ± 0.01 | 1.92 ± 0.15 | 0.97 | |
| biomass burning (*BB*) aerosol | | 3584 ± 28 | single | 1 | 1 | 0.15 ± 0.01 | 1.15 ± 0.13 | 0.96 | |
| | | | | 1 | 0.93 ± 0.01 | 0.14 ± 0.01 | 0.89 ± 0.06 | 0.99 | |
| mixed pollution (*MPOL*) aerosol | *MPOL-LRT* | 1277 ± 6 | single | 1 | 1 | 0.16 ± 0.01 | 1.70 ± 0.08 | 0.99 | |
| | *MPOL-BB* | 2777 ± 38 | single | 1 | 1 | 0.28 ± 0.01 | 1.60 ± 0.04 | 0.99 | |
| wet season | | 323 ± 2 | single | 1 | 1 | 0.35 ± 0.01 | 1.80 ± 0.06 | 0.99 | fit parameters for seasonal CCN efficiency spectra from part 1 study (M. Pöhlker et al., 2016) |
| LRT season | | 426 ± 3 | single | 1 | 1 | 0.22 ± 0.01 | 2.39 ± 0.10 | 0.98 | |
| transitions periods | | 943 ± 4 | single | 1 | 1 | 0.28 ± 0.01 | 1.70 ± 0.05 | 0.99 | |
| dry season | | 1528 ± 5 | single | 1 | 1 | 0.18 ± 0.01 | 1.57 ± 0.11 | 0.98 | |
| "cloud-processed smoke" conditions | | 2000 - 8000 | single | 1 | 1 | 0.47 ± 0.02 | 1.64 ± 0.15 | 0.97 | fit parameters for CCN efficiency spectra for biomass burning smoke conditions from Andreae et al. (2004) |
| "fresh smoke" conditions | | 2000 - 8000 | single | 1 | 1 | 1.00 ± 0.05 | 1.56 ± 0.12 | 0.98 | |
| wet season period (Mar/Apr 1998) | | 390 ± 250 | single | 1 | 1 | 0.62 ± 0.02 | 1.46 ± 0.07 | 0.99 | fit parameters for CCN efficiency spectra representing wet season period from Roberts et al. (2001) |

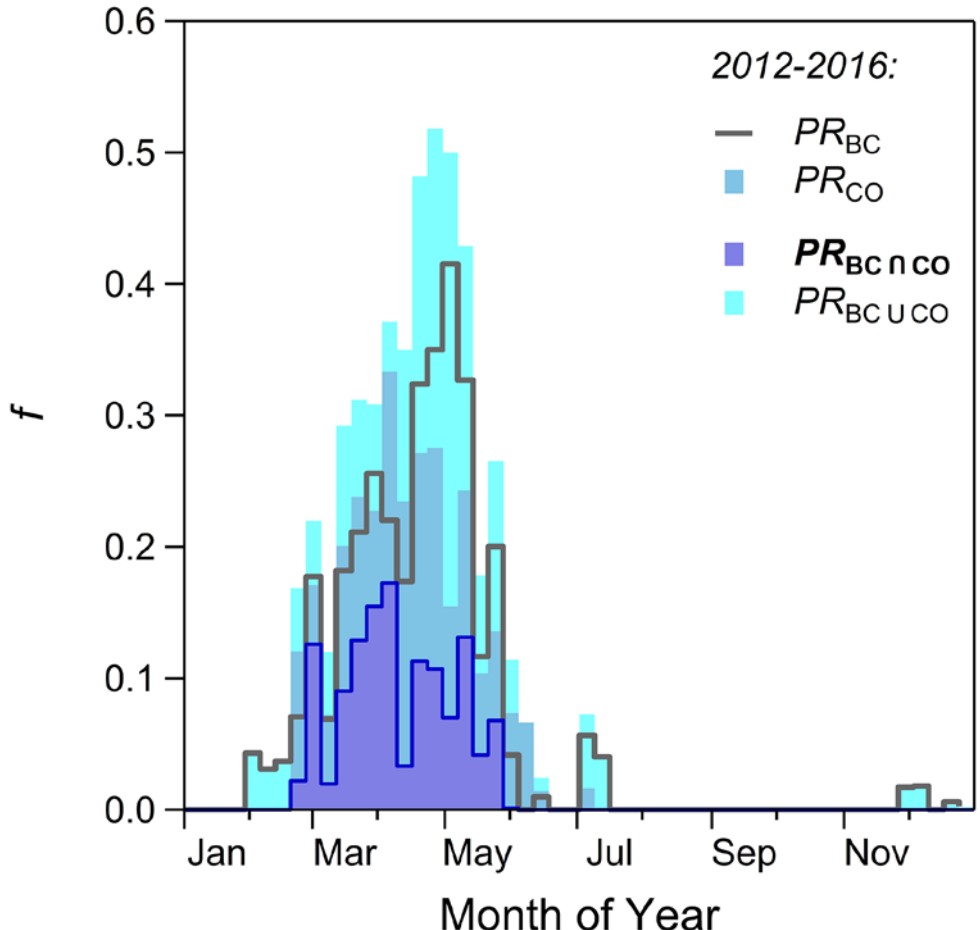

**Figure 1. Seasonality of frequency of occurrence, *f*, of empirically pristine rain forest (*PR*) aerosol conditions at ATTO by means of *PR_BC*, *PR_CO*, *PR_{BC ∩ CO}*, and *PR_{BC ∪ CO}* filters (see Sect. 2.7). *PR_{BC ∩ CO}* as most robust case is used as main *PR* filter throughout this work. Data is shown as weekly averages for time period from Mar 2012 until June 2016. Analogous representation for year 2014 as CCN focal period of this study is shown in Fig. S4.**

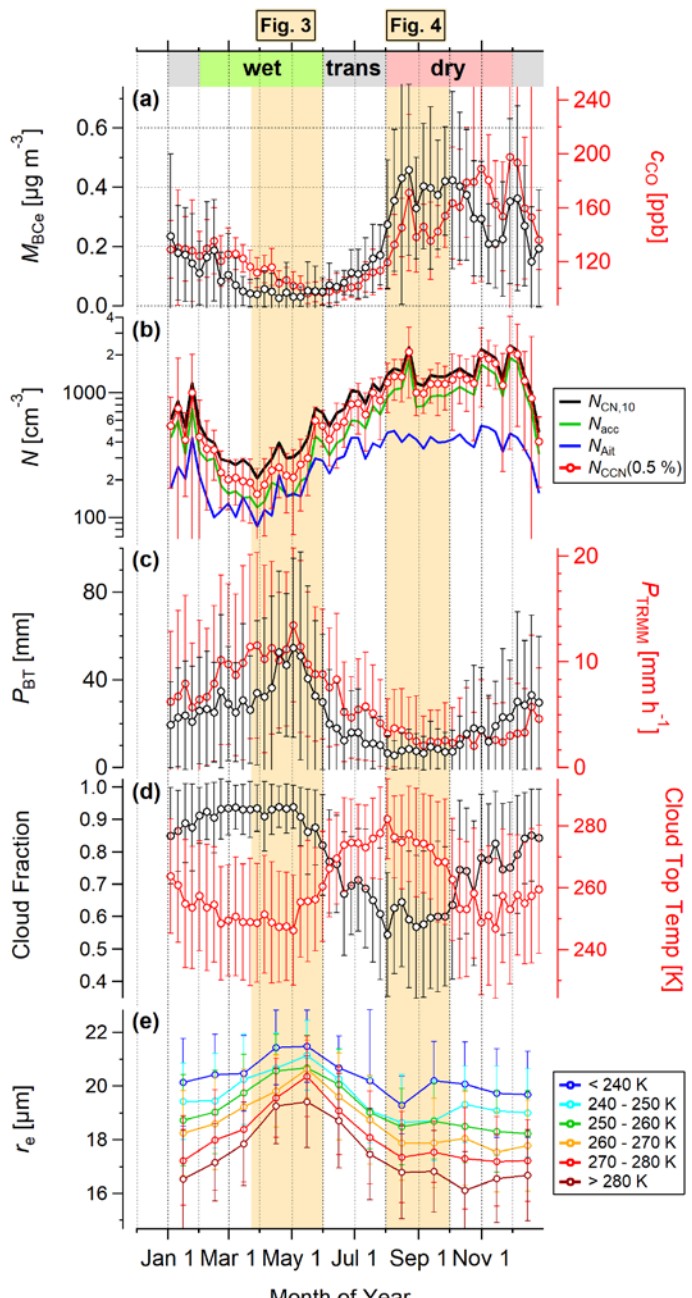

**Figure 2. Seasonal cycle of selected trace gas, aerosol, and cloud microphysical parameters. (a) Pollution tracers, $M_{BСe}$ and $c_{CO}$. (b) Total aerosol number concentration, $N_{CN,10}$. (c) Precipitation products, $P_{BT}$, representing cumulative precipitation along BT tracks, and, $P_{TRMM}$, representing TRMM-derived precipitation within the $ROI_{ATTO}$ as defined in Fig. S1. (d) Satellite-derived cloud fraction and cloud top temperature within the $ROI_{ATTO}$. (e). Satellite-derived cloud droplet effective radius, $r_e$, within the $ROI_{ATTO}$. For a detailed characterization of the land use and recent land use change in the ATTO footprint, including the $ROI_{ATTO}$, we refer to C. Pöhlker et al. (2018). Data in (a) to (d) is shown as weekly averages. Data in (e) is shown as monthly averages. Error bars represent one standard deviation. Vertical orange shading represents 2-month time frames of representative clean and polluted conditions as shown in detail in Fig. 3 and 4.**

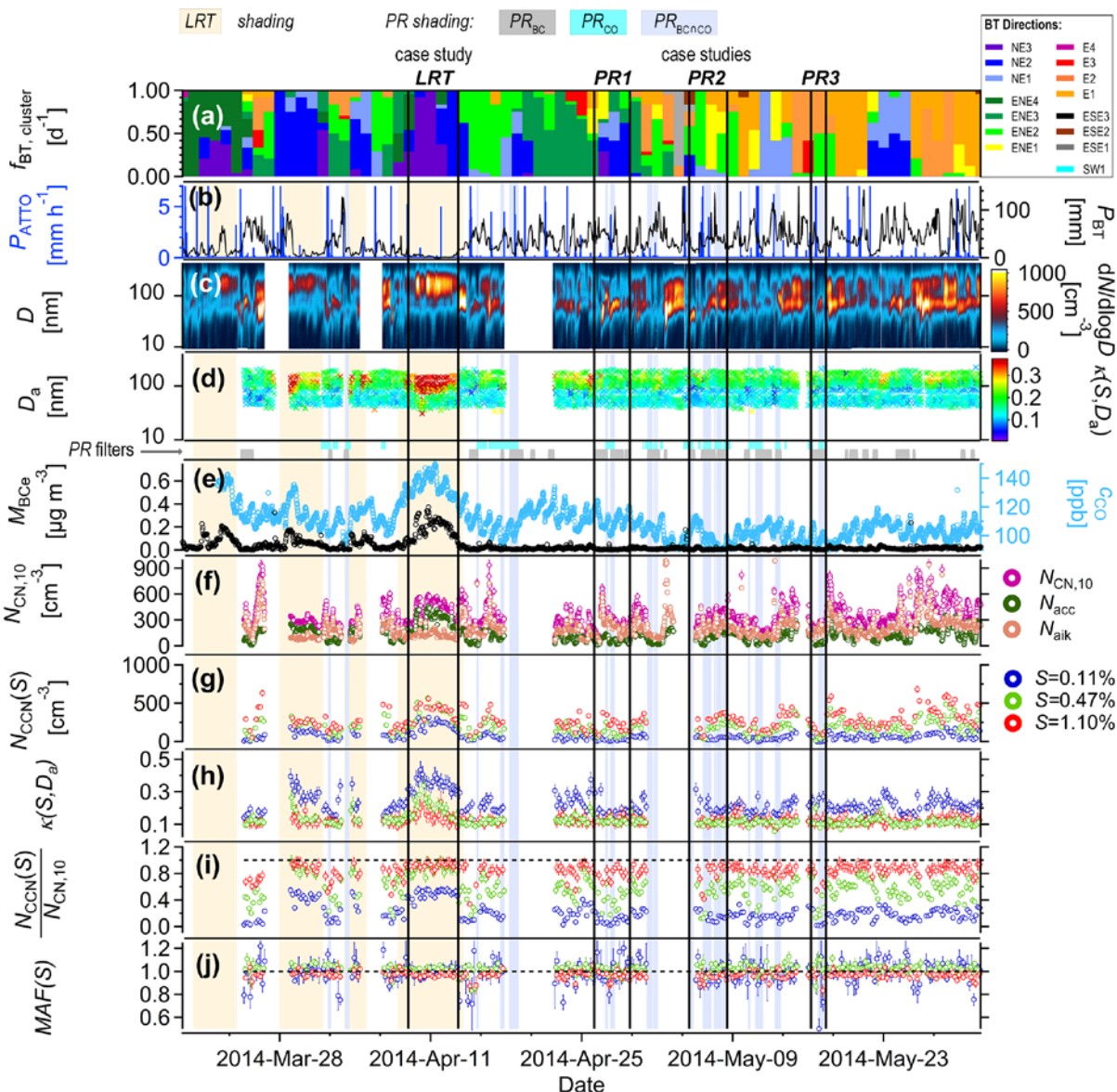

**Figure 3. Overview plot illustrating selected meteorological, trace gas, aerosol, and CCN time series for representative wet season conditions in the central Amazon. The shown time period from 23 Mar to 31 May covers a comparatively clean and extended time frame throughout the entire CCN measurement period. Individual panels represent: (a) Daily frequency of occurrence of 15 different BT clusters, $f_{BT,cluster}$, with color code corresponding to Fig. S1. (b) Precipitation rate, $P_{ATTO}$, measured locally at ATTO and cumulative precipitation from BT analysis, $P_{BT}$, as measure for aerosol wet deposition. (c) SMPS-derived time series of number size distributions spanning nucleation, Aitken, and accumulation modes. (d) CCNC-derived time series of $\kappa(S,D_a)$ size distributions. (e) Concentrations of biomass burning tracers, CO mole fraction, $c_{CO}$, and $BC_e$ mass concentration, $M_{BCe}$. (f) Total number concentrations of the entire aerosol population, $N_{CN,10}$, Aitken mode particles, $N_{Ait}$, and accumulation mode particles, $N_{acc}$. (g) CCN concentrations, $N_{CCN}(S)$, for selected supersaturations $S$. (h) Hygroscopicity parameter, $\kappa(S,D_a)$, for selected $S$, (i) CCN efficiencies, $N_{CCN}(S)/N_{CN,10}$, for selected $S$. (j) Maximum activated fraction, $MAF(S)$, for selected $S$. All CCN data is provided at the original time resolution of about 4.5 h. Light blue vertical shadings represent empirically pristine rain forest (PR) aerosol conditions according to the $PR_{BC \cap CO}$ filter as defined in Sect. 2.7. Vertical lines highlight PR1, PR2, and PR3 periods selected for detailed analysis in Sect. 3.4. Light orange shading represents LRT episodes according to Moran-Zuloaga et al. (2017). Vertical lines highlight LRT case study for detailed analysis in Sect. 3.5.**

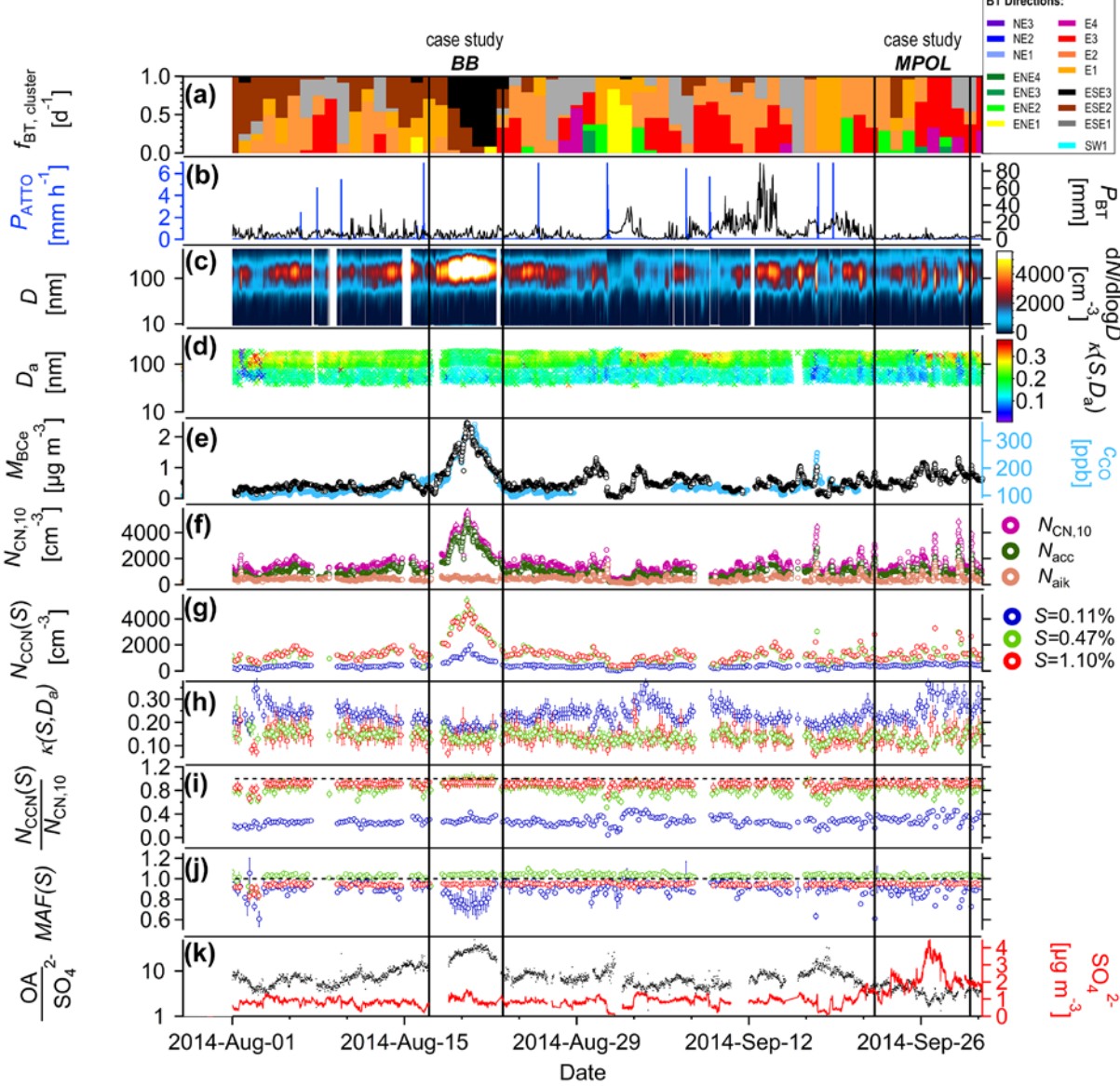

**Figure 4. Overview plot illustrating selected meteorological, trace gas, aerosol, and CCN time series for representative dry season conditions in the central Amazon. The shown time period from 01 Aug to 31 Sep covers the most polluted time frame throughout the entire CCN measurement period. Individual panels represent: (a) Daily frequency of occurrence of 15 different BT clusters, $f_{BT,cluster}$, with color code corresponding to Fig. S1. (b) Precipitation rate, $P_{ATTO}$, measured locally at ATTO and cumulative precipitation from BT analysis, $P_{BT}$, as measure for aerosol wet deposition. (c) SMPS-derived time series of number size distributions spanning nucleation, Aitken, and accumulation modes. (d) CCNC-derived time series of $\kappa(S,D_a)$ size distributions. (e) Concentrations of biomass burning tracers, CO mole fraction, $c_{CO}$, and $BC_e$ mass concentration, $M_{BCe}$. (f) Total number concentrations of the entire aerosol population, $N_{CN,10}$, Aitken mode particles, $N_{Ait}$, and accumulation mode particles, $N_{acc}$. (g) CCN concentrations, $N_{CCN}(S)$, for selected supersaturations $S$. (h) Hygroscopicity parameter, $\kappa(S,D_a)$, for selected $S$. (i) CCN efficiencies, $N_{CCN}(S)/N_{CN,10}$, for selected $S$. (j) Maximum activated fraction, $MAF(S)$, for selected $S$. (k) ACSM-derived sulfate mass concentration, $M_{sulfate}$, and organic-to-sulfate ratio, $OA/SO_4^{2-}$. All CCN data is provided at the original time resolution of about 4.5 h. Vertical lines highlight *BB* case study on biomass burning conditions for detailed analysis in Sect. 3.6 and *MPOL* case study on mixed pollution conditions for detailed analysis in Sect. 3.7.**

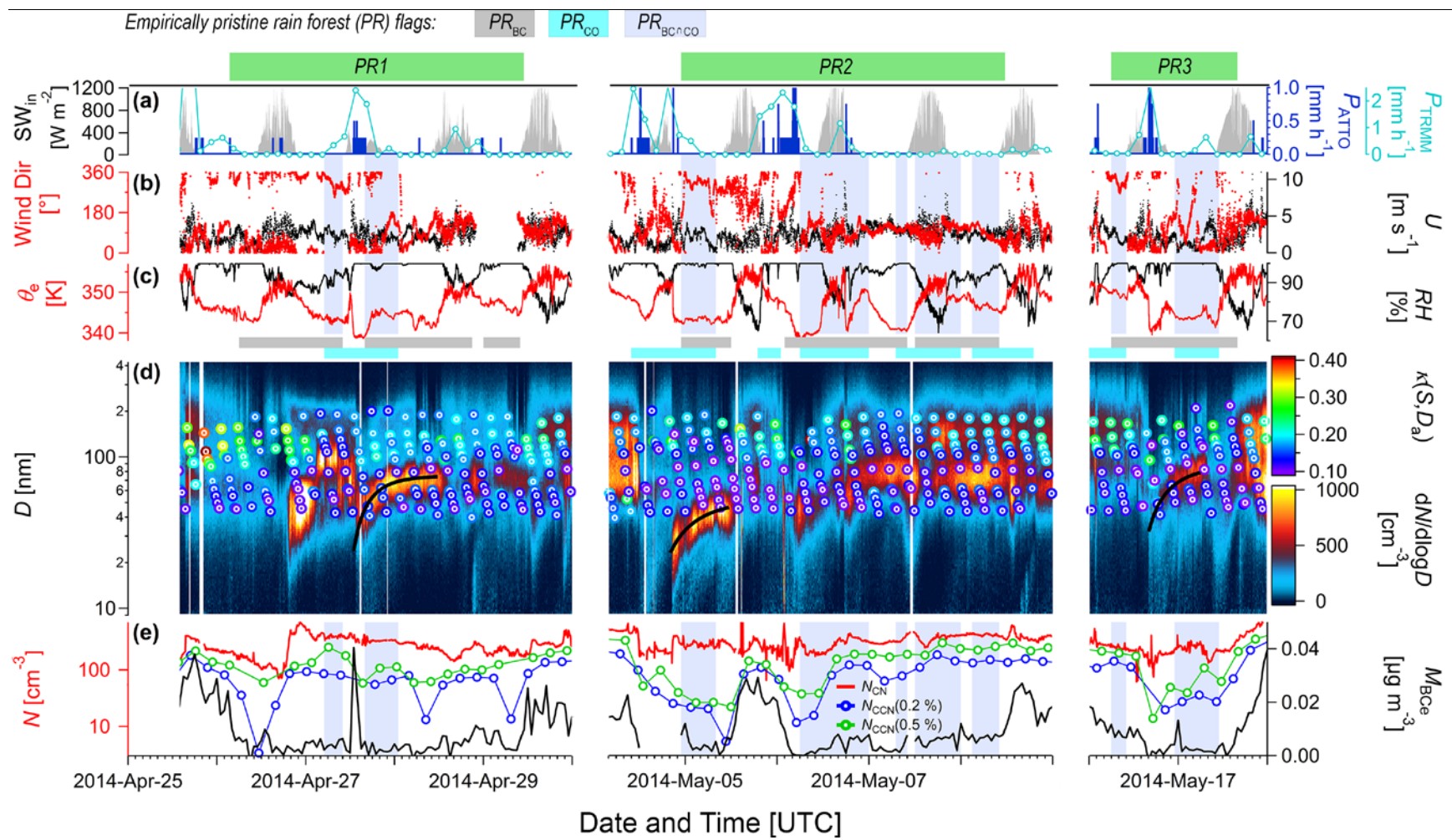

**Figure 5. Selected meteorological, aerosol, and CCN time series from ATTO measurements for the empirically pristine rain forest (*PR*) case study periods *PR1*, *PR2*, and *PR3* (see Fig. 3). (a) Incoming short-wave radiation, SW$_{in}$, precipitation rates from TRMM satellite mission, $P_{TRMM}$, and *in situ* measurements at ATTO, $P_{ATTO}$. (b) Wind direction and wind speed, *U*, at ATTO. (c) Equivalent potential temperature, $\theta_e$, and relative humidity, RH, at ATTO, (d) Overlay of two data layers showing aerosol number size distribution contour plot, d*N*/dlog*D*, as well as color coded markers, representing time series of $\kappa(S,D_a)$ size distributions, and (e) CCN concentrations, $N_{CCN}(S)$, for two selected *S* levels, total aerosol number concentration, $N_{CN,10}$, and BC$_e$ mass concentration, $M_{BCe}$. Vertical shadings represent *PR*$_{BC \cap CO}$ periods according to definition in Sect. 2.7.**

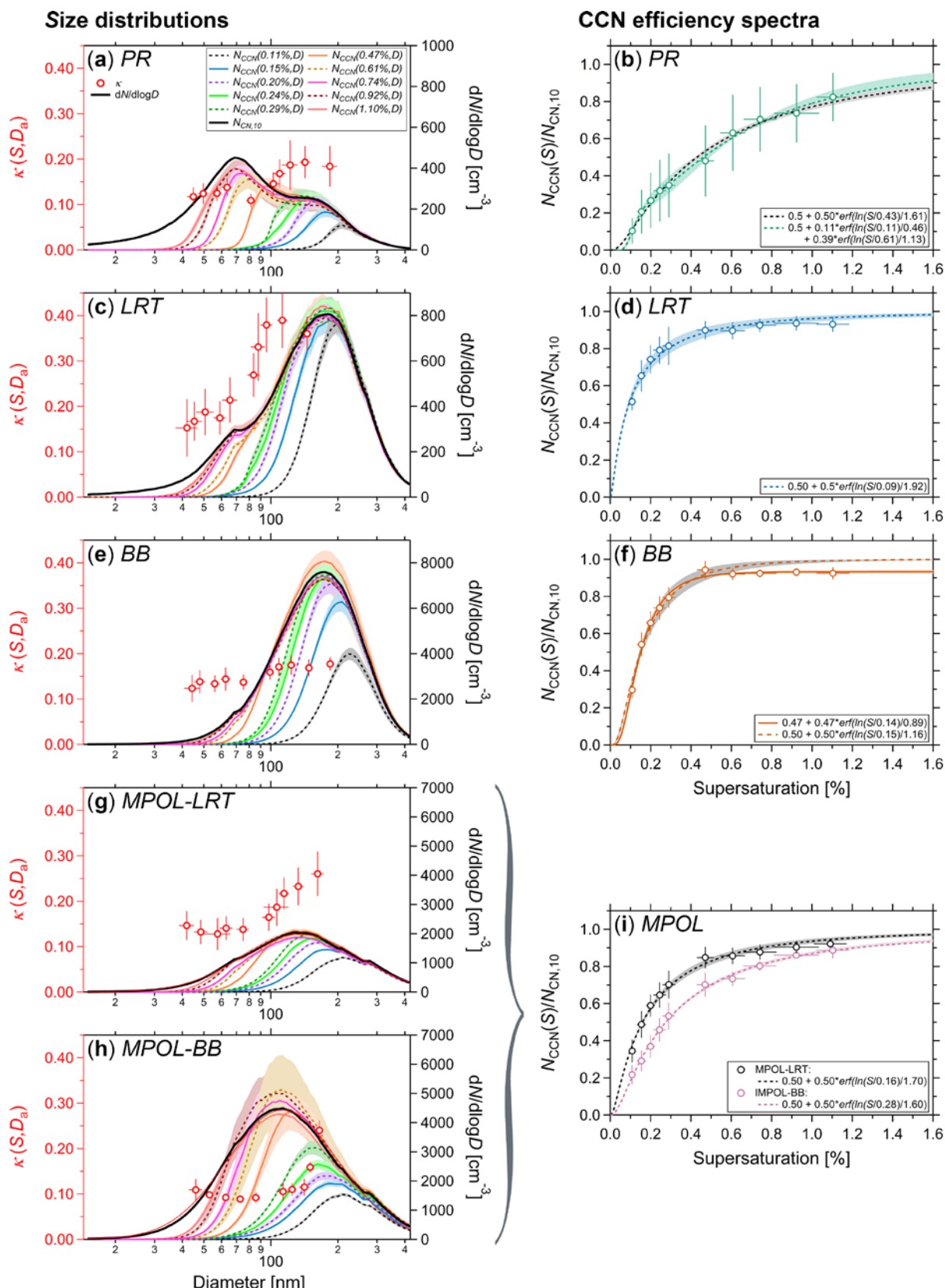

**Figure 6. Overview of case study conditions for empirically pristine rain forest (*PR*), long-range transport of African dust (*LRT*), biomass burning (*BB*), and mixed pollution of African and Amazonian sources (*MPOL*) showing: Size dependence of hygroscopicity parameter, $\kappa(S,D_a)$, number size distributions of total aerosol particles, $N_{CN}(D)$, and number**

size distributions of cloud condensation nuclei, $N_{CCN}(S,D)$, at all 10 $S$ levels ($S = 0.11$-$1.10$ %) (left side) and CCN efficiency spectra with erf fits (right side). For the size distributions (left): Values of $\kappa(S,D_a)$ for every $S$ level are plotted against their corresponding midpoint activation diameter $D_a(S)$. For $\kappa(S,D_a)$, the error bars represent one std. For $D_a(S)$, the experimentally derived error is shown. The standard errors of the number size distributions – $N_{CN}(D)$ and $N_{CCN}(S,D)$ – are indicated as shading of the individual lines. For the CCN efficiency spectra (right): $N_{CN,10}$ was chosen as reference concentration. The experimental data was fitted with single- or double-erf fits (dashed lines with shading as uncertainty of the fits). The error bars at the markers represent the measurement error in $S$ and one std in $N_{CCN}(S)/N_{CN,10}$ dimension. The shading represents the uncertainty of the fits. An overview of the erf fits from all case study conditions and seasonal averages can be found in Fig. 11. The parameters of all erf fits are summarized in Table 4. Data for *PR* conditions in (a) and (b) represent averages of all $PR_{BC \cap co}$ episodes during the entire CCN measurement period as defined in Sect. 2.7. For the *PR* CCN efficiency spectrum, the double-erf fit is the better representation, although the single-erf fit also works as a good approximation. Data for *LRT* conditions, shown in (c) and (d), represent the *LRT3* period as shown in Fig. 3. A single-erf fit describes the experimental data accurately. Data for *BB* conditions, shown in (c) and (d), represent time period in Aug 2014 as shown in Fig. 4. For the *BB* case, the experimental data has been fitted with a single-erf fit with two modifications: (i) The 'default' fit with predefined variable $a_1 = 1$ as utilized for all other case studies tends to overestimate $N_{CCN}(S)/N_{CN,10}$ at high $S$. (ii) A corresponding fit with a free variable $a_1$ describes the experimental data more accurately. Data for *MPOL* conditions, were separated into a *MPOL-LRT* case (g), presenting sulfate-rich African aerosols and a *MPOL-BB* case (h), representing plumes from close-by fires. *MPOL* CCN efficiency spectra are combined in (i) including single-erf fits. All fit parameters of the erf fits shown here are summarized in Table 4. The colors of the CCN efficiency spectra were chosen according to Wong (2011).

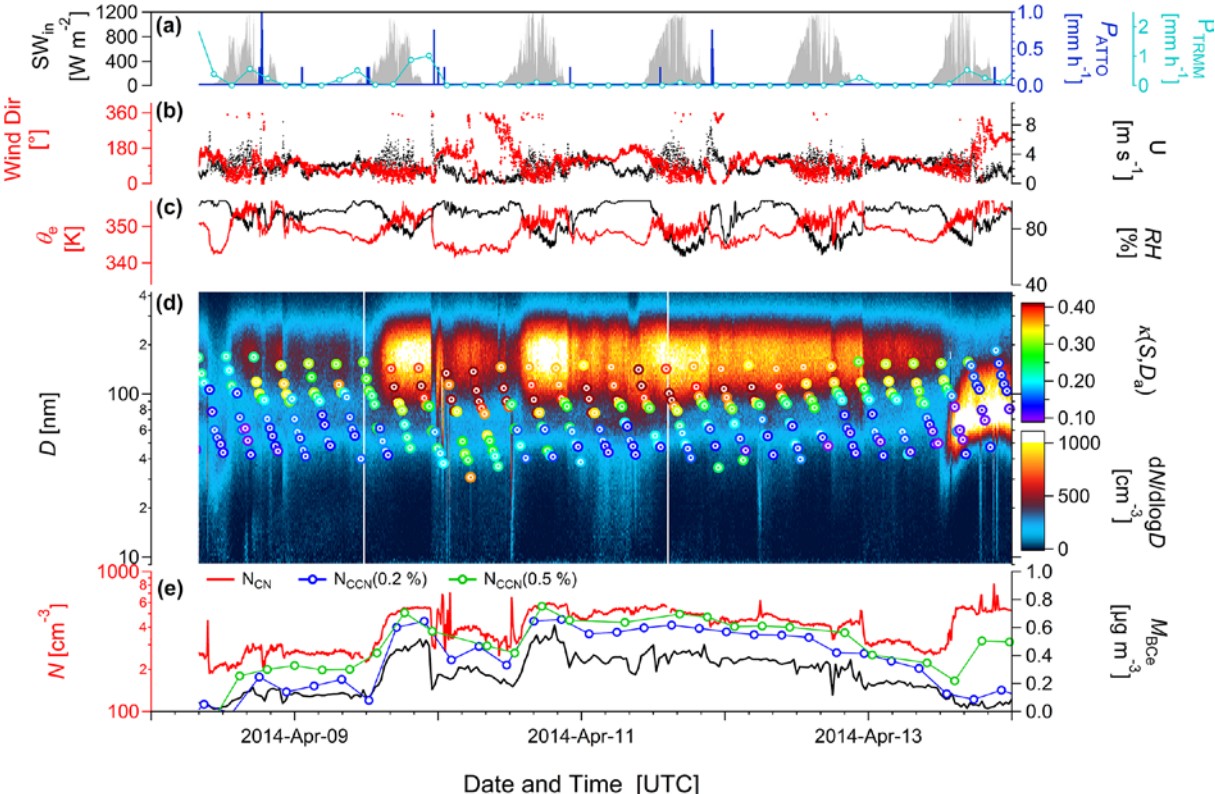

**Figure 7. Selected meteorological, aerosol, and CCN time series from ATTO measurements for the long-range transport study period, *LRT* (see Fig. 3). (a) Incoming short-wave radiation, $SW_{in}$, precipitation rates from TRMM satellite mission, $P_{TRMM}$, and *in situ* measurements at ATTO, $P_{ATTO}$. (b) Wind direction and wind speed, *U*, at ATTO. (c) Equivalent potential temperature, $\theta_e$, and relative humidity, RH, at ATTO, (d) Overlay of two data layers showing aerosol number size distribution contour plot, d$N$/dlog$D$, as well as color coded markers, representing time series of $\kappa(S,D_a)$ size distributions, and (e) CCN concentrations, $N_{CCN}(S)$, for two selected *S* levels, total aerosol number concentration, $N_{CN,10}$, and $BC_e$ mass concentration, $M_{BCe}$.**

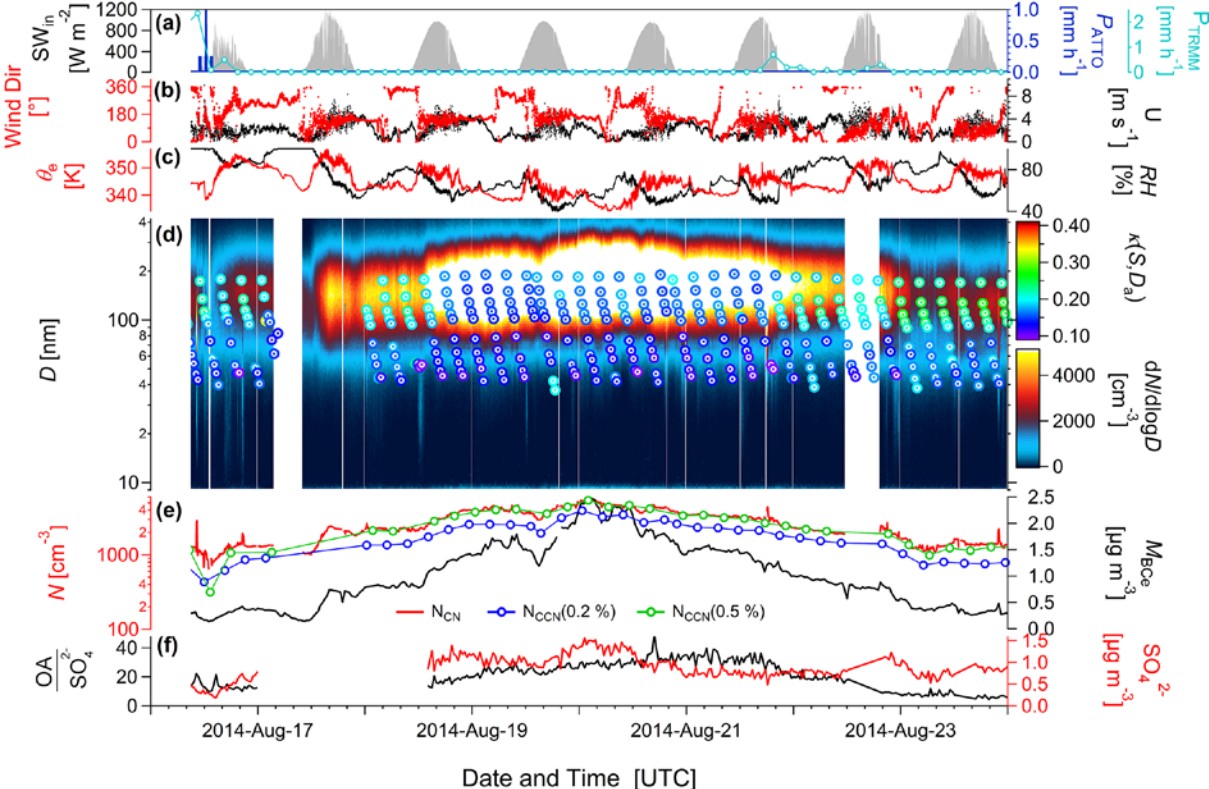

**Figure 8. Selected meteorological, aerosol, and CCN time series from ATTO measurements for the biomass burning case study period, *BB* (see Fig. 4). (a) Incoming short-wave radiation, $SW_{in}$, precipitation rates from TRMM satellite mission, $P_{TRMM}$, and *in situ* measurements at the ATTO site, $P_{ATTO}$. (b) Wind direction and wind speed, $U$, at ATTO. (c) Equivalent potential temperature, $\theta_e$, and relative humidity, RH, at ATTO, (d) Overlay of two data layers showing aerosol number size distribution contour plot, d$N$/dlog$D$, as well as color coded markers, representing time series of $\kappa(S,D_a)$ size distributions, and (e) CCN concentrations, $N_{CCN}(S)$, for two selected $S$ levels, total aerosol number concentration, $N_{CN,10}$, and $BC_e$ mass concentration, $M_{BCe}$. (f) ACSM-derived sulfate mass concentrations and organic-to-sulfate mass ratio.**

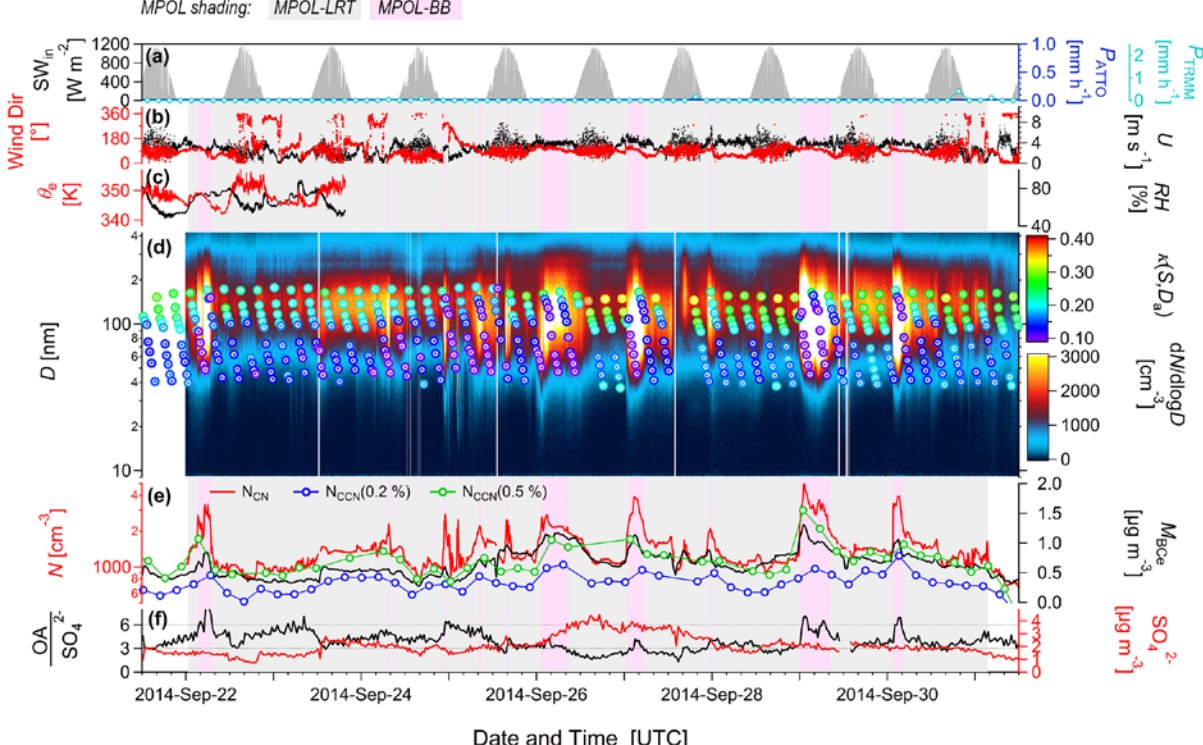

**Figure 9. Selected meteorological, aerosol, and CCN time series from ground-based ATTO site measurements for the mixed pollution case study period,** *MPOL* **(see Fig. 4). (a) Incoming short-wave radiation, $SW_{in}$, precipitation rates from TRMM satellite mission, $P_{TRMM}$, and** *in situ* **measurements at the ATTO site, $P_{ATTO}$. (b) Wind direction and wind speed, $U$, at ATTO. (c) Equivalent potential temperature, $\theta_e$, and relative humidity, RH, at ATTO, (d) Overlay of two data layers showing aerosol number size distribution contour plot, $dN/d\log D$, as well as color coded markers, representing time series of $\kappa(S,D_a)$ size distributions, and (e) CCN concentrations, $N_{CCN}(S)$, for two selected $S$ levels, total aerosol number concentration, $N_{CN,10}$, and $BC_e$ mass concentration, $M_{BCe}$. (f) ACSM-derived sulfate mass concentrations and organic-to-sulfate mass ratio. The vertical shading highlights episodes under the influence of local/regional fires (***MPOL-BB***) vs. periods that are dominated by long-range transport of African aerosols (***MPOL-LRT***).**

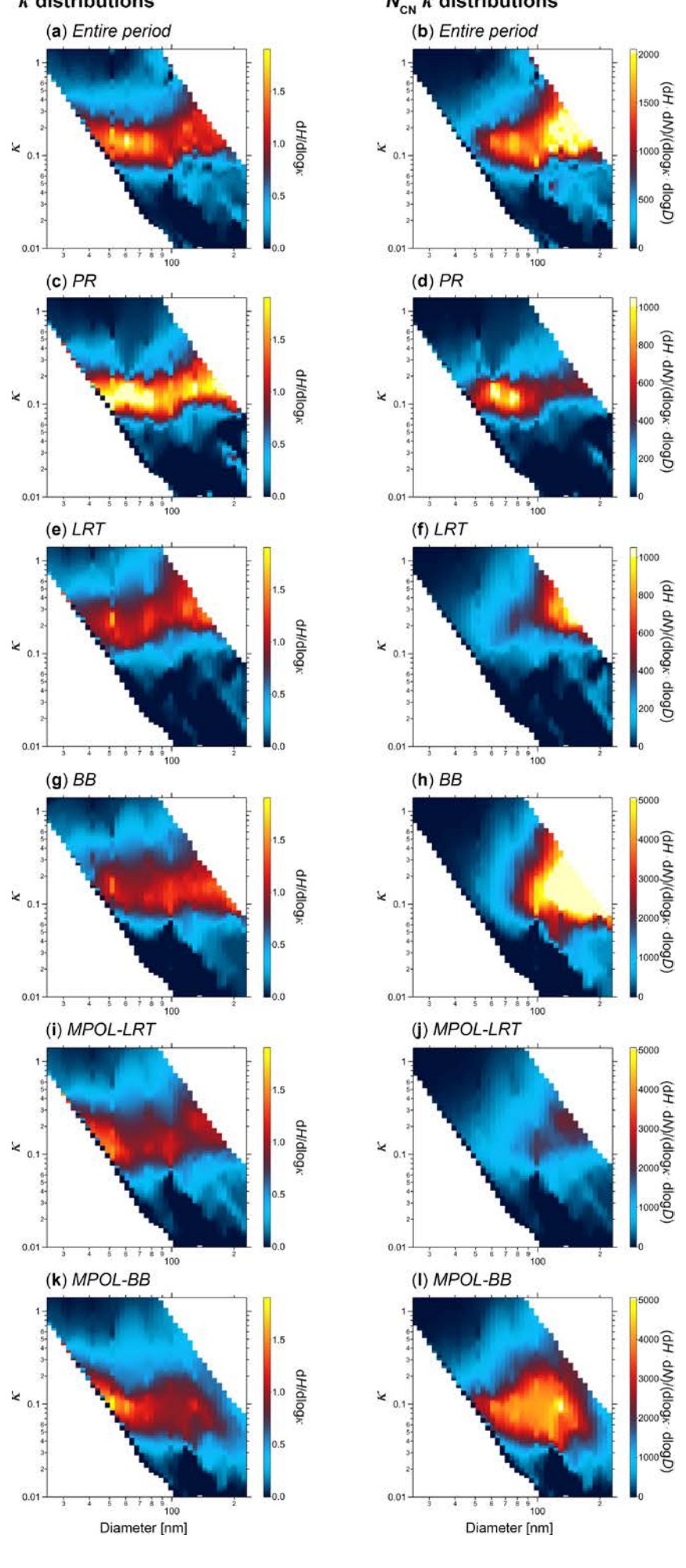

**Figure 10. Average probability distribution of particle hygroscopicity, d$H$/dlog$\kappa$, on the left side and the same quantity weighted by the particle number size distribution, (d$H$·d$N$)/(dlog$\kappa$·dlog$D$), on the right side, plotted over the effective hygroscopicity parameter, $\kappa$, and dry particle diameter, $D$, for (a and b) the entire measurement period as well as (c and d) *PR*, (e and f) *LRT*, (g and h) *BB*, (i and j) *MPOL-LRT*, and (k and l) *MPOL-BB* conditions. The particle size distributions used for the weighting are shown in Fig. 6.**

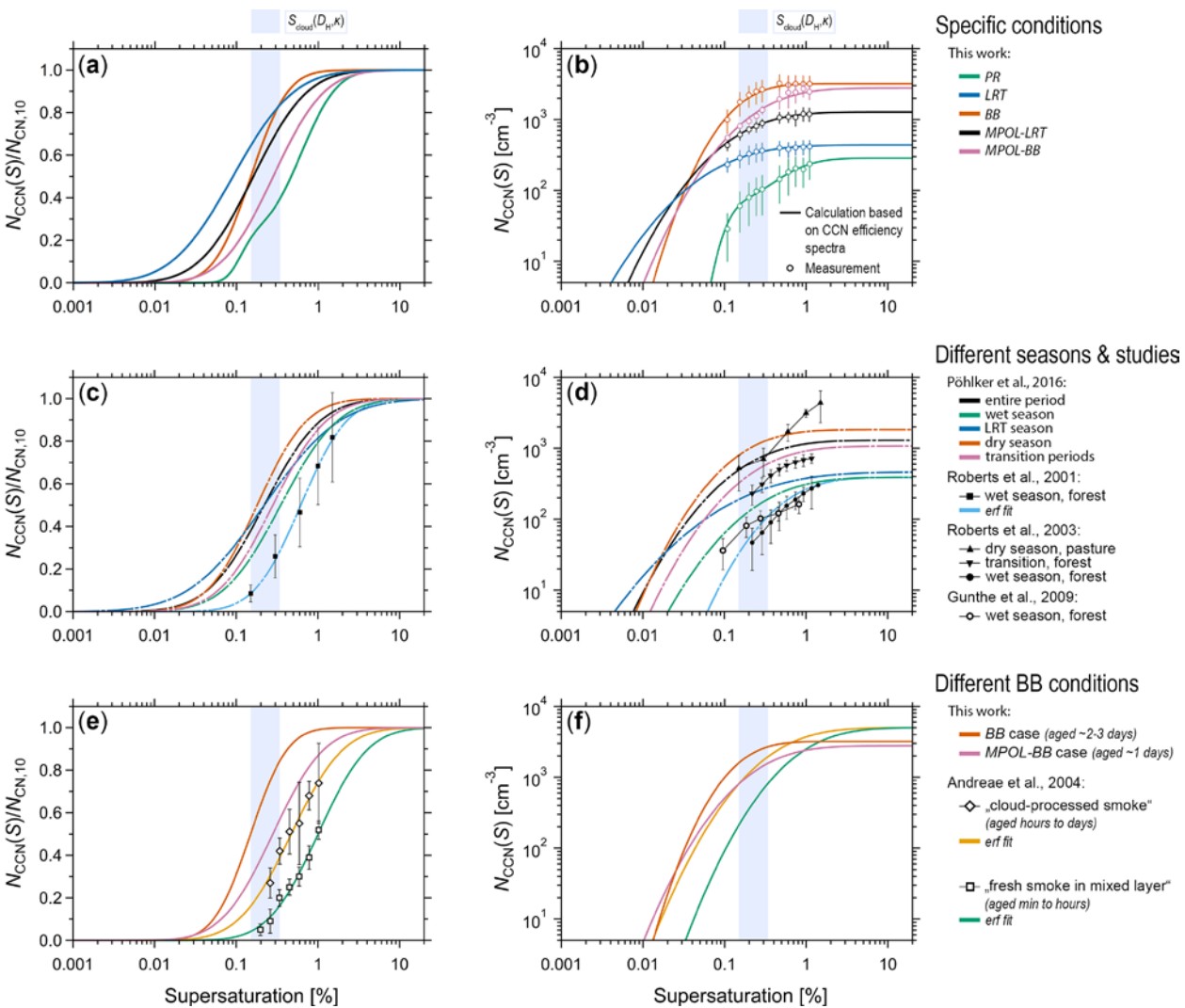

**Figure 11. Overview and comparison of normalized CCN efficiency spectra and number-concentration-based CCN spectra for characteristic conditions and seasons. (a and b) *PR*, *LRT*, *BB*, *MPOL-LRT*, and *MPOL-BB* conditions as defined in this paper. (c and d) Seasonally averaged spectra from the companion part 1 paper (M. Pöhlker et al., 2016), CCN efficiency spectrum from Roberts et al. (2001) as well as CCN spectra from Roberts et al. (2003) and Gunthe et al. (2009). (e and f) Conditions representing different aging states of biomass burning aerosols based on *BB* and *MPOL-BB* conditions from present work and data from a previous study by Andreae et al. (2004). For clarity the erf fit with the pre-defined variable $a_1 = 1$ was used for the *BB* case (compare Fig. 6f). CCN spectra in (b) were obtained from multiplication of CCN efficiency spectra in (a) with corresponding average aerosol number concentrations in Table 4. Markers represent measured average CCN concentrations for case study periods. Error bars at markers represent one std. Good agreement of CCN spectra and markers underlines reliability of CCN efficiency spectra in representation of CCN population. Blue vertical shading represent estimated peak supersaturations at cloud base, $S_{cloud}(D_H, \kappa)$, in the ATTO region according to M. Pöhlker et al. (2016) and present study. The colors were chosen according to Wong (2011).**