# Peer review of "Long-term observations of cloud condensation nuclei over the Amazon rain forest – Part 2: Variability and characteristics of biomass burning, long-range transport, and pristine rain forest aerosols"

_Atmospheric Chemistry and Physics, 2017_

## Referee Comment (RC1) · Anonymous Referee #1 · 1 Dec 2017

General comments: The paper "Long-term observations of cloud condensation nuclei in the Amazon rain forest – Part 2: Variability and characteristic differences under near-pristine, biomass burning, and long-range transport conditions" by M. Pohlker, et al. 2017 provide comprehensive surface CCN measurement dataset in the Amazon rain forest. The authors present very useful observations on CCN activation properties and the results are consistent with the previous studies. However, the paper did not clearly address the main effects on CCN properties for different conditions, such as

distinguishing the effect of chemical composition from size distribution or mixing state of aerosol.

Specific comments: P4, section 1.2: Please provide additional information about how four cases are representative the typical CCN variability in the Amazon basin. For example, how many days are dominated by NP or BB condition?

P6, section 2.3, The definition of the near-pristine periods is a little bit weak. It seemed that it was only based on BC. Will other urban pollution tracer be considered?

P7, section 2.4: ATTO tower is 325 m tall. What is the uncertainty we expect from the BT analysis start height of 1000 m?

P13, line 5-10, What is the percentage of stable northeasterly wind direction for the periods in Figure 5a?

Figure 5, 7 and 8, If possible, please do not overlap k(S, Da) with the size distribution plot. It is very hard to read the color map in k(S,Da).

P15, line 5-10, The results here are not well supported. Andreae et al. 2017 showed that the UT particles consist predominately of organic material for aerosol size larger than 90 nm. For aerosol less than 60 nm, AMS had a hard time to determine chemical composition with good sensitivity.

P16, section 3.5, The Saharan dust confirmed by EDX are larger than 1 micron. The CCN discussed in this section are in much smaller size range. It is confusing to classify the LRT influence as Saharan dust influence.

P18, line 23-24, from Table 3, except LRT case, the rest of cases all have reasonably good agreement between kp and k(0.11%). It is very stretching to state the LRT case is in a good agreement.

P19, section 31, authors said that "…correspond to a clear drop in aerosol hygroscopicity…". Please clarify the "drop", compared to what cases?

[Figure]

P22, line 37, what is the OA/SO4 ratio for LRT pollution periods in Figure 7? Are they consistent with this case, around 3?

Figure 10, the different shape of the CCN efficiency spectra may related to the mixing state of aerosols for each case. It will be interesting to include that discussion.

[Figure]

---

## Referee Comment (RC2) · Anonymous Referee #2 · 4 Dec 2017

Review of "Long-term observations of cloud condensation nuclei in the Amazon rain forest – Part 2: Variability and characteristic differences under near-pristine, biomass burning, and long-range transport conditions" by M. Pöhlker et al.

This study describes in detail the characteristics of air masses and CCN under a range of conditions that are frequently observed at the ATTO site. The authors distinguish between near-pristine, biomass burning, long-range transport and mixed-pollution con-

ditions. The study builds on a companion paper "Long-term observations of cloud condensation nuclei in the Amazon rain forest – Part 1: Aerosol size distribution, hygroscopicity, and new model parameterizations for CCN prediction" as well as on a number of other studies that have been conducted on aerosol, trace gases and air mass characteristics at ATTO. Many detailed results are presented that help the understanding of aerosol processes, and the main finding of the present work highlights the different sensitivities of CCN towards changes in various supersaturations ranges. While in clean conditions CCN are sensitive to the entire supersaturation range, in the various pollution cases sensitivity is enhanced towards the lower end. Overall, the study is of high quality and well written. While I have generally only minor comments, it is very important that the authors clarify the meaning of "near-pristine" before the work can be published.

General comments:

The meaning of "near-pristine" is not very well defined. In section 2.3 the definition of NP excludes periods with concentrations of EBC that surpass a certain threshold to identify "the cleanest aerosol conditions" (p. 6, l. 22). Clean aerosol conditions and pristine conditions are however not the same. "Pristine" hints towards a natural aerosol state. Before anthropogenic emissions natural aerosol conditions would have included LRT of Saharan dust and natural forest fire emissions occasionally. Hence, defining "near-pristine" as the absence of any other natural aerosols and only taking into account the rain forest generated particles is not representative of the variability of pristine conditions. Calling them "near"-pristine does not help, because this is with reference to the unavoidable anthropogenic influences. From how the "near-pristine" conditions are identified and described, I understand that they are meant to reflect unperturbed aerosol conditions that are dominated by the Amazon rain forest as particle and precursor source. It is also evident from page 12, l. 39 that the authors hope to provide data for the preindustrial-like Amazonian atmosphere with the near-pristine cases. In this context it is even more important to acknowledge that clean cases exclude other

conditions as mentioned above that were however present in the preindustrial time. For that reason, I urge the authors to rename their "near-pristine" classification into "clean" conditions as this reflects the actual definition much better, "near-pristine" is misleading. As more and more work is being done to extract preindustrial-like information in various environments it is important to be clear about the terminology.

I am also surprised that "near-pristine" periods are only defined via EBC. Why did the authors not include CO and ozone mixing ratios or back trajectories which are all at hand? If particles were removed through precipitation, EBC can be low, while CO is elevated and hence the air mass cannot be classified as "near-pristine" or rather clean, even though the focus is on aerosol particles.

The results of the paper would be even more informative if the authors provided the fraction in time throughout which the four different conditions are present. How often do clean conditions prevail? How often the mixed-polluted? It would be very informative if these simple statistics could be added.

The paper is well written, however there many qualifiers and redundancies in the text. In the technical comments section some instances are mentioned, but it would be helpful if the authors worked through the manuscript again to remove those. Additionally, the use of some terms is rather unusual. The term "pulses" should be replaced by "events". Also the precise meaning of "aerosol cycling" needs to be explained. Is cloud processing meant, or atmospheric processing? Please use the exact description. The word "trend" is frequently used in contexts, where no trends can be observed. The meaning is rather "variability, curve shape etc.". Below this is pointed out in detail.

Specific comments:

Please go through the references again. Often XY et al. yyyyb comes before "a".

p.2, l. 14: At this point it is not clear what the efficiency spectrum is, a brief explanation is required.

[Figure]

p.2, l. 27: the reference to volcanic emissions is confusing without any explanation, either provide one or do not mention volcanic influence here.

p.2, l. 30f: No D_ait is provided for the size distributions, but kappa_ait is given. Be consistent in providing all information.

p.2, l. 33: Why is the sensitivity of the CCN population towards changing supersaturation not mentioned for the mixed pollution cases?

p. 3, l. 36: "multi-month trends" are not trends but rather changes or variability across the time period, also l. 38, not diurnal "trends" rather "diurnal cycle"

p. 4, l. 1: Include info on what is the relevant size range?

p. 4, l. 7: please provide a reference for "previous studies"

p.4, l. 13f: Provide information on what the relevant supersaturation range is for the ATTO region. Regarding the introduction, I would have expected a short paragraph that introduces the relevant literature on the ATTO site on which this study builds. After reading the abstract one expects to learn at least more about air mass transport towards ATTO in the introduction. Please introduce a short paragraph on this.

Why do the authors invent a new abbreviation for equivalent black carbon? The standard is "EBC" (Petzold et al., 2013). Please use this one.

The equations are not numbered. There is no need to repeat Eq. 1 without ammonium. It suffices to describe the change in the text as it is trivial.

p. 5, l. 32: Generally the phrase is "sth. corresponds to" not "with". Please replace all cases throughout the manuscript.

In the methodology section, information on how the CCNC was operated is needed. Even though this is described in the Part 1 paper, readers should get the essential information here (e.g., monodisperse mode, duration of a scan, covered range of diameters).

[Figure]

p. 6, l. 10: What was done with a2? Was there an assumption or was it fitted?

p. 7, l. 6: Are the 1000 m above sea level or above ground? If a.s.l. how high above the measurement site?

p. 9, l. 31: Does this statement refer to the Amazon or is it a general statement?

There is no need for footnotes. They can all be included in the main text which is already quite detailed. It is not apparent why some details are moved to footnotes. Specifically on p. 10, delete the 2nd sentence in the footnote, because it is a repetition.

p. 11, first paragraph: What about the influence of more local anthropogenic emissions?

p. 16, l. 4: What is meant by "perception"? Please clarify.

p. 18, l. 19f: Is the fraction of long-range transported mineral dust and sea salt really large? Please include the values in the text: What is the fraction of supermicron dust and salt in mass or number concentration?

p. 20, l. 21: How can the ratios of 26 and 49 be consistent? That's a factor 2.

p. 21, l. 32: Define aerosol "cycling"

p. 22, l. 29: The African volcanic emission appear out of nowhere. Were they never observed before at ATTO? Why can the authors be so sure that they have observed volcanic emissions?

p. 24, l. 35: The here defined NP conditions are very selective because they describe only clean conditions. They are hence not an approximation of a preindustrial state, but represent only a potential fraction of the preindustrial atmosphere in that region.  

Technical comments:

p.2, l. 24: What do you mean by "CCN cycling in relation to aerosol-cloud interactions"?

p.3, l. 20: change to "Ndb ranges from few hundred... for clean to 1000 and ... for polluted conditions"

p. 3, l. 32: delete "of this endeavor"

p. 4, l. 3: D_a has not been defined yet.

p. 4, l. 10f: Delete the last sentence of the paragraph, this information is not really needed and the paper can benefit from being shortened.

p. 4, l. 7: Delete " As a particularly...concept". This is a qualifier that does not help the reader to understand the efficiency spectra better.

p.4, l. 29f: delete "particularly interesting"

p. 4, l. 38: What is an "efficient CCN prediction?" Is this method particularly fast or do you mean effective?

p. 5, l. 19: delete "predicted"

P. 9, l. 4: Spell out Mar

p. 9, l. 7f: The sentence beginning with "First, the..." is not a grammatically correct sentence.

p.9, l. 17: a reference is missing

p. 9, l. 28: replace "trend" by "behavior"

p. 9, l. 32: Shorten this paragraph: "Figure 1 provides... sections will focus on detailed aerosol and CCN characteristics in wet... year 2014." Delete the rest as is has been mentioned already.

p. 10, l. 28: delete "in the context of this study".

p. 11, l. 10: replace "trend" with "observations"

p. 11, l. 13 ff: delete this sentence, it has already been said.

p. 11, l. 31: replace "in more detail of" with "on"

p. 12, l. 1: preplace "pendant" with "counterpart"

p. 12, l. 13: Do you mean $N\_cn10$? There are a couple of other instances where "10" has been omitted.

p. 12, l. 28: delete "interesting" this qualifier is not needed.

p. 13, l. 16: What is meant by aerosol cycling here?

p. 14, l. 39: delete "mode"

p. 15, l. 14: replace "ranged" with "was", and its "contributions"

p. 15, l. 29: delete "solid"

p. 15, l. 35: write just "a tool"

p. 16, l. 34f: remove "as a particularly instructive example"

p. 17, l. 11: shouldn't the units of $SW\_in$ the figures be W per square meter?

p. 17, l. 24: A "LRT pulse" does not make sense, replace by "LRT event"

p. 17, l. 28: remove "exactly"

p. 18, l. 10: Do the authors refer to natural or anthropogenic fires, or both?

p. 19, l. 20: remove "fortunate"

p. 19, l. 23: remove "rather"

p. 20, l. 27: remove the sentence beginning with "It. . .". This has already been said.

p. 21, l. 28: Delete this sentence, redundancy.

p. 21, l. 31: delete "pulses"

p. 22, l. 24: delete "such as industrial emissions, . . .River", This has already been said.

p. 22, l. 33: delete "far"

p. 23, l. 34f: shorten to "CCN efficiency spectra serve as CCN signatures."

p. 23, l. 38: delete "trends" because trends are not discussed in this paper.

p. 23, l, 39: Why "could"? Can they or can they not?

p. 24, l. 1: it seems that $D\_H$ has not been introduced.

p. 24, l. 12f: replace "trends" with "curve shapes", and again with "shapes"

p. 24, l. 17: replace "pulses" with "events"

p. 24, l. 38: replace "pulses" with "events"

p. 25, l. 9: delete "very"

p. 25, l. 15f: "growth as well as to enhance the . . ." otherwise the sentence is grammatically not correct.

p. 25, l. 33: Be more specific regarding what is meant with cycling.

p. 26, l. 5 delete "instructive"

p. 26, l. 14: delete "very" and "highly"

p. 26, l. 22: "The aerosol particles are composed of. . ."

p. 26, l. 31: "on event-basis" instead of "pulse-wise"

p. 27, l. 11: "mixed" instead of "superimposed"

p. 27, l. 25: "behaviors" instead of "trends"

p. 27, l. 25 f: The sentence starting with "The array of . . .." is cryptic for saying that there exists large variability, rephrase.

Reference: Petzold, A., Ogren, J. A., Fiebig, M., Laj, P., Li, S. M., Baltensperger, U.,

Holzer-Popp, T., Kinne, S., Pappalardo, G., Sugimoto, N., Wehrli, C., Wiedensohler, A., and Zhang, X. Y.: Recommendations for reporting "black carbon" measurements, Atmos. Chem. Phys., 13, 8365-8379, 2013.

---

## Editor Comment (EC1) · G. Fisch (Editor) · 13 Dec 2017

Dear Dra. Pohlker

1) You have received comments/suggestions from 2 reviewers. They have raised very good points and I would like to encourage you to reflect and properly answer them. Once you upload the new version, please also send me a letter answering each comments made. best regards gfisch

---

## Author Comment (AC4) · 18 May 2018

**[1.3]**     Referee comment: P7, section 2.4: ATTO tower is 325 m tall. What is the uncertainty we expect from the BT analysis start height of 1000 m?

Author Response: We have conducted a sensitivity test and found that backward trajectory start heights at 200 m and 1000 m gave similar results. A corresponding statement can be found in Sect. 2.3:

> "A sensitivity test confirmed that starting heights of the BTs at 200 and 1000 m AGL gave similar results. Accordingly, the chosen start height at 1000 m appears to be a good representation of the origin of the boundary layer air masses at ATTO."

Details on the backward trajectory analysis can be found in C. Pöhlker et al. (2018), which has been submitted to ACP. This reference is cited several times throughout the text to clarify aspects of backward trajectory patterns and land use in the ATTO footprint region.

Editor comment: I have an additional doubt about the point below. Could you explain how winds at 200 and 1000 m showed the same pattern?! Which wind field are you using for these backtrajectories?!

Author Response: The HYSPLIT backward trajectories (BT) analysis has been based on meteorological input data from the global data assimilation system (GDAS1) as outlined in Sect. 2.3 in the present study. The GDAS1 data on a 1° by 1° grid is comparatively coarse. However, it sufficiently resolves the direction of air mass advection towards ATTO. The aerosol and CCN population at ATTO that is analyzed in detail in this work originates from regional and well long-range transport influences. In terms of regional influences, a study by Lammel et al. (2003) reported characteristic formation times of secondary aerosols of about 48-72 h as well as the fact that coarse mode particles "were derived from emissions < 36 h back". Accordingly, 3-day HYSPLIT BTs represent an appropriate choice to map the spatiotemporal variability of the effectively overpassed regions and its land cover with the potential aerosol sources.

   Figure R1 shows BT ensembles for the starting heights of 200 vs. 1000 m. Note that the direction of advection of the BTs is almost exclusively directed towards easterly directions. Moreover, the BT ensembles of both starting heights shows similar spatiotemporal patterns and, thus, cover rather similar 'footprint regions'. In the present study, we have used BTs to characterize the overall directions of air mass advection. For this analysis, BTs at starting heights of 1000 m were used. Figure R1 underlines that for this purpose the choice of 200 vs. 1000 m as starting height makes no substantial difference.

**References**

Lammel, G., Bruggermann, E., Gnauk, T., Müller, K., Neususs, C., and Rohrl, A.: A new method to study aerosol source contributions along the tracks of air parcels and its application to the near-ground level aerosol chemical composition in central Europe, Journal of Aerosol Science, 34, 1-25, 10.1016/s0021-8502(02)00134-9, 2003.

Pöhlker, C., Walter, D., Paulsen, H., Könemann, T., Rodríguez-Caballero, E., Moran-Zuloaga, D., Brito, J., Carbone, S., Degrendele, C., Després, V. R., Ditas, F., Holanda, B. A., Kaiser, J. W., Lammel, G., Lavrič, J. V., Jing, M., Pickersgill, D., Pöhlker, M. L., Praß, M., Ruckteschler, N., Saturno, J., Sörgel, M., Wang, Q., Weber, B., Wolff, S., Artaxo, P., Pöschl, U., and Andreae, M. O.: Land cover and its transformation in the backward trajectory footprint region of the Amazon Tall Tower Observatory, Atmos. Chem. Phys. Diss, submitted, 2018.

[Figure]

**Figure R1.** HYSPLIT backward trajectory (BT) ensembles representing ATTO-relevant air mass residence time maps (**a** and **b**). The BT ensembles comprise all 8760 individual 3-day BTs, spanning the time period relevant for the present study (i.e., hourly BTs from 01 Mar 2014 until 28 Feb 2015). The BT analysis was conducted for the starting heights 200 m and 1000 m above ground level. Contour lines (in green) were added to the image plots in **a** and **b**. These contour lines representing the 200 and 1000 m cases were combined in **c** for direct comparison.

---

## Author Response (AR1)

**Response to Gilberto Fisch (editor)**

Manuscript: **M. L. Pöhlker et al., Long-term observations of cloud condensation nuclei over the Amazon rain forest – Part 2: Variability and characteristics of biomass burning, long-range transport, and pristine rain forest aerosols**, ACP-2017-847)

Editor comment:

Dear Dra. Pohlker

You have received comments/suggestions from 2 reviewers. They have raised very good points and I would like to encourage you to reflect and properly answer them. Once you upload the new version, please also send me a letter answering each comments made.

best regards gfisch

Author Response:

Dear Prof. Fisch,

we fully agree that both reviewers brought up important aspects and we highly appreciate this constructive criticism. In fact, the reviewers' comments encouraged us to revise the manuscript profoundly, which clearly improved the quality of the manuscript. Furthermore, the revision extended the scope and relevance of the study. All comments brought up by the reviewers have been addressed point-by-point in our responses.

In the course of the revision, we have implemented two new sections (Sect. 2.7 and 3.8 in the revised version), one new main text figure (Fig. 10 in the revised version), and seven new supplement figures. We further revised and improved the entire text as well as all figures and tables. The changes in the manuscript are marked with blue font.

We would like to thank you at this point for your constructive contributions and pleasant help with the review process.

Best regards,

Mira Pöhlker and coauthors

**Response to the referee #1**

Manuscript: **M. L. Pöhlker et al., Long-term observations of cloud condensation nuclei over the Amazon rain forest – Part 2: Variability and characteristics of biomass burning, long-range transport, and pristine rain forest aerosols**, ACP-2017-847)

We appreciate the comments by Referee #1, which have been considered carefully and helped to improve the quality of our manuscript. The referee's comments and our responses are outlined in detail below:

**[1.1]**    Referee comment: P4, section 1.2: Please provide additional information about how four cases are representative the typical CCN variability in the Amazon basin. For example, how many days are dominated by NP or BB condition?

Author Response: We agree with the referee. Please refer to our response to comment **[2.3]** of referee #2, where this aspect is discussed in detail.

**[1.2]**    Referee comment: P6, section 2.3, The definition of the near-pristine periods is a little bit weak. It seemed that it was only based on BC. Will other urban pollution tracer be considered?

Author Response: The definition and analysis of near-pristine and pristine states as ATTO has been revised very carefully. Please refer to our detailed response to comment **[2.2]** by referee #2.

**[1.3]**    Referee comment: P7, section 2.4: ATTO tower is 325 m tall. What is the uncertainty we expect from the BT analysis start height of 1000 m?

Author Response: We have conducted a sensitivity test and found that backward trajectory start heights at 200 m and 1000 m gave similar results. A corresponding statement can be found in Sect. 2.3:

> "A sensitivity test confirmed that starting heights of the BTs at 200 and 1000 m AGL gave similar results. Accordingly, the chosen start height at 1000 m appears to be a good representation of the origin of the boundary layer air masses at ATTO."

Details on the backward trajectory analysis can be found in C. Pöhlker et al. (2018), which has been submitted to ACP. This reference is cited several times throughout the text to clarify aspects of backward trajectory patterns and land use in the ATTO footprint region.

**[1.4]**    Referee comment: P13, line 5-10, What is the percentage of stable northeasterly wind direction for the periods in Figure 5a?

Author Response: According the backward trajectory analysis, the percentage of NE and ENE trajectories is ~70 % during the wet season (i.e., Feb to May). A corresponding statement has been added in Sect., 2.7:

> "The BTs from the northern hemisphere (i.e., NE and ENE BT clusters) account for ~70 % of all BTs during the wet season. Among these ~70 %, ~30 % can be attributed to the NE BT clusters."

**[1.5]** Referee comment: Figure 5, 7 and 8, If possible, please do not overlap k(S, Da) with the size distribution plot. It is very hard to read the color map in k(S,Da).

Author Response: We modified the layout of the figures 5, 7, 8, and 9, trying to improve the readability (i.e., changing the shape and thickness of $\kappa(S,D_a)$ markers). However, we prefer to keep the overlaid kappa and size distribution representation since this layout emphasizes best the corresponding patterns in both data sets and their close agreement, which is an important aspect of the corresponding discussion. In general, we are aware that the time series figures 5, 7, 8, and 9 are rich in data and therefore rather complex. Accordingly, we implement these (and all other figures) in high resolution for the revised version of the manuscript, which allows to zoom into the pdf files electronically in order to explore the details of the figures.

**[1.6]** Referee comment: P15, line 5-10, The results here are not well supported. Andreae et al. 2017 showed that the UT particles consist predominately of organic material for aerosol size larger than 90 nm. For aerosol less than 60 nm, AMS had a hard time to determine chemical composition with good sensitivity.

Author Response: It is true that the sensitivity of AMS measurements drops with decreasing particle size. However, we feel that our statement on the proposed organic composition of the Aitken mode under *PR* conditions is well supported by (i) our observation of relatively low kappa levels, which suggest the presence or organic material and (ii) the concept of upper tropospheric nucleation and growth of ultra-fine particles as proposed by Andreae et al., 2018, which allow the plausible assumption that the chemical composition of the ~60 nm and ~90 nm particle is not fundamentally different.

**[1.7]** Referee comment: P16, section 3.5, The Saharan dust confirmed by EDX are larger than 1 micron. The CCN discussed in this section are in much smaller size range. It is confusing to classify the LRT influence as Saharan dust influence.

Author Response: The LRT plumes contain rather complex (internal) mixtures of African smoke, Saharan dust, and marine aerosols. We feel that the manuscripts is clear in stating that the LRT plumes not *only* contain Saharan dust but also further components such as sea salt, smoke etc. Examples can be found in Sect. 3.2:

> "These LRT plumes mostly comprise a complex mixture of Saharan dust, African biomass burning smoke, and marine aerosols from the transatlantic air passage (e.g., Talbot et al., 1990; Swap et al., 1992; Gläser et al., 2015)"

and Sect. 3.5:

> "The African *LRT* plumes that frequently impact the Amazon Basin during the wet season comprise a complex mixture of different aerosols components, including (i) a fraction of Saharan dust (mostly >1 µm), (ii) biomass burning aerosols from fires in West Africa (mostly in the accumulation mode), and (iii) marine aerosols from the plume's transatlantic passage (in coarse and accumulation modes) (Andreae et al., 1986; Talbot et al., 1990; Swap et al., 1992; Weinzierl et al., 2017)."

In fact, the referee is right that the Saharan dust component in the complex LRT plumes is likely not the key component that primarily influences the CCN population. Instead, the sea spray and African smoke components likely play a more important role in altering the $\kappa(S,D_a)$ and particle size distributions as they reach further into the relevant CCN size range. These aspects are discussed in detail in the manuscript.

A further aspect worth mentioning is that most of the dust particles are indeed >1 µm. However, a certain fraction has been found also <1 µm by Weinzierl et al. (2017) in the Caribbean region. These results show that a dust influence on the CCN population cannot be excluded completely. The reference to the study by Weinzierl et al. (2017) has been newly added to the manuscript. A study, which analyzes the size ranges of the dust, salt and smoke fractions in the LRT plumes at ATTO is currently being prepared. First results of this analysis show – similar to Weinzierl et al. (2017) – that certain dust contributions can also be found in the submicron size range and thus reach into the CCN relevant range discussed here.

**[1.8]** Referee comment: P18, line 23-24, from Table 3, except LRT case, the rest of cases all have reasonably good agreement between kp and k(0.11%). It is very stretching to state the LRT case is in a good agreement.

Author Response: In the corresponding statement we say that $\kappa_p$ and $\kappa(0.11\%)$ "agree reasonably well", which we still regard as an appropriately careful wording. Evidently, large fractions of the LRT aerosols are not detected by the ACSM, which only measures non-refractory components. This instrumental limitation likely explains the relatively large difference between $\kappa_p$ and $\kappa(0.11\%)$. This is stated as follows in the corresponding text section: "The remaining difference between $\kappa_p$ and $\kappa(0.11\ \%)$ can likely be explained by further refractory inorganics that were not covered by the ACSM."

**[1.9]** Referee comment: P19, section 31, authors said that ": : :correspond to a clear drop in aerosol hygroscopicity: : :". Please clarify the "drop", compared to what cases?

Author Response: Thanks for pointing out that this aspect lacks clarity. We modified the corresponding statement to:

> "At the same time, the presence of the pyrogenic aerosols correspond to a clear drop in aerosol hygroscopicity in both, the Aitken ($\Delta\kappa_{Ait} \approx$ -0.05) and accumulation modes ($\Delta\kappa_{Acc} \approx$ -0.1), relative to the conditions before and after the major *BB* plume (see overlay of $\kappa(S,D_a)$ size distributions and the d$N$/dlog$D$ contour plot in Fig. 8d)."

**[1.10]** Referee comment: P22, line 37, what is the OA/SO4 ratio for LRT pollution periods in Figure 7? Are they consistent with this case, around 3?

Author Response: The OA/$SO_4^{2-}$ ratios of the *LRT* and *MPOL* case studies are not consistent (see corresponding results in Table 3). Actually, they should not be consistent since the wet season LRT plumes and the LRT influence during the *MPOL* period in the dry season transport rather different aerosol mixtures. However, we are aware that using the abbreviation LRT in both contexts tends to be confusing. Accordingly, we revised the corresponding sections in the text to clarify that the wet season LRT influence is typically quite different from the dry season LRT influence.

**[1.11]** Referee comment: Figure 10, the different shape of the CCN efficiency spectra may related to the mixing state of aerosols for each case. It will be interesting to include that discussion.

Author Response: That is correct. The mixing state of the aerosol has an influence on the shape of the CCN efficiency spectra. To clarify this aspect, the following statement has been added to Sect. 3.9:

> "Their shape is influenced by (i) the shape of the $N_{CN}$ size distribution (i.e., relative strength of Aitken vs. accumulation modes), (ii) the aerosol composition through the $\kappa(S,D_a)$ values and its

size dependence as well as (iii) the mixing state of the aerosol as represented in the $\kappa$ and $N_{CN}\,\kappa$ distributions."

To what extent the mixing state defines the shape of the spectra cannot be answered easily and requires an in-depth analysis. We agree with the referee's suggestion that the mixing state of the aerosol population has not been adequately addressed in the manuscript yet. Therefore, we conducted a thorough analysis of the aerosol mixing state with the help of the hygroscopicity distribution concept introduced by Su et al. (2010). Since this analysis provides interesting additional insights into the properties of the contrasting CCN conditions, we decided to include a new figure (i.e., Fig. 10 in the revised version) and new dedicated text paragraph (Sect. 3.8 in the revised version) into the manuscript to discuss the relevant aspects as follows:

[revised manuscript text omitted]

We appreciate the very thorough and helpful comments by Referee #2, which have been considered carefully and helped to improve the quality of our manuscript. The referees' comments and our responses are outlined in detail below:

**[2.1]** Referee comment: The meaning of "near-pristine" is not very well defined. In section 2.3 the definition of NP excludes periods with concentrations of EBC that surpass a certain threshold to identify "the cleanest aerosol conditions" (p. 6, l. 22). Clean aerosol conditions and pristine conditions are however not the same. "Pristine" hints towards a natural aerosol state. Before anthropogenic emissions natural aerosol conditions would have included LRT of Saharan dust and natural forest fire emissions occasionally. Hence, defining "near-pristine" as the absence of any other natural aerosols and only taking into account the rain forest generated particles is not representative of the variability of pristine conditions. Calling them "near"-pristine does not help, because this is with reference to the unavoidable anthropogenic influences. From how the "near-pristine" conditions are identified and described, I understand that they are meant to reflect unperturbed aerosol conditions that are dominated by the Amazon rain forest as particle and precursor source. It is also evident from page 12, l. 39 that the authors hope to provide data for the preindustrial-like Amazonian atmosphere with the near-pristine cases. In this context it is even more important to acknowledge that clean cases exclude other conditions as mentioned above that were however present in the preindustrial time. For that reason, I urge the authors to rename their "near-pristine" classification into "clean" conditions as this reflects the actual definition much better, "near-pristine" is misleading. As more and more work is being done to extract preindustrial-like information in various environments it is important to be clear about the terminology.

Author Response: We understand the referee's concerns and tried to clarify the corresponding aspects. Please refer to our detailed response to comment **[2.2]**, which is closely linked to comment **[2.1]**.

**[2.2]** Referee comment: I am also surprised that "near-pristine" periods are only defined via EBC. Why did the authors not include CO and ozone mixing ratios or back trajectories which are all at hand? If particles were removed through precipitation, EBC can be low, while CO is elevated and hence the air mass cannot be classified as "near-pristine" or rather clean, even though the focus is on aerosol particles.

Author Response: We revised our analysis on pristine and near-pristine conditions at ATTO very carefully and extended the corresponding analysis. Besides black carbon, we have now taken CO, total aerosol concentrations, and backward trajectories into account. Based on these markers we defined four different filters to approximate pristine conditions to the best of our instrumental capabilities. We agree with the referee that the wording of measured or modelled (near-)pristine conditions is crucially important as this topic is quite complex. Therefore, we precisely define what we consider as pristine states at ATTO. Moreover we discuss the corresponding limitations of our definitions. The former Sect. 2.3 (now Sect. 2.7) has been extended substantially and now discusses the role and relevance of pristine conditions at ATTO as follows:

[revised manuscript text omitted]

**[2.3]**    Referee comment: The results of the paper would be even more informative if the authors provided the fraction in time throughout which the four different conditions are present. How often do clean conditions prevail? How often the mixed-polluted? It would be very informative if these simple statistics could be added.

Author Response: Thank you for the suggestion. We have added the requested statistics in a dedicated new paragraph in Sect. 1.2 as follows:

"Second, we will analyze the following four case studies that represent characteristic events and conditions in the central Amazon region:
- During certain wet season episodes, when no tracers of pollution aerosols are detectable any more, the aerosol population can be regarded as empirically not distinguishable from pristine, i.e., completely unpolluted rain forest conditions. This empirically pristine state of the rain forest (*PR*) aerosol prevails during 10 to 40 days per year (depending on *PR* definition, see Sect. 2.7).
- Long-range transport (*LRT*) aerosol advection during the wet season brings Saharan dust, African biomass burning smoke, as well as marine aerosol particles from the transatlantic passage. The *LRT* case study represents conditions that prevail between 50 and 60 days per year (see Moran-Zuloaga et al., 2017).

- Biomass burning (*BB*) smoke from man-made forest fires in the various deforestation hotspots in the basin influences the atmospheric state at ATTO almost permanently during the dry season and for extended episodes during the transition periods (>100 days per year) (Saturno et al., 2017a). The *BB* case study in this work analyzes large deforestation fires in the southeastern basin, whose smoke reached ATTO after few days of atmospheric processing. Accordingly, the *BB* case study characterizes the typical conditions of aged smoke influencing the atmospheric state at ATTO.
- Mixed pollution (*MPOL*) from African *LRT* and local/regional fires represents a frequent aerosol scenario at ATTO (Saturno et al., 2017a). The advected African aerosols mainly comprise biomass and fossil fuel combustions emissions, although the exact composition of these dry season *LRT* plumes is still poorly analyzed. The *MPOL* case study focusses on a period when African volcanogenic aerosols were advected to ATTO – an event that has been well documented in Saturno et al., 2017b. We selected this episode since the microphysical properties of the volcanogenic aerosol are characteristic enough to discriminate them from the local/regional smoke emissions. Accordingly, the alternating pattern of *LRT* vs. local/regional pollution can be clearly resolved for the *MPOL* period. However, note that volcanogenic plumes are comparatively rare events, whereas African combustion emissions, which are much harder to discriminate from the local/regional emissions, are a more common scenario. Accordingly, the *MPOL* case study is an example for a complex aerosol mixture due to alternating African vs. local/region influences during the dry season."

**[2.4]** Referee comment: The paper is well written, however there many qualifiers and redundancies in the text. In the technical comments section some instances are mentioned, but it would be helpful if the authors worked through the manuscript again to remove those. Additionally, the use of some terms is rather unusual. The term "pulses" should be replaced by "events". Also the precise meaning of "aerosol cycling" needs to be explained. Is cloud processing meant, or atmospheric processing? Please use the exact description. The word "trend" is frequently used in contexts, where no trends can be observed. The meaning is rather "variability, curve shape etc.". Below this is pointed out in detail.

Author Response: We agree with the referee and appreciate the hints. We have addressed the semantic aspects brought up to clarify our statements as requested. Furthermore, we worked through the entire manuscript and removed unnecessary qualifiers and redundancies. Details are outlined in the technical comment section below.

**[2.5]** Referee comment: Please go through the references again. Often XY et al. yyyyb comes before "a".

Author Response: Has been corrected.

**[2.6]** Referee comment: p.2, l. 14: At this point it is not clear what the efficiency spectrum is, a brief explanation is required.

Author Response: We have implemented the following explanation for the CCN efficiency spectrum into the corresponding sentence: "… (CCN divided by aerosol number concentration plotted against water vapor supersaturation) …".

**[2.7]** Referee comment: p.2, l. 27: the reference to volcanic emissions is confusing without any explanation, either provide one or do not mention volcanic influence here.

Author Response: We have revised the abstract to make it clearer and more concise in this (and various other aspects). The relevant sentence here have been changed to:

> "Mixed pollution (*MPOL*) conditions with a superposition of African and Amazonian aerosol emissions during the dry season. During the *MPOL* episode presented here as a case study, we observed African aerosols with a broad monomodal distribution ($D \approx 130$ nm, $N \approx 1300$ cm$^{-3}$), with high sulfate mass fractions from volcanic sources ($\sim 20$ %), and correspondingly high hygroscopicity ($\kappa_{<100nm} \approx 0.14$, $\kappa_{>100nm} \approx 0.22$) that were periodically mixed with fresh smoke from nearby fires ($D \approx 110$ nm, $N_{acc} \approx 2800$ cm$^{-3}$) with an organic-dominated composition and sharply decreased hygroscopicity ($\kappa_{<150nm} \approx 0.10$, $\kappa_{>150nm} \approx 0.20$)."

**[2.8]**   Referee comment: p.2, l. 30f: No D_ait is provided for the size distributions, but kappa_ait is given. Be consistent in providing all information.

Author Response: The broad size distribution of the case study *MPOL* (previously called MixPol) did not allow a meaningful bimodal fit. Accordingly we used a monomodal fit to describe the distribution. Accordingly only one modal diameter is reported. However, since kappa shows a size dependence we reported kappa for the size range >100 nm and <100 nm. For clarification we placed $\kappa_{Ait}$ by $\kappa_{<100nm}$ and $\kappa_{acc}$ by $\kappa_{>100nm}$. Moreover, we edited the relevant text sections throughout the manuscript and Table 2 to clarify that only a monomodal fit is reasonable for the *MPOL* case study.

**[2.9]**   Referee comment: p.2, l. 33: Why is the sensitivity of the CCN population towards changing supersaturation not mentioned for the mixed pollution cases?

Author Response: The abstract has been revised completely. The sensitivity towards *S* for all cases is not outlined in the abstract any more. Instead, the following statement has been added:

> "The measurement results suggest that CCN activation and droplet formation in convective clouds are mostly aerosol-limited under *PR* and *LRT* conditions and updraft limited under *BB* and *MPOL* conditions."

**[2.10]**   Referee comment: p. 3, l. 36: "multi-month trends" are not trends but rather changes or variability across the time period, also l. 38, not diurnal "trends" rather "diurnal cycle"

Author Response: We have replaced "trends" by "variability" as suggested.

**[2.11]**   Referee comment: p. 4, l. 1: Include info on what is the relevant size range?

Author Response: We have removed "CCN-relevant size range" to avoid confusion here.

**[2.12]**   Referee comment: p. 4, l. 7: please provide a reference for "previous studies"

Author Response: The companion part 1 paper contains a literature synthesis section with all previous CCN studies in the Amazon region. Thus, we have added the following reference:

"… (see M. Pöhlker et al., 2016 and references therein) …".

**[2.13]**   Referee comment: p.4, l. 13f: Provide information on what the relevant supersaturation range is for the ATTO region. Regarding the introduction, I would have expected a short paragraph that introduces the relevant literature on the ATTO site on which this study builds. After reading the abstract one expects to learn at least more about air mass transport towards ATTO in the introduction. Please introduce a short paragraph on this.

Author Response: We have added the following statement in the introduction (page 3) to specify the relevant supersaturation range in the ATTO region:

> "The relevant peak $S$ levels in the Amazon are assumed to range from ~0.1 % to ~1 % (e.g., Andreae, 2009; Reutter et al., 2009; Pöschl et al., 2010; C. Pöhlker et al., 2012; Farmer et al., 2015; M. Pöhlker et al., 2016). Moreover, a recent study suggests that also substantially higher $S$ ($\gg 1$ %) can be reached in deep convective clouds under certain conditions (Fan et al., 2018). However, a systematic and quantitative assessment of relevant peak $S$ distributions in the Amazonian atmosphere is still lacking (see discussion in Sect. 3.9)."

Moreover, the following section on relevant supersaturation levels has been included in the discussion in Sect. 3.9:

> "The peak supersaturation and its variability is an essential parameter to understand aerosol-cloud interactions. However, only spares quantitative information on the actually relevant $S$ range is available. The spectra in Fig. 11 are plotted for a broad $S$ range from 0.001 to 20 %. According to Reutter et al. (2009), this range can be generally subdivided into very low ($S < 0.1$ %), low ($S < 0.2$ %), transitional ($0.2$ % $< S < 0.5$ %), and high ($S > 0.5$ %) supersaturation regimes. Furthermore, a recent study by Fan et al. (2018) suggests that under certain conditions even very high supersaturations ($S \gg 1$ %) can be reached in deep convective clouds. One approach to actually quantify ATTO-relevant average peak $S$ at cloud base, $S_{cloud}(D_H,\kappa)$, uses the position of the Hoppel minimum, $D_H$, as outlined in Hoppel et al. (1986) and Krüger et al. (2014). According to results in our companion part 1 paper and the present study (see Table 2), ATTO-relevant $S_{cloud}(D_H,\kappa)$ values range from ~0.15 % to ~0.34 %. This $S_{cloud}(D_H,\kappa)$ range is shown in as a '$S$ landmark' in Fig. 11 for orientation."

In terms of characteristic air mass transport to ATTO, we have added the following paragraph in Sect. along with relevant references:

> "To explore essential biogeochemical processes, such as aerosol-cloud interactions, in the Amazon rain forest, the Amazon Tall Tower Observatory (ATTO) has been established in 2010/11 (for details see Andreae et al., 2015). The central Amazon Basin is characterized by a pronounced seasonality in atmospheric composition in response to the north-south oscillation of the intertropical convergence zone. During the wet season, the ATTO site receives comparatively clear air masses of marine origin from the northeast that travel over mostly untouched rain forest areas, whereas during the dry season strongly polluted air masses are advected from the southeast, originating from numerous fires in the Amazon's arc of deforestation (for details see C. Pöhlker et al., 2018). Detailed information on characteristic differences in atmospheric state and processes for the contrasting wet vs. dry season conditions in the ATTO region can be found in a number of recent studies (e.g., Nölscher et al., 2016; M. Pöhlker et al., 2016; Moran-Zuloaga et al., 2017; Saturno et al., 2017a, Yañez-Serrano et al., 2017)."

**[2.14]**  Referee comment: Why do the authors invent a new abbreviation for equivalent black carbon? The standard is "EBC" (Petzold et al., 2013). Please use this one.

Author Response: The abbreviation $BC_e$ has not been invented here, but has been used in numerous aerosol studies from the Amazon region and beyond (e.g., Andreae and Gelencser, 2006; Brito et al., 2014; Andreae et al., 2015; Moran-Zuloaga et al., 2017; Saturno et al., 2017). Obviously, different nomenclatures are co-existing here. In this particularly case, we prefer to keep $BC_e$ in order to be consistent at least with the studies on Amazonian aerosols.

**[2.15]**  Referee comment: The equations are not numbered. There is no need to repeat Eq. 1 without ammonium. It suffices to describe the change in the text as it is trivial.

Author Response: Numbers have been added to the equations. Moreover, the repeated equation has been deleted as suggested.

**[2.16]**  Referee comment: p. 5, l. 32: Generally the phrase is "sth. corresponds to" not "with". Please replace all cases throughout the manuscript.

Author Response: Agree. We found 6 sentence, in which "with" has been replaced by "to".

**[2.17]**  Referee comment: In the methodology section, information on how the CCNC was operated is needed. Even though this is described in the Part 1 paper, readers should get the essential information here (e.g., monodisperse mode, duration of a scan, covered range of diameters).

Author Response: Agree. We have added the following explanation:

> "Briefly, size-resolved CCN measurements were conducted using a continuous-flow streamwise thermal gradient CCN counter (model CCN-100, DMT, Longmont, CO, USA) in combination with a differential mobility analyzer (model M, Grimm Aerosol Technik, Ainring, Germany) and a condensation particle counter (Grimm Aerosol Technik). The DMA-selected size range spans from 20 to 245 nm. The analyzed supersaturation range spans from 0.11 to 1.10 %. A complete measurement cycle with scanning of all particle diameters and supersaturations took ~ 4.5 h."

**[2.18]**  Referee comment: p. 6, l. 10: What was done with a2? Was there an assumption or was it fitted?

Author Response: In the discussion paper, this aspect was indeed incorrect - thanks for pointing this out! We have reworked the erf fitting procedure in Sect. 2.2 and eliminated the error. The revised version of Sect. 2.2. reads as follows:

> "The CCN efficiency spectra parameterization as introduced in part 1 plays a key role in the present manuscript. Note that we slightly revised and improved the fitting procedure from the companion part 1 paper. The main change implies that the fits are now forced through zero, which is physically more plausible and makes the erf fit parametrization more applicable for modelling studies. The single-error function (erf) fit (mode = 1) is represented by the following function

$$\frac{N_{CCN}(S)}{N_{CN,10}} = \frac{a_1}{2} + \frac{a_1}{2} \cdot erf\left(\frac{\ln\frac{S}{S_1}}{w_1}\right) \qquad (3)$$

with $a_1$ as prefactor, $S_1$ as the supersaturation, at which half of the maximum activated fraction (*MAF*) of the aerosol particles acts as CCN (e.g., 50 % for $a_1 = 1$), and $w_1$ as the width of the erf fit. To simplify the fitting procedure, $a_1 = 1$ was assumed. For $a_1 = 1$ the erf converges against unity, corresponding to an activation of all particles as CCN at high $S$, which is adequate in most cases. Analogously, the double-erf fit (mode = 2) is represented by the function

$$\frac{N_{CCN}(S)}{N_{CN,10}} = \frac{a_1}{2} + \frac{a_2}{2} \cdot erf\left(\frac{\ln\frac{S}{S_1}}{w_1}\right) + \left(\frac{a_1-a_2}{2}\right) \cdot erf\left(\frac{\ln\frac{S}{S_2}}{w_2}\right) \qquad (4)$$

with the index 1 and 2 specifying the variables for both modes. To simplify the fitting procedure, $a_1 = 1$ was assumed.

Note further that in part 1 we tested different reference aerosol number concentrations, $N_{CN,Dcut}$ (e.g., $N_{CN,10}$ and $N_{CN,50}$), for the CCN efficiency spectra parametrization. In this study, we use only one reference concentration for clarity – namely $N_{CN,10}$. The choice of $N_{CN,10}$ can be explained by the fact that it is experimentally rather easily accessible (e.g., via condensation particle counter, CPC, measurements), whereas reference concentrations such as $N_{CN,50}$ require more elaborated experimental setups (e.g., scanning mobility particle sizer, SMPS, data)."

**[2.19]**  Referee comment: p. 7, l. 6: Are the 1000 m above sea level or above ground? If a.s.l. how high above the measurement site?

Author Response: 1000 m refers to above ground. We have added "(above ground level)" to specify this aspect.

**[2.20]**  Referee comment: p. 9, l. 31: Does this statement refer to the Amazon or is it a general statement?

Author Response: This is a general statement. We specified it by adding "at ATTO and further locations worldwide".

**[2.21]**  Referee comment: There is no need for footnotes. They can all be included in the main text which is already quite detailed. It is not apparent why some details are moved to footnotes. Specifically on p. 10, delete the 2nd sentence in the footnote, because it is a repetition.

Author Response: We have revised and streamlined the footnotes to remove unnecessary repetitions.

**[2.22]**  Referee comment: p. 11, first paragraph: What about the influence of more local anthropogenic emissions?

Author Response: Please refer to our response to comment [1.2], which addresses this questions.

**[2.23]**  Referee comment: p. 16, l. 4: What is meant by "perception"? Please clarify.

Author Response: To clarify this aspect, we have replaced the sentence:

> "This perception of CCN efficiency spectra stands in certain analogy to a concept introduced by Reutter et al. (2009) …"

by:

> "The CCN efficiency spectra can be linked to a concept introduced by Reutter et al. (2009) …"

**[2.24]**   Referee comment: p. 18, l. 19f: Is the fraction of long-range transported mineral dust and sea salt really large? Please include the values in the text: What is the fraction of supermicron dust and salt in mass or number concentration?

Author Response: The influence of dust and seas salt can be remarkably strong during certain episodes. In several relevant sections of the text, we added and emphasized the reference to a recent ATTO study by Moran-Zuloaga et al., 2017, which provides a detailed discussion of the role and relevance of long-range transported dust and sea salt in the Amazonian atmosphere.

**[2.25]**   Referee comment: p. 20, l. 21: How can the ratios of 26 and 49 be consistent? That's a factor of 2.

Author Response: Emission ratios such as $\Delta N_{CN}/\Delta c_{CO}$ and $\Delta N_{CCN}(S)/\Delta c_{CO}$ typically reveal quite some variability as a function of fuel type and burning phase (flaming vs. smoldering). Accordingly, emission ratios are characteristic within certain ranges - for example $\Delta N_{CN}/\Delta c_{CO} = 30$ +/- 15 cm$^{-3}$ ppb$^{-1}$ for vegetation fires. With respect to these uncertainties, the results by Kuhn et al. (2010) ($\Delta N_{CCN}(0.6\%)/\Delta c_{CO} = 26$ cm$^{-3}$ ppb$^{-1}$ for biomass burning) and our results ($\Delta N_{CCN}(0.61\%)/\Delta c_{CO} = 18$ cm$^{-3}$ ppb$^{-1}$ also for biomass burning) are indeed quite consistent. To clarify this statement, we removed the results for urban plumes ($\Delta N_{CCN}(0.6\%)/\Delta c_{CO} = 49$ cm$^{-3}$ ppb$^{-1}$) by Kuhn et al., (2010), which are only peripheral information here and may lead to confusion. Moreover, we added "(i.e., $\Delta N_{CCN}(0.61\%)/\Delta c_{CO} = 17.9 \pm 1.3$ cm$^{-3}$ ppb$^{-1}$, see Table S4)" for clarification.

**[2.26]**   Referee comment: p. 21, l. 32: Define aerosol "cycling"

Author Response: We have removed the entire sentence "Ultimately, these observations are discussed in the light of the related CCN cycling." as it is not essential here and may lead to confusion.

**[2.27]**   Referee comment: p. 22, l. 29: The African volcanic emission appear out of nowhere. Were they never observed before at ATTO? Why can the authors be so sure that they have observed volcanic emissions?

Author Response: A recent study by Saturno et al., 2017b ("b" in the revised version) analyzes the African volcanogenic plume that impacted ATTO during Sep 2014 in details. The reported evidence for volcanogenic influence on the aerosol composition in the central Amazon is quite convincing. We have added the reference to this study in the corresponding section of the paper for clarification.

**[2.28]**   Referee comment: p. 24, l. 35: The here defined NP conditions are very selective because they describe only clean conditions. They are hence not an approximation of a preindustrial state, but represent only a potential fraction of the preindustrial atmosphere in that region.

Author Response: See our response to comment **[2.2]**.

**[2.29]**  Referee comment: p.2, l. 24: What do you mean by "CCN cycling in relation to aerosol-cloud interactions"?

Author Response: The abstract has been revised and the corresponding statement remove for clarity.

**[2.30]**  Referee comment: p.3, l. 20: change to "Ndb ranges from few hundred: : : for clean to 1000 and : : : for polluted conditions"

Author Response: Has been changed.

**[2.31]**  Referee comment: p. 3, l. 32: delete "of this endeavor"

Author Response: Deleted.

**[2.32]**  Referee comment: p. 4, l. 3: D_a has not been defined yet.

Author Response: We added "with $D_a$ as midpoint activation diameter" to define it.

**[2.33]**  Referee comment: p. 4, l. 10f: Delete the last sentence of the paragraph, this information is not really needed and the paper can benefit from being shortened.

Author Response: Done.

**[2.34]**  Referee comment: p. 4, l. 7: Delete " As a particularly: : :concept". This is a qualifier that does not help the reader to understand the efficiency spectra better.

Author Response: Deleted.

**[2.35]**  Referee comment: p.4, l. 29f: delete "particularly interesting"

Author Response: Deleted.

**[2.36]**  Referee comment: p. 4, l. 38: What is an "efficient CCN prediction?" Is this method particularly fast or do you mean effective?

Author Response: "efficient" has been replaced by "effective". Thanks.

**[2.37]**  Referee comment: p. 5, l. 19: delete "predicted"

Author Response: Deleted.

**[2.38]**   Referee comment: P. 9, l. 4: Spell out Mar

Author Response: Done

**[2.39]**   Referee comment: p. 9, l. 7f: The sentence beginning with "First, the: : :" is not a grammatically correct sentence.

Author Response: True. The sentence has been modified to " First, $P_{TRMM}$ data represents the area-averaged precipitation rate in the $ROI_{ATTO}$ (see Fig. S1)."

**[2.40]**   Referee comment: p.9, l. 17: a reference is missing

Author Response: The following reference has been added: "(e.g., Moran-Zuloaga et al., 2017)".

**[2.41]**   Referee comment: p. 9, l. 28: replace "trend" by "behavior"

Author Response: Replaced.

**[2.42]**   Referee comment: p. 9, l. 32: Shorten this paragraph: "Figure 1 provides: : : sections will focus on detailed aerosol and CCN characteristics in wet: : : year 2014." Delete the rest as is has been mentioned already.

Author Response: Modified as suggested.

**[2.43]**   Referee comment: p. 10, l. 28: delete "in the context of this study".

Author Response: Deleted.

**[2.44]**   Referee comment: p. 11, l. 10: replace "trend" with "observations"

Author Response: Done.

**[2.45]**   Referee comment: p. 11, l. 13 ff: delete this sentence, it has already been said.

Author Response: Deleted.

**[2.47]**   Referee comment: p. 11, l. 31: replace "in more detail of" with "on"

**[2.48]**   Referee comment: p. 12, l. 1: replace "pendant" with "counterpart"

Author Response: Replaced.

**[2.49]**   Referee comment: p. 12, l. 13: Do you mean N_cn10? There are a couple of other instances where "10" has been omitted.

Author Response: Has been corrected.

**[2.50]**   Referee comment: p. 12, l. 28: delete "interesting" this qualifier is not needed.

Author Response: Deleted.

**[2.51]**   Referee comment: p. 13, l. 16: What is meant by aerosol cycling here?

Author Response: "Cycling" has been replaced by "variability".

**[2.52]**   Referee comment: p. 14, l. 39: delete "mode"

Author Response: Has been deleted.

**[2.53]**   Referee comment: p. 15, l. 14: replace "ranged" with "was", and its "contributions"

Author Response: Done.

**[2.54]**   Referee comment: p. 15, l. 29: delete "solid"

Author Response: Done

**[2.55]**   Referee comment: p. 15, l. 35: write just "a tool"

Author Response: Done.

**[2.56]**   Referee comment: p. 16, l. 34f: remove "as a particularly instructive example

Author Response: Removed.

**[2.57]**   Referee comment: p. 17, l. 11: shouldn't the units of SW_in the figures be W per square meter?

Author Response: Yes – has been changed.

**[2.58]**   Referee comment: p. 17, l. 24: A "LRT pulse" does not make sense, replace by "LRT event"

Author Response: This has been changed: "pulses" has been replaced by "events".

**[2.59]**   Referee comment: p. 17, l. 28: remove "exactly"

Author Response: Removed.

**[2.60]**   Referee comment: p. 18, l. 10: Do the authors refer to natural or anthropogenic fires, or both?

Author Response: The fires in Africa are to significant extent of anthropogenic origin. This clarify this aspects, we added the following sentence: "Note that the African fires in West Africa are to a large extent agriculture-related and, thus, man-made (e.g., Barbosa et al., 1999; Capes et al., 2008)."

**[2.61]**   Referee comment: p. 19, l. 20: remove "fortunate"

Author Response: Removed.

**[2.62]**   Referee comment: p. 19, l. 23: remove "rather"

Author Response: Removed.

**[2.63]**   Referee comment: p. 20, l. 27: remove the sentence beginning with "It: : :". This has already been said.

Author Response: Has been removed.

**[2.64]**   Referee comment: p. 21, l. 28: Delete this sentence, redundancy.

Author Response: Deleted.

**[2.65]**   Referee comment: p. 21, l. 31: delete "pulses"

Author Response: Deleted.

**[2.66]**   Referee comment: p. 22, l. 24: delete "such as industrial emissions, : : :River", This has already been said.

Author Response: Deleted.

**[2.67]** Referee comment: p. 22, l. 33: delete "far""

Author Response: Done.

**[2.68]** Referee comment: p. 23, l. 34f: shorten to "CCN efficiency spectra serve as CCN signatures."

Author Response: Has been shortened.

**[2.69]** Referee comment: p. 23, l. 38: delete "trends" because trends are not discussed in this paper.

Author Response: Deleted.

**[2.70]** Referee comment: p. 23, l, 39: Why "could"? Can they or can they not?

Author Response: "Could" has been replaced by "can".

**[2.71]** Referee comment: p. 24, l. 1: it seems that D_H has not been introduced.

Author Response: We have added "… uses the position of the Hoppel minimum, $D_H$, …" to define $D_H$.

**[2.72]** Referee comment: p. 24, l. 12f: replace "trends" with "curve shapes", and again with "shapes"

Author Response: Done.

**[2.73]** Referee comment: p. 24, l. 17: replace "pulses" with "events"

Author Response: Done.

**[2.74]** Referee comment: p. 24, l. 38: replace "pulses" with "events"

Author Response: Done.

**[2.75]** Referee comment: p. 25, l. 9: delete "very"

Author Response: Deleted.

**[2.76]** Referee comment: p. 25, l. 15f: "growth as well as to enhance the : : :" otherwise the sentence is grammatically not correct.

Author Response: Thanks – has been corrected.

**[2.77]**  Referee comment: p. 25, l. 33: Be more specific regarding what is meant with cycling.

Author Response: We agree – "aerosol cycling and cloud properties" has been replaced by "aerosol-cloud interactions".

**[2.78]**  Referee comment: p. 26, l. 5 delete "instructive"

Author Response: Deleted.

**[2.79]**  Referee comment: p. 26, l. 14: delete "very" and "highly"

Author Response: Both have been deleted.

**[2.80]**  Referee comment: p. 26, l. 22: "The aerosol particles are composed of: : :"

Author Response: "comprise" has been replaced by "are composed of".

**[2.81]**  Referee comment: p. 26, l. 31: "on event-basis" instead of "pulse-wise"

Author Response: Has been changed.

**[2.82]**  Referee comment: p. 27, l. 11: "mixed" instead of "superimposed"

Author Response: Has been changed.

**[2.83]**  Referee comment: p. 27, l. 25: "behaviors" instead of "trends"

Author Response: "differences, similarities and trends" has been replaced by "observations".

**[2.84]**  Referee comment: p. 27, l. 25 f: The sentence starting with "The array of : : :." is cryptic for saying that there exists large variability, rephrase.

Author Response: That is true. The sentence has been removed in the course of a major revision of the summary and conclusion section.

[revised manuscript text omitted]

**List of changes**

All relevant changes have been marked blue in the revised version of the manuscript. The most important changes are listed below.

Title:

The title has been changed from:

[revised manuscript text omitted]

Author List:

The following persons have been added as co-authors:
Bruna A. Holanda[1] and Ovid O. Krüger[1]
[1]*Multiphase Chemistry and Biogeochemistry Departments, Max Planck Institute for Chemistry, 55020 Mainz, Germany.*

The Author affiliation and Author names have been changed from:
Alessandro Araùjo[2]
Jorge Saturno[1]
Joel Brito[3,a]
Samara Carbone[3,b]
Xuguang Chi[1,c]
Diana Rose[11]
Ryan Thalman[12,d]
[1]*Multiphase Chemistry and Biogeochemistry Departments, Max Planck Institute for Chemistry, 55020 Mainz, Germany.*
[2]*Empresa Brasileira de Pesquisa Agropecuária (EMBRAPA), Belém, PA, Brazil*
[3] *Institute of Physics, University of São Paulo, São Paulo 05508-900, Brazil.*
[11] *Institut für Atmosphäre und Umwelt, Goethe Universität, 60438 Frankfurt, Germany.*
[12] *Biological, Environmental & Climate Sciences Department, Brookhaven National Laboratory, Upton, NY 11973-5000, USA.*
[a]*now at: Laboratoire de Météorologie Physique, Université Clermont Auvergne, Aubière, France*
[b]*now at: Institute of Agrarian Sciences, Federal University of Uberlândia, Uberlândia, Minas Gerais, Brazil*
[c]*now at: Institute for Climate and Global Change Research & School of Atmospheric Sciences, Nanjing University, Nanjing, 210093, China*
[d]*now at: Department of Chemistry, Snow College, Richfield, UT, USA*

to:

Alessandro C. Araùjo[2]

Jorge Saturno[1,a]

Joel Brito[3,b,]

Samara Carbone[3,c]

Xuguang Chi[1,d]

Diana Rose[11,e]

Ryan Thalman[12,f]

[1]*Multiphase Chemistry and Biogeochemistry Departments, Max Planck Institute for Chemistry, 55020 Mainz, Germany.*

[2] *Empresa Brasileira de Pesquisa Agropecuária (EMBRAPA), Trav. Dr. Enéas Pinheiro, Belém, PA, 66095-100, Brazil.*

[3] *Institute of Physics, University of São Paulo, São Paulo 05508-900, Brazil.*

[11] *Institut für Atmosphäre und Umwelt, Goethe Universität, 60438 Frankfurt, Germany.*

[12] *Biological, Environmental & Climate Sciences Department, Brookhaven National Laboratory, Upton, NY 11973-5000, USA.*

[a] *now at: Physikalisch-Technische Bundesanstalt, Bundesallee 100, D-38116 Braunschweig, Germany.*

[b] *now at: Laboratoire de Météorologie Physique, Université Clermont Auvergne, Aubière, France.*

[c] *now at: Institute of Agrarian Sciences, Federal University of Uberlândia, Uberlândia, Minas Gerais, Brazil.*

[d] *now at: Institute for Climate and Global Change Research & School of Atmospheric Sciences, Nanjing University, Nanjing, 210093, China.*

[e] *now at: Hessian Agency for Nature Conservation, Environment and Geology, Rheingaustr. 186, 65203 Wiesbaden, Germany.*

[f] *now at: Department of Chemistry, Snow College, Richfield, UT, USA.*

Figures: All figures of the manuscript and the supplementary material have been modified regarding optimal clearness and to improve the graphic accessibility for colorblind. The former figure 3 has been replaced by the present figure 1. Figure 10 was added to the manuscript. CCN spectra were included in figure 11. In the supplementary material the following figures have been added: S2, S3, S4, S5, S6, S7, S8, S9. The former figure S7 has been removed.

Tables: All numbers have been recalculated, furthermore the standard deviation (std) of all means have been replaced by the standard error (se). Black carbon mass concentrations have been included in Table 3. Literature values and the corresponding fit have been included in Table 4. In the supplementary material the following tables have been added: S1, S2, S3.

General changes: The term "near-pristine (NP)" was replaced by "empirically pristine rain forest (PR)" in the entire study. The abbreviations for the mixed pollution case study were changed from "MixPol" to "MPOL", from "MixPol low κ" to "MPOL-BB" and from "MixPol high κ" to "MPOL-LRT".

Page 3, line 15: The following section has been added: To explore essential biogeochemical processes, such as aerosol-cloud interactions, in the Amazon rain forest, the Amazon Tall Tower Observatory

(ATTO) has been established in 2010/11 (for details see Andreae et al., 2015). The central Amazon Basin is characterized by a pronounced seasonality in atmospheric composition in response to the north-south oscillation of the intertropical convergence zone. During the wet season, the ATTO site receives comparatively clear air masses of marine origin from the northeast that travel over mostly untouched rain forest areas, whereas during the dry season strongly polluted air masses are advected from the southeast, originating from numerous fires in the Amazon's arc of deforestation (for details see C. Pöhlker et al., 2018). Detailed information on characteristic differences in atmospheric state and processes for the contrasting wet vs. dry season conditions in the ATTO region can be found in a number of recent studies (e.g., Nölscher et al., 2016; M. Pöhlker et al., 2016; Moran-Zuloaga et al., 2017; Saturno et al., 2017a, Yañez-Serrano et al., 2018).

Page 3, line 28: The following section has been added: The relevant peak $S$ levels in the Amazon are assumed to range from ~0.1 % to ~1.0 % (e.g., Andreae, 2009; Reutter et al., 2009; Pöschl et al., 2010; C. Pöhlker et al., 2012; Farmer et al., 2015; M. Pöhlker et al., 2016). Moreover, a recent study suggests that also substantially higher $S$ ($\gg$ 1 %) can be reached in deep convective clouds under certain conditions (Fan et al., 2018). However, a systematic and quantitative assessment of relevant peak $S$ distributions in the Amazonian atmosphere is still lacking (see discussion in Sect. 3.9).

The former page 7, line 5 has been replaced from:
(i) the cleanest observed episodes, which can be regarded as a near-pristine atmospheric state, (ii) long-range transport (LRT) aerosol advection during the wet season, which typically brings Saharan dust, African combustion aerosols, as well as marine aerosol particles, (iii) advection of biomass burning smoke from large deforestation fires in southeast Basin after few days of atmospheric processing, and (iv) prevalence of mixed pollution from African LRT and local fires, representing a frequent pollution scenario at the ATTO site.

To the present page 5, line 4:
- During certain wet season episodes, when no tracers of pollution aerosols are detectable any more, the aerosol population can be regarded as empirically not distinguishable from pristine, i.e., completely unpolluted rain forest conditions. This empirically pristine state of the rain forest (*PR*) aerosol prevails during 10 to 40 days per year (depending on *PR* definition, see Sect. 2.7).
- Long-range transport (*LRT*) aerosol advection during the wet season brings Saharan dust, African biomass burning smoke, as well as marine aerosol particles from the transatlantic passage. The *LRT* case study represents conditions that prevail between 50 and 60 days per year (see Moran-Zuloaga et al., 2017).
- Biomass burning (*BB*) smoke from man-made forest fires in the various deforestation hotspots in the basin influences the atmospheric state at ATTO almost permanently during the dry season and for extended episodes during the transition periods (>100 days per year) (Saturno et al., 2017a). The *BB* case study in this work analyzes large deforestation fires in the southeastern basin, whose smoke reached ATTO after few days of atmospheric processing. Accordingly, the *BB* case study characterizes the typical conditions of aged smoke influencing the atmospheric state at ATTO.
- Mixed pollution (*MPOL*) from African *LRT* and local/regional fires represents a frequent aerosol scenario at ATTO (Saturno et al., 2017a). The advected African aerosols mainly comprise biomass

and fossil fuel combustions emissions, although the exact composition of these dry season *LRT* plumes is still poorly analyzed. The *MPOL* case study focusses on a period when African volcanogenic aerosols were advected to ATTO – an event that has been well documented in Saturno et al., 2017b. We selected this episode since the microphysical properties of the volcanogenic aerosol are characteristic enough to discriminate them from the local/regional smoke emissions. Accordingly, the alternating pattern of *LRT* vs. local/regional pollution can be clearly resolved for the *MPOL* period. However, note that volcanogenic plumes are comparatively rare events, whereas African combustion emissions, which are much harder to discriminate from the local/regional emissions, are a more common scenario. Accordingly, the *MPOL* case study is an example for a complex aerosol mixture due to alternating African vs. local/region influences during the dry season.

Equation 2 has been changed from:

$$\kappa_p = f_{OA} \cdot 0.1 \ + \ (1 - f_{OA}) \cdot 0.6$$

to:

$$\kappa_p = f_{OA} \cdot 0.1 \ + \ f_{inorg} \cdot 0.71 \ + \ f_{BCe} \cdot 0$$

with $f_{inorg}$ including $SO_4^{2-}$, $NO_3^-$, $NH_4^+$, and $Cl^-$.

Former page 8, line 6: the equation has been removed.

Page 6, line 37: The following section has been added: Briefly, size-resolved CCN measurements were conducted using a continuous-flow streamwise thermal gradient CCN counter (model CCN-100, DMT, Longmont, CO, USA) in combination with a differential mobility analyzer (model M, Grimm Aerosol Technik, Ainring, Germany) and a condensation particle counter (Grimm Aerosol Technik). The DMA-selected size range spans from 20 to 245 nm. The analyzed supersaturation range spans from 0.11 to 1.10 %. A complete measurement cycle with scanning of all particle diameters and supersaturations took ~ 4.5 h.

Equations 3 and 4 have been changed from:
The single-error function (erf) fit (mode = 1) is represented by the following function

$$\frac{N_{CCN}(S)}{N_{CN,10}} = v_1 + a_1 \cdot erf \left( \frac{ln \frac{S}{S_1}}{w_1} \right)$$

with $v_1$ as offset, $a_1$ as prefactor, $S_1$ as the supersaturation, at which half of the maximum activated fraction (*MAF*) of the aerosol particles acts as CCN (e.g., 50 % for $v_1$ = 0.5), and $w_1$ as the width of the erf fit. To simplify the fitting procedure, $a_1$ = 0.5 was assumed. Further, $v_1$ = 0.5 can be used as an input parameter in most cases. The combination, $a_1$ = 0.5 and $v_1$ = 0.5, yields convergence of the erf against unity, corresponding with an activation of all particles as CCN at high *S*, which is adequate in most cases. Analogously, the double-erf fit (mode = 2) is represented by the function

$$\frac{N_{CCN}(S)}{N_{CN,10}} = v_1 + a_1 \cdot erf\left(\frac{ln\frac{S}{S_1}}{w_1}\right) + a_2 \cdot erf\left(\frac{ln\frac{S}{S_2}}{w_2}\right)$$

with the index 1 and 2 specifying the variables for both modes. To simplify the fitting procedure, $a_1$ = 0.5 and $v_1$ = 0.5 was assumed.

to:
Note that we slightly revised and improved the fitting procedure from the companion part 1 paper. The main change implies that the fits are now forced through zero, which is physically more plausible and makes the single-error function (erf) fit parametrization more applicable for modelling studies. The erf fit (mode = 1) is represented by the following function

$$\frac{N_{CCN}(S)}{N_{CN,10}} = \frac{a_1}{2} + \frac{a_1}{2} \cdot erf\left(\frac{ln\frac{S}{S_1}}{w_1}\right) \qquad (3)$$

with $a_1$ as prefactor, $S_1$ as the supersaturation, at which half of the maximum activated fraction (*MAF*) of the aerosol particles acts as CCN (e.g., 50 % for $a_1$ = 1), and $w_1$ as the width of the erf fit. To simplify the fitting procedure, $a_1$ = 1 was assumed. For $a_1$ = 1 the erf converges against unity, corresponding to an activation of all particles as CCN at high $S$, which is adequate in most cases. Analogously, the double-erf fit (mode = 2) is represented by the function

$$\frac{N_{CCN}(S)}{N_{CN,10}} = \frac{a_1}{2} + \frac{a_2}{2} \cdot erf\left(\frac{ln\frac{S}{S_1}}{w_1}\right) + \left(\frac{a_1-a_2}{2}\right) \cdot erf\left(\frac{ln\frac{S}{S_2}}{w_2}\right) \qquad (4)$$

with the index 1 and 2 specifying the variables for both modes. To simplify the fitting procedure, $a_1$ = 1 was assumed.

[revised manuscript text omitted]

Page 19 the name of section 3.5 has been changed from:
  **3.5 Case study *LRT* on Saharan dust influence**
to:
  **3.5 Case study *LRT* on Saharan dust, African smoke and Atlantic marine aerosols**

Page 19 the following footnote was added to the manuscript:
[1] For completeness, the CCN efficiency spectra for all four $PR$ filters according Sect. 2.7 (i.e., $PR_{BC}$, $PR_{CO}$, $PR_{BC \cup CO}$, $PR_{BC \cap CO}$) are shown in the comparison Figure S8.

Page 22, line 22 and table 1: The time for the BB case study has been adjusted from "17 to 23 August" to "18 to 22 August".

The former page 26, line 36 has been replaced:

[revised manuscript text omitted]

Page 31, line 22: The following section has been added: Hygroscopicity distributions were analyzed for all conditions, providing detailed and characteristic insights into the mixing state of the different types of aerosols. We found that the spread of $\kappa$ increases with size for all conditions. The $\kappa$ distributions suggest that the *PR* aerosol population is rather internally mixed, whereas the *BB*, *LRT*, and *MPOL* aerosols show more external mixing states. In essence, the $\kappa$ distributions and $N_{CN}$ $\kappa$ distributions represent valuable overview representations, combining characteristic aerosol properties in terms of particle size, particle concentration, $\kappa$ levels, and aerosol mixing state in a fingerprint-like manner. This representation helps to further understand aerosol-cloud interactions, such as the shapes of the CCN efficiency spectra: As general tendencies, more externally mixed aerosols, resulting in broader $\kappa$ distributions, also broaden the CCN efficiency spectra – analogous to broad $N_{CN}$ distributions. Internally mixed aerosols with more defined $\kappa$ distributions tend towards steeper segments in the CCN efficiency spectra. Furthermore, externally mixed aerosols with distinctly different $\kappa$ levels tend to introduce further steps/plateaus into the CCN efficiency spectra, in addition to plateaus caused by multimodal $N_{CN}$ distributions. However, the CCN efficiency spectra for the conditions reported here are primarily shaped by the particle size

distributions and average $\kappa$ levels, whereas the diversity of $\kappa$ seems to play a minor role. To clarify exactly how the signatures and patterns in $\kappa$ and $N_{CN}$ $\kappa$ distributions are related to the signatures and shapes of the CCN efficiency spectra, dedicated future studies at contrasting locations and modelling support is required.